# Differentiable Constraint-Based Causal Discovery

**Jincheng Zhou**[1]  **Mengbo Wang**[1]  **Anqi He**[2]  **Yumeng Zhou**[2]  **Hessam Olya**[2]
**Murat Kocaoglu**[3]  **Bruno Ribeiro**[1]
[1]Purdue University  [2]Ford Motor Company  [3]Johns Hopkins University
{zhou791,wang4887,ribeiro}@purdue.edu
{AHE6,yzhou173,molya}@ford.com
mkocaoglu@jhu.edu

## Abstract

Causal discovery from observational data is a fundamental task in artificial intelligence, with far-reaching implications for decision-making, predictions, and interventions. Despite significant advances, existing methods can be broadly categorized as constraint-based or score-based approaches. Constraint-based methods offer rigorous causal discovery but are often hindered by small sample sizes, while score-based methods provide flexible optimization but typically forgo explicit conditional independence testing. This work explores a third avenue: developing differentiable $d$-separation scores, obtained through a percolation theory using soft logic. This enables the implementation of a new type of causal discovery method: gradient-based optimization of conditional independence constraints. Empirical evaluations demonstrate the robust performance of our approach in low-sample regimes, surpassing traditional constraint-based and score-based baselines on a real-world dataset. Code and data of the proposed method are publicly available at https://github.com/PurdueMINDS/DAGPA.

## 1 Introduction

Causal discovery—the task of inferring causal relationships from observational data—is a fundamental problem in machine learning and statistics with far-reaching implications across multiple disciplines, including biology, economics, social sciences, and medicine [19, 25, 28, 33, 45]. Directed Acyclic Graphs (DAGs) provide a powerful framework for representing causal relationships, enabling the estimation of the effects of interventions or actions. However, in many complex systems, the underlying causal graphs remain unknown, and conducting randomized controlled trials (RCTs) to establish causal relationships can be prohibitively expensive or impractical. This has significant implications in various domains, including planning, explainability, and fairness [26, 47, 49], highlighting the need for reliable methods to infer causal structures from observational data alone.

The task of causal discovery is inherently challenging, specifically under small sample sizes. Considering Markov, acyclicity, and faithfulness assumptions [25, 34], some recent works have focused on two popular avenues [13, 39, 44]: differentiable score-based methods, which leverage differentiability to optimize objective functions, and constraint-based approaches, which rely on conditional independence tests to infer causal structures. Common methods for constraint-based causal discovery, such as the PC algorithm [34], are vulnerable to errors in datasets with small sample sizes, due to uncertainty in conditional independence tests (CI tests). More recent constraint-based methods, such as LOCI and $k$-PC [18, 42] among others, have ameliorated these challenges with smaller conditioning sets.

In recent years, significant progress has been made on addressing the challenges of small sample sizes in causal discovery via differentiable score-based methods, casting the combinatorial graph search problem as a continuous optimization problem on weighted adjacency matrices representing directed

39th Conference on Neural Information Processing Systems (NeurIPS 2025).

graphs [4, 36, 51, 52]. These differentiable methods rely either on linear models or increasingly complex neural networks to model the underlying functional relationship between variables.

**Contributions.** A promising avenue for advancing causal discovery, complementing existing approaches, is the development of hybrid methods [13, 37, 39]. Our work tackles a key challenge in this domain by introducing *differentiable functions of $d$-separation and $d$-connection*, capable of scoring conditional independencies within a probabilistic causal graph, bridging constraint-based and gradient-based methods. To achieve this, we propose a novel framework grounded in a *percolation measure* applied to a continuous relaxation of the causal graph structure. This new measure effectively captures the inherent dependencies among paths in a causal graph, addressing a significant limitation of conventional diffusion-based methods, such as matrix powers, which would be unable to adequately model $d$-separation.

The distinction between diffusion and percolation lies at the heart of deriving a differentiable metric for $d$-separation, which essentially boils down to measuring reachability over random graphs [16]. To illustrate this concept, consider a typical setup in gradient-based graph optimization, where a matrix $\boldsymbol{A} \in [0, 1]^{n \times n}$ represents edge probabilities [41, 51], with each entry $\boldsymbol{A}_{XY}$ encoding the likelihood of an edge from node $X$ to node $Y$. A straightforward approach to assess connectivity between $X$ and $Y$ might involve computing $\boldsymbol{A}_{XY}^n$, which represents the sum of all path probabilities of length $n$ from $X$ to $Y$. However, this method implicitly assumes independence between paths, akin to a diffusion process. For instance, in the paths $X \to Z \to Y$ and $X \to Z \to W \to Y$, the simultaneous absence of the sampled $X \to Z$ edge would invalidate both paths, introducing a probabilistic dependence that is not captured by matrix powers or diffusion in general.

In contrast, a $d$-separation measure over $\boldsymbol{A}$ is a type of percolation measure, which bounds the probability of sampling valid paths between $X$ and $Y$, accounting for edge dependencies. Although characterizing percolation involves combinatorial complexity, we derive a differentiable bound for $d$-separation using soft logic, enabling gradient-based optimization of causal structures. We instantiate this differentiable $d$-separation framework in an algorithm (DAGPA, for DAG Percolation Apartness), where we combine CI-based scores, state-of-the-art techniques in multi-task learning, and Bayesian sampling. While DAGPA instantiates this novel framework, we believe differentiable $d$-separation is of independent interest and can be integrated into existing gradient-based methods.

Empirically, DAGPA shows that differentiable $d$-separation yields strong results in the small-sample regime, demonstrating good robustness and accuracy compared to popular constraint-based and differentiable score-based baselines. Moreover, on the real-world Sachs dataset [28], DAGPA shows that differentiable $d$-separation offers accurate modeling of the independence patterns in the data, outperforming baselines in our metrics.

## 2 Notation and background

**Notation.** We denote the integer set $\{1, \ldots, d\}$ as $[d]$ and a dataset of $n$ samples and $d$ variables as $\mathcal{D}$. We represent a weighted directed graph with $d$ nodes and edge weights in the range $[0, 1]$ by a real square matrix $\boldsymbol{W} \in [0, 1]^{d \times d}$. Given a node $z \in [d]$, we denote $\boldsymbol{W}_{-z}$ the submatrix obtained by removing the $z$-th row and column of $\boldsymbol{W}$, which is equivalent to removing the node $z$ and their connecting edges from the graph. When the square matrix is binary, e.g. $\boldsymbol{A} \in \{0, 1\}^{d \times d}$, we interpret it as an unweighted directed graph. Here, we write $x \to_{\boldsymbol{A}} y$ to say "$y$ is connected from $x$ via a directed edge in $\boldsymbol{A}$," and $x \rightsquigarrow_{\boldsymbol{A}} y$ to say "$y$ is reachable from $x$ via a directed path (a path where every edge has direction from $x$ to $y$) in $\boldsymbol{A}$." Conversely, $x \not\to_{\boldsymbol{A}} y$ and $x \not\rightsquigarrow_{\boldsymbol{A}} y$ stand for negations of these statements. Finally, we use $\perp\!\!\!\perp_{\mathcal{D}}$ for (conditional) independence statements in the dataset $\mathcal{D}$ [1], and if $\boldsymbol{A}$ is acyclic, $\perp\!\!\!\perp_{\boldsymbol{A}}$ for d-separation statements in the DAG $\boldsymbol{A}$.

**Constraint-based causal discovery.** Methods in this category aim to identify a collection of DAGs from a dataset $\mathcal{D}$ such that each DAG $\boldsymbol{A}$ in it is an I-Map of $\mathcal{D}$, meaning all d-separations in $\boldsymbol{A}$ imply conditional independencies in $\mathcal{D}$ with a sufficiently large dataset. Namely, $\forall x, y \in [d], \mathcal{Z} \subseteq [d]$,

$$(x \perp\!\!\!\perp_{\boldsymbol{A}} y \mid \mathcal{Z}) \implies (x \perp\!\!\!\perp_{\mathcal{D}} y \mid \mathcal{Z}). \tag{1}$$

The collection of all DAGs satisfying this criterion constitutes the Markov Equivalence Class (MEC). Our method, as most methods for learning DAGs, relies on the Causal Faithfulness Assumption, which

---

[1] Assuming infinite samples in $\mathcal{D}$, $x \perp\!\!\!\perp_{\mathcal{D}} y \mid z$ means $P(x \mid y, z) = P(x \mid z)$ in the true data distribution.

posits the unique existence of a DAG $\boldsymbol{A}^*$ such that the observed data $\mathcal{D}$ exhibits only the conditional independencies represented by the d-separations in $\boldsymbol{A}$. This assumption is equivalent to the converse direction of Equation (1). However, challenges emerge when dealing with large conditioning sets $\mathcal{Z}$, as statistical tests for conditional independence become increasingly unreliable. To mitigate this issue, recent approaches like LOCI [42] and $k$-PC [18] restrict their search to DAGs with low-order conditioning sets (typically limited to a cardinality of 2 or less). As a result, their solution space is not the entire MEC, but rather the $k$-equivalence class (or $k$-essential graphs) [18], which comprises DAGs whose low-order d-separation statements are reflected as low-order conditional independencies in $\mathcal{D}$. In a similar vein, our approach focuses on conditioning sets with a cardinality of 1 or less ($|\mathcal{Z}| \leq 1$), allowing for more reliable and efficient causal discovery.

**Continuous DAG Acyclicity Constraint.** Pioneered by NOTEARS [36], recent score-based methods provide an interesting alternative in DAG search by reformulating it as a continuous optimization problem over weighted adjacency matrices. This transformation relies on the development of DAG acyclicity regularizers, which quantify the degree of cyclicity in weighted adjacency matrices. Our work follows this practice and uses the log-determinant DAG regularizer introduced by DAGMA [4], defined as:

$$\mathcal{L}_{\text{DAG}}(\boldsymbol{W}, s) = -\log \det(s\boldsymbol{I} - \boldsymbol{W}) + d \log s,$$

where $s$ is a hyperparameter that controls the size of the valid domain of this function, denoted as $\mathbb{W}^s = \{\boldsymbol{W} \in \mathbb{R}^{d \times d} \mid s \geq \rho(\boldsymbol{W})\}$, with $\rho(\boldsymbol{W})$ representing the spectral radius of $\boldsymbol{W}$. As established by Bello et al. [4], for $\boldsymbol{W} \in \mathbb{W}^s$, the condition $\mathcal{L}_{\text{DAG}}(\boldsymbol{W}, s) = 0$ implies that $\boldsymbol{W}$ maps to a DAG.

## 3 Differentiable d-Separation Framework

Our core contribution lies in the transformation of $d$-separation, which is discrete and binary computed via combinatorial algorithms [8, 19, 32], into a continuously differentiable function over $\boldsymbol{W} \in [0, 1]^{d \times d}$. This transformation satisfies three central properties: (1) it exactly recovers conventional d-separation statements when evaluated on discrete DAGs $\mathbf{A} \in \{0, 1\}^{d \times d}$, (2) it provides a principled probabilistic interpretation when $\boldsymbol{W}$ is viewed as a parameterization of a distribution over graphs, and (3) it is a differentiable function admitting gradient-based optimization on $\boldsymbol{W}$.

The key insight is grounding the score in a percolation-based graph connectivity measure. Specifically, we quantify how well a probabilistic graph structure (parameterized by $\boldsymbol{W}$) aligns with conditional independence (CI) patterns observed in data, such as those measured through statistical tests yielding p-values, which pave the way to a comprehensive set of differentiable objective functions that measure violations of low-order I-Mapness, Faithfulness, and Acyclicity constraints.

We achieve this differentiable framework through three key steps: (1) recasting $d$-separation as first-order logic (FOL) formulae operating on graph reachability (Section 3.1), (2) applying soft logic operators to transform these discrete statements into continuously differentiable functions with valid probabilistic interpretations (Section 3.2), and (3) combining these soft d-separation measures from the graph with conditional independence measures from data to construct objective functions that enforce low-order I-Mapness and Faithfulness constraints while maintaining acyclicity (Section 4).

### 3.1 FOL Formulation of d-Separation

Conventionally, d-separations are checked by discrete, combinatorial algorithms using data structures such as hash sets and double-ended queues [8, 19, 32]. The inherently sequential and combinatorial nature of these algorithms presents significant challenges in developing continuous relaxations and enabling gradient-based optimization. To address this limitation, we introduce a First-Order Logic (FOL) framework that provides an equivalent characterization of d-separation computations in discrete settings while maintaining differentiability properties. This reformulation bridges discrete reasoning and continuous optimization paradigms, as it allows us to leverage established techniques from probabilistic soft logic and fuzzy logic systems for end-to-end differentiation of the computation.

Specifically, we observe that the conventional notion of (low-order) d-separation and d-connection on any given discrete DAG $\boldsymbol{A} \in \{0, 1\}^{d \times d}$ can be precisely captured by FOL formulæ based *solely* on graph reachability information. That is, given the graph reachability logical predicate

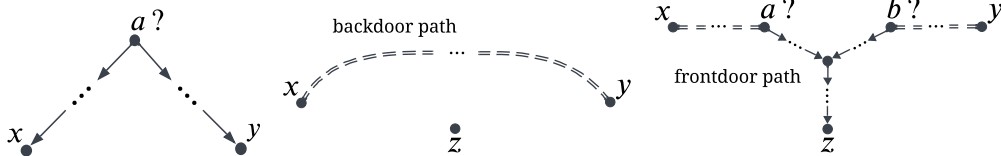

(a) 0th-order d-connection pattern: checking existence of common ancestor

(b) 1st-order d-connection pattern A: checking existence of backdoor path that does not involve $z$

(c) 1st-order d-connection pattern B: checking existence of frontdoor path that involves $z$

Figure 1: Illustration of the graph connectivity patterns checked by the low-order d-separation/d-connection FOL formulæ (Definition 3.1) to determine whether the given query $((x, y)$ or $(x, y \mid z))$ is d-separated or d-connected. The double-dashed line $== \cdots ==$ refers to a 0th-order d-connecting path, and the arrowed line $\rightarrow \cdots \rightarrow$ refers to a directed path. In either pattern, the variable nodes $a$ and $b$ could be the query node $x$ or $y$ themselves.

$R_{\boldsymbol{A}} : [d]^2 \rightarrow \{0, 1\}$ defined as

$$R_{\boldsymbol{A}}(x, y) = 1 \text{ if and only if } x \rightsquigarrow y \text{ in } \boldsymbol{A}.$$

We can define the FOL formulæ expressing the 0th-order (i.e. unconditional) and 1st-order (i.e. conditioning on one variable) d-separation and d-connection statements as follows:

**Definition 3.1.** (Low-order d-separation and d-connection FOL formulæ) Given a discrete directed graph $\boldsymbol{A} \in \{0, 1\}^{d \times d}$, the 0th-order d-separation formula $S_{\boldsymbol{A}}^{(0)} : [d]^2 \rightarrow \{0, 1\}$ and the 1st-order d-separation formula $S_{\boldsymbol{A}}^{(1)} : [d]^3 \rightarrow \{0, 1\}$ are defined as:

$$S_{\boldsymbol{A}}^{(0)}(x, y) := \forall a \in [d], \neg \left( R_{\boldsymbol{A}}(a, x) \wedge R_{\boldsymbol{A}}(a, y) \right), \tag{2}$$

$$S_{\boldsymbol{A}}^{(1)}(x, y \mid z) := S_{\boldsymbol{A}_{-z}}^{(0)}(x, y) \wedge \left( \left( \forall a \in [d] \setminus \{z\}, S_{\boldsymbol{A}_{-z}}^{(0)}(x, a) \vee \neg R_{\boldsymbol{A}}(a, z) \right) \right.$$
$$\left. \vee \left( \forall b \in [d] \setminus \{z\}, S_{\boldsymbol{A}_{-z}}^{(0)}(y, b) \vee \neg R_{\boldsymbol{A}}(b, z) \right) \right). \tag{3}$$

Equivalently, the 0-th and 1-st order d-connection statements $C_{\boldsymbol{A}}^{(0)}$ and $C_{\boldsymbol{A}}^{(1)}$ are:

$$C_{\boldsymbol{A}}^{(0)}(x, y) := \exists a \in [d], R_{\boldsymbol{A}}(a, x) \wedge R_{\boldsymbol{A}}(a, y), \tag{4}$$

$$C_{\boldsymbol{A}}^{(1)}(x, y \mid z) := C_{\boldsymbol{A}_{-z}}^{(0)}(x, y) \vee \left( \left( \exists a \in [d] \setminus \{z\}, C_{\boldsymbol{A}_{-z}}^{(0)}(x, a) \wedge R_{\boldsymbol{A}}(a, z) \right) \right.$$
$$\left. \wedge \left( \exists b \in [d] \setminus \{z\}, C_{\boldsymbol{A}_{-z}}^{(0)}(y, b) \wedge R_{\boldsymbol{A}}(b, z) \right) \right). \tag{5}$$

To understand why these formulæ correctly capture d-separation and d-connection when $\boldsymbol{A}$ is a DAG, consider first the 0th-order formulae $S_{\boldsymbol{A}}^{(0)}(x, y)$ and $C_{\boldsymbol{A}}^{(0)}(x, y)$. These formulæ essentially determine whether any node (including possibly $x$ or $y$ themselves) serves as a common ancestor to both $x$ and $y$ (e.g. the node $a$? in Figure 1a). If such a common ancestor exists, it creates either a chain structure (when $a = x$ or $a = y$, meaning $x$ is the ancestor of $y$ or vice versa) or a fork structure (when $a \neq x$ and $a \neq y$, and a third node is the common ancestor). In either case, $x$ and $y$ are unconditionally d-connected, yielding $C_{\boldsymbol{A}}^{(0)}(x, y) = 1$ as expressed in Equation (4).

The 1st-order formulæ $S_{\boldsymbol{A}}^{(1)}(x, y \mid z)$ and $C_{\boldsymbol{A}}^{(1)}(x, y \mid z)$ involve more intricate logic but follow an intuitive two-step process. Take $C_{\boldsymbol{A}}^{(1)}(x, y \mid z)$ for instance. It first examines, via the first term $C_{\boldsymbol{A}_{-z}}^{(0)}(x, y)$, whether $x$ and $y$ remain d-connected in the subgraph $\boldsymbol{A}_{-z}$ where conditioning node $z$ is removed — effectively checking for any *backdoor* path that bypasses $z$ (Figure 1b). If no such backdoor path exists, the formulæ then evaluate whether there exists a valid *frontdoor* path through $z$. For this frontdoor path to d-connect $x$ and $y$ when conditioning on $z$, $z$ must be a collider or a descendant of a collider. This translates to checking whether there is any 0th-order d-connecting path that does not involve $z$ and that connects $x$ to *any ancestor of $z$ or $z$ itself,* and similarly whether there is such a path connecting $y$ to any ancestor of $z$ or $z$ itself. As illustrated in Figure 1c, this is checking

if the 0th-order d-connecting path $x == \cdots == a?$ and the directed path $a? \to \cdots \to z$ both exist at the same time. The following theorem formalizes the correctness of these logical characterizations.

**Theorem 3.2.** *For any DAG $\boldsymbol{A}$ with $d$ nodes and any three nodes $x, y, z \in [d]$ that are distinct, $x \perp\!\!\!\perp_{\boldsymbol{A}} y$ if and only if $S_{\boldsymbol{A}}^{(0)}(x, y) = 1$, and $x \perp\!\!\!\perp_{\boldsymbol{A}} y \mid z$ if and only if $S_{\boldsymbol{A}}^{(1)}(x, y \mid z) = 1$. Similarly, $x \not\perp\!\!\!\perp_{\boldsymbol{A}} y$ if and only if $C_{\boldsymbol{A}}^{(0)}(x, y) = 1$, and $x \not\perp\!\!\!\perp_{\boldsymbol{A}} y \mid z$ if and only if $C_{\boldsymbol{A}}^{(1)}(x, y \mid z) = 1$.* [2]

The d-separation and d-connection formulæ presented in Theorem 3.2 rely fundamentally on the graph reachability predicate $R_{\boldsymbol{A}}$. Crucially, we observe that $R_{\boldsymbol{A}}$ itself can be expressed through FOL formulæ that operate directly on the adjacency matrix $\boldsymbol{A}$, by taking a recursive form analogous to the generalized Bellman-Ford algorithm [3].

**Definition 3.3.** (Graph Reachability FOL formulæ) Given a discrete directed graph $\boldsymbol{A} \in \{0, 1\}^{d \times d}$, we define the reachability formulæ $R_{\boldsymbol{A}}^{(l)}(x, y) : [d]^2 \to \{0, 1\}$ for path lengths no greater than $l$ recursively as follows:

$$R_{\boldsymbol{A}}^{(0)}(x, y) = \mathbb{1}(x = y), \tag{6}$$

$$R_{\boldsymbol{A}}^{(l)}(x, y) = \left( \bigvee_{u \in [d]} \left( R_{\boldsymbol{A}}^{(l-1)}(x, u) \wedge \boldsymbol{A}_{u,y} \right) \right) \vee R_{\boldsymbol{A}}^{(l-1)}(x, y). \tag{7}$$

In practice, we take $R_{\boldsymbol{A}}(x, y) := R_{\boldsymbol{A}}^{(d-1)}(x, y)$ where $d$ is the number of nodes, since the longest directed path in a DAG with $d$ nodes has length $d - 1$. The correctness of these formulæ is articulated in the following lemma:

**Lemma 3.4.** *For any discrete graph $\boldsymbol{A} \in \{0, 1\}^{d \times d}$ with maximum directed path length $l$ and for all pair of nodes $x, y \in [d]$, $x \rightsquigarrow_{\boldsymbol{A}} y$ if and only if $R_{\boldsymbol{A}}^{(l)}(x, y) = 1$.*

By integrating the formulæ in Definition 3.1 with those in Definition 3.3, we establish a complete framework that can systematically determine d-separation and d-connection relationships in DAGs using purely FOL operations, laying the foundation for the differentiable relaxation that follows.

## 3.2 Continuous Relaxation via Soft Logic

Having established d-separation and d-connection as FOL formulæ, we can now leverage the rich literature on probabilistic soft logic (PSL) and fuzzy logic to transform these discrete statements into continuously differentiable functions. These soft logic frameworks systematically relax Boolean operations (conjunction, disjunction, existential and universal quantification) into continuous functions that operate over the interval [0,1] via various t-norms and t-conorms—continuous analogs of logical AND and OR operations [2, 5, 17, 38]. In doing so, the logical semantics necessary to defined reachability (a percolation metric) are preserved while enabling gradient-based optimization and sampling.

In this work, we adopt the LogLTN framework proposed by Badreddine et al. [2], which implements the product t-norm (for logical AND) and max t-conorm (for logical OR) in the logarithmic space. Given a set of $m$ values $\{x_i\}_{i \in [m]}$, each $x_i \in [0, 1]$, and their logarithmic representations $\{x_i'\}_{i \in [m]}$, each $x_i' = \log(x_i)$, the standard product t-norm and max t-conorm $T_m, O_m : [0, 1]^m \to [0, 1]$, and LogLTN's logarithmic versions $\tilde{T}_m, \tilde{O}_m : \mathbb{R}^m \to \mathbb{R}_-$, where $\mathbb{R}_- := \{x \in \mathbb{R} \mid x \leq 0\}$, are:

$$T_m(\{x_i\}_{i \in [m]}) := \prod_{i=1}^{m} x_i \ , \quad O_m(\{x_i\}_{i \in [m]}) := \max\{x_i\}_{i \in [m]} \ ,$$

$$\tilde{T}_m(\{x_i'\}_{i \in [m]}) := \sum_{i=1}^{m} x_i' \ , \quad \tilde{O}_m(\{x_i'\}_{i \in [m]}) := \alpha \left( C + \log \left( \frac{\sum_{i=1}^{n} e^{x_i'/\alpha - C}}{m} \right) \right) \ . \tag{8}$$

where $C = \max(x_1'/\alpha, \ldots, x_m'/\alpha)$ and $\alpha \in (0, 1]$ is the temperature controlling the approximation accuracy (the closer $\alpha$ is to 0 the more accurate). Notably, the logarithmic t-norm $\tilde{T}_m$ exactly

---

[2]The proofs of Theorem 3.2 and all following theorems and lemmas can be found in Appendix A.

represents its standard counterpart, i.e., $\tilde{T}_m(\cdot) = \log(T_m(\cdot))$, whereas the logarithmic t-conorm $\tilde{O}_m$ implements the Log-Mean-Exponential operation that provides a lower bound approximation of $\log(O_m(\cdot))$ with bounded error [2]:

$$\log(O_m(\mathbf{x})) - \alpha \log(m) \leq \tilde{O}_m(\mathbf{x}') \leq \log(O_m(\mathbf{x})).$$

Our choice of the LogLTN operators is motivated by two key considerations. First, a comparative analysis by van Krieken et al. [38] shows that the product t-norm provides the most stable and informative gradients for differentiable learning, while Badreddine et al. [2] shows that the Log-MeanSum approximation of the max t-conorm in the logarithmic space further enhances gradient stability. Second, as we will demonstrate next, this specific combination of max-product operators consistently yields principled lower bounds on the quantities being estimated and allows us to derive continuously differentiable functions, which we name the *differentiable d-separation/d-connection*, that compute lower bounds on *expected d-separation/d-connection statements* over a distribution of DAGs parameters by the weighted adjacency matrix $\boldsymbol{W}$. Specifically, we now transform the FOL formulæ in Definition 3.1 and Definition 3.3 into their continuous counterparts that operate on the weighted adjacency matrices $\boldsymbol{W} \in [0,1]^{d \times d}$, using the LogLTN logical operators.

**Definition 3.5.** (Differentiable Graph Reachability and Unreachability Percolations) Given a weighted adjacency matrix $\boldsymbol{W} \in [0,1]^{d \times d}$, we define the relaxed reachability and unreachability functions, $\tilde{R}_{\boldsymbol{W}}^{(l)}(x,y), \tilde{U}_{\boldsymbol{W}}^{(l)}(x,y) : [d]^2 \to \mathbb{R}_-$, for path lengths no greater than $l$ recursively as follows:

$$\tilde{R}_{\boldsymbol{W}}^{(0)}(x,y) := \log(\mathbb{1}(x=y)) \quad \tilde{R}_{\boldsymbol{W}}^{(l)}(x,y) := \tilde{O}_{d+1}\Big(\{\tilde{T}_2(\tilde{R}_{\boldsymbol{W}}^{(l-1)}(x,u), \log(W_{uy}))\}_{u \in [d]} \cup \{\tilde{R}_{\boldsymbol{W}}^{(l-1)}(x,y)\}\Big),$$

$$\tilde{U}_{\boldsymbol{W}}^{(0)}(x,y) := \log(\mathbb{1}(x \neq y)) \quad \tilde{U}_{\boldsymbol{W}}^{(l)}(x,y) := \tilde{T}_{d+1}\Big(\{\tilde{O}_2(\tilde{U}_{\boldsymbol{W}}^{(l-1)}(x,u), \log(1-W_{uy}))\}_{u \in [d]} \cup \{\tilde{U}_{\boldsymbol{W}}^{(l-1)}(x,y)\}\Big),$$

where we take $\log(0) = -\infty$.

**Definition 3.6.** (Differentiable d-Separation and d-Connection) Given a weighted adjacency matrix $\boldsymbol{W} \in [0,1]^{d \times d}$, we define the 0th-order differentiable d-separation and d-connection function $\tilde{S}_{\boldsymbol{W}}^{(0)}(x,y), \tilde{C}_{\boldsymbol{W}}^{(0)}(x,y) : [d]^2 \to \mathbb{R}_-$ and the 1st-order differentiable d-separation and d-connection function $\tilde{S}_{\boldsymbol{W}}^{(1)}(x,y|z), \tilde{C}_{\boldsymbol{W}}^{(1)}(x,y|z) : [d]^3 \to \mathbb{R}_-$ as follows:

$$\tilde{S}_{\boldsymbol{W}}^{(0)}(x,y) := \tilde{T}_d\left(\left\{\tilde{O}_2\left(\tilde{U}_{\boldsymbol{W}}^{(d)}(a,x), \tilde{U}_{\boldsymbol{W}}^{(d)}(a,y)\right)\right\}_{a \in [d]}\right), \quad \tilde{C}_{\boldsymbol{W}}^{(0)}(x,y) := \tilde{O}_d\left(\left\{\tilde{T}_2\left(\tilde{R}_{\boldsymbol{W}}^{(d)}(a,x), \tilde{R}_{\boldsymbol{W}}^{(d)}(a,y)\right)\right\}_{a \in [d]}\right),$$

$$\tilde{S}_{\boldsymbol{W}}^{(1)}(x,y|z) := \tilde{T}_2\bigg(\tilde{S}_{\boldsymbol{W}_{-z}}^{(0)}(x,y), \tilde{O}_2\bigg(\tilde{T}_{d-1}\left(\left\{\tilde{O}_2\left(\tilde{S}_{\boldsymbol{W}_{-z}}^{(0)}(x,a), \tilde{U}_{\boldsymbol{W}}^{(d)}(a,z)\right)\right\}_{a \in [d]\setminus\{z\}}\right), \tilde{T}_{d-1}\left(\left\{\tilde{O}_2\left(\tilde{S}_{\boldsymbol{W}_{-z}}^{(0)}(y,b), \tilde{U}_{\boldsymbol{W}}^{(d)}(b,z)\right)\right\}_{b \in [d]\setminus\{z\}}\right)\bigg)\bigg),$$

$$\tilde{C}_{\boldsymbol{W}}^{(1)}(x,y|z) := \tilde{O}_2\bigg(\tilde{C}_{\boldsymbol{W}_{-z}}^{(0)}(x,y), \tilde{T}_2\bigg(\tilde{O}_{d-1}\left(\left\{\tilde{T}_2\left(\tilde{C}_{\boldsymbol{W}_{-z}}^{(0)}(x,a), \tilde{R}_{\boldsymbol{W}}^{(d)}(a,z)\right)\right\}_{a \in [d]\setminus\{z\}}\right), \tilde{O}_{d-1}\left(\left\{\tilde{T}_2\left(\tilde{C}_{\boldsymbol{W}_{-z}}^{(0)}(y,b), \tilde{R}_{\boldsymbol{W}}^{(d)}(b,z)\right)\right\}_{b \in [d]\setminus\{z\}}\right)\bigg)\bigg).$$

Notably, for the d-separation functions relying on graph *un*reachability, we directly derive unreachability from the negation of reachability FOL formulæ rather than computing $\log(1-\exp(\tilde{R}_{\boldsymbol{W}}^{(l)}(x,y)))$. This is essential for preserving the correct probabilistic interpretation, as we now demonstrate.

Consider $\boldsymbol{W}$ as representing a parametric distribution over discrete graphs $\boldsymbol{A}$. Namely, to sample $\boldsymbol{A}$, we sample each edge $x \to y$ independently via $\boldsymbol{A}_{xy} \sim \text{Bern}(\boldsymbol{W}_{xy})$ (i.e. Bernoulli distribution). We denote the entire graph's sampling as $\boldsymbol{A} \sim \text{Bern}(\boldsymbol{W})$. We first notice that the differentiable graph reachability and graph unreachability functions on $\boldsymbol{W}$ given in Definition 3.5 always yield a *lower bound* to the expected reachability and unreachability on $\boldsymbol{A}$ respectively, when $\boldsymbol{A} \sim \text{Bern}(\boldsymbol{W})$. Intuitively, this is because the max operator provides a lower bound on the probability of a union of events (logical OR), while the product operator correctly computes the probability of an intersection of events (logical AND) when said events are mutually independent or provides a lower bound when they are positively correlated, which is precisely the case for overlapping paths in graphs needed in reachability (a percolation metric). As these operators compose recursively in our reachability computation, they preserve the lower bound properties, leading to the following lemma.

**Lemma 3.7** (Reachability Percolation Lower Bound). *Given a weighted adjacency matrix $\boldsymbol{W} \in [0,1]^{d \times d}$, for any $0 \leq l < d$, and for any pair of nodes $x, y \in [d]$, we have*

$$\tilde{R}_{\boldsymbol{W}}^{(l)}(x,y) \leq \log \mathbb{E}_{\boldsymbol{A} \sim \text{Bern}(\boldsymbol{W})}\left[R_{\boldsymbol{A}}^{(l)}(x,y)\right] \quad \text{and} \quad \tilde{U}_{\boldsymbol{W}}^{(l)}(x,y) \leq \log \mathbb{E}_{\boldsymbol{A} \sim \text{Bern}(\boldsymbol{W})}\left[U_{\boldsymbol{A}}^{(l)}(x,y)\right].$$

The lower bound property of both our reachability and unreachability functions is critical—note that naively computing unreachability as $\log(1 - \exp(\tilde{R}_{\mathbf{W}}^{(l)}(x,y)))$ would yield an upper bound instead. This consistency in providing lower bounds is essential for extending our theoretical guarantees to the differentiable d-separation and d-connection functions. As the following lemma demonstrates, these functions also preserve the lower bound property relative to their expected values:

**Theorem 3.8** (Lower Bound on Expected $d$-Separation Statements). *Given a weighted adjacency matrix $\mathbf{W} \in [0,1]^{d \times d}$, for any three nodes $x, y, z \in [d]$,*

$$\tilde{S}_{\mathbf{W}}^{(0)}(x,y) \leq \log \mathbb{E}_{\mathbf{A} \sim Bern(\mathbf{W})} \left[ S_{\mathbf{A}}^{(0)}(x,y) \right], \quad \tilde{S}_{\mathbf{W}}^{(1)}(x,y \mid z) \leq \log \mathbb{E}_{\mathbf{A} \sim Bern(\mathbf{W})} \left[ S_{\mathbf{A}}^{(1)}(x,y \mid z) \right],$$

$$\tilde{C}_{\mathbf{W}}^{(0)}(x,y) \leq \log \mathbb{E}_{\mathbf{A} \sim Bern(\mathbf{W})} \left[ C_{\mathbf{A}}^{(0)}(x,y) \right], \quad \tilde{C}_{\mathbf{W}}^{(1)}(x,y \mid z) \leq \log \mathbb{E}_{\mathbf{A} \sim Bern(\mathbf{W})} \left[ C_{\mathbf{A}}^{(1)}(x,y \mid z) \right].$$

These lemmas establish that our differentiable functions provide lower bounds on the logarithm of expected d-connection and d-separation statements when viewing $\mathbf{W}$ as parameterizing a distribution over discrete graphs. This mathematical guarantee opens a principled pathway for gradient-based optimization by maximizing these lower bounds.

# 4 DAGPA (DAG Percolation Apartness)

We now introduce DAGPA, a practical instantiation to demonstrate the validity of our differentiable $d$-separation scores (Section 3.2). This implementation employs certain heuristics and simplifications to facilitate optimization. The primary goal is not to claim DAGPA's optimality, but rather to provide empirical validation of the framework's potential for causal discovery and open avenues for future research exploring more sophisticated realizations of the core principles.

In DAGPA, we parameterize the weighted adjacency matrix $\mathbf{W}$ with parameters $\theta \in \mathbb{R}^{d \times d}$ via the sigmoid transformation $\mathbf{W} = \sigma(\theta)$. We develop objective functions that encourage alignment between the d-separation/d-connection statements implied by the graph structure $\mathbf{W}$ and conditional independence patterns in data. Rather than using binary decisions based on thresholded p-values, DAGPA directly incorporates raw p-values from statistical independence tests as soft CI labels from data. This approach makes the model naturally robust to uncertainty in CI testing, as gradient-based optimization can prioritize clear cases (p-values near 0 indicating dependence or near 1 indicating independence) over borderline ones.

Our objective functions consists of three components: Acyclicity, "True Positive," and "True Negative" losses[3]. A true positive occurs when the model predicts a high probability of d-separation for variable pairs showing high independence p-values in the data. Conversely, a true negative occurs when the model predicts a high probability of d-connection for pairs showing low p-values (strong dependence).

**Definition 4.1** (Multi-task Constraint-based CI Statement Losses of DAGPA). Given a dataset $\mathcal{D}$ with $n$ samples and $d$ variables, and the p-values $p_{\mathcal{D}}(\cdot)$ of some statistical conditional independence tests on all the 0th-order and 1st-order statements. Let $\mathbb{I}_0 = \{(x,y) \mid x, y \in [d], x > y\}$ and $\mathbb{I}_1 = \{(x,y,z) \mid x, y, z \in [d], x > y, x \neq z, y \neq z\}$. The CI Statement "true positive (TP)" losses $\mathcal{L}_{\text{TP-0}}$ and $\mathcal{L}_{\text{TP-1}}$, the "true negative (TN)" losses $\mathcal{L}_{\text{TN-0}}$ and $\mathcal{L}_{\text{TN-1}}$, as well as the log-determinant DAG acyclicity loss $\mathcal{L}_{\text{DAG}}$ from DAGMA [4], defined on the model parameters $\theta \in \mathbb{R}^{d \times d}$, are as follows:

$$\mathcal{L}_{\text{TP-0}}(\theta, \mathcal{D}) = -\sum_{(x,y) \in \mathbb{I}_0} \tilde{S}_{\sigma(\theta)}^{(0)}(x,y) p_{\mathcal{D}}(x,y), \qquad \mathcal{L}_{\text{TP-1}}(\theta, \mathcal{D}) = -\sum_{(x,y,z) \in \mathbb{I}_1} \tilde{S}_{\sigma(\theta)}^{(1)}(x,y \mid z) p_{\mathcal{D}}(x,y \mid z),$$

$$\mathcal{L}_{\text{TN-0}}(\theta, \mathcal{D}) = -\sum_{(x,y) \in \mathbb{I}_0} \tilde{C}_{\sigma(\theta)}^{(0)}(x,y)(M_0 - p_{\mathcal{D}}(x,y)), \quad \mathcal{L}_{\text{TN-1}}(\theta, \mathcal{D}) = -\sum_{(x,y,z) \in \mathbb{I}_1} \tilde{C}_{\sigma(\theta)}^{(1)}(x,y \mid z)(M_1 - p_{\mathcal{D}}(x,y \mid z)),$$

$$\mathcal{L}_{\text{DAG}}(\theta, s) = -\log \det(s\mathbf{I} - \sigma(\theta)) + d \log s.$$

where $M_0 = \max_{\mathbb{I}_0} p_{\mathcal{D}}(x,y)$, $M_1 = \max_{\mathbb{I}_1} p_{\mathcal{D}}(x,y \mid z)$, $\sigma$ is the sigmoid function, and $s$ is the hyperparameter controlling the domain of the log-determinant DAG loss of DAGMA [4].

We note that, assuming the p-values accurately reflect the true CI patterns, minimizing the "TP" and "TN" losses maximizes $\tilde{S}$ and $\tilde{C}$ on the correct queries. Furthermore, since $\tilde{S}$ and $\tilde{C}$ are lower bounds of the expected d-separation and d-connection statements over $\text{Bern}(\mathbf{W})$ (Theorem 3.8), we are maximizing lower bounds - the correct optimization direction. The following lemmas show the consistency of the losses and the validity of taking sum aggregation over individual CI queries.

---

[3] We use these terms loosely, as p-values in practice may not perfectly reflect true CI statements.

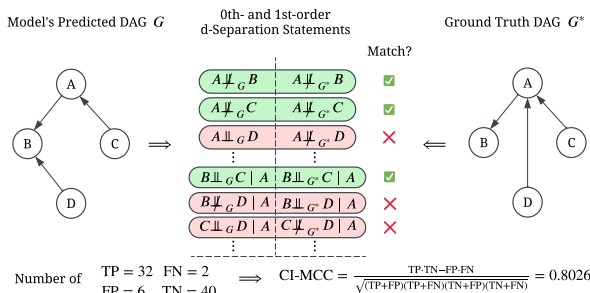

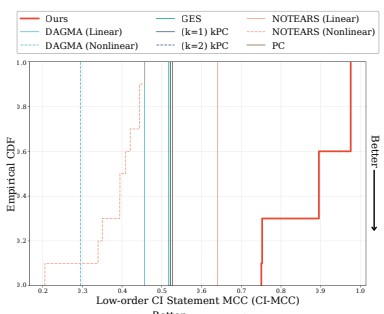

Figure 2: The Low-order CI Statement Matthews Correlation Coefficient (CI-MCC) metric measures alignment between the low-order d-separation statements predicted by the causal discovery model, and those implied from the ground-truth DAG (symmetric statements are considered).

Figure 3: The empirical CDF of CI-MCC on Sachs [28]. The closer the curves to the bottom-right ($\searrow$), the better the performance. **DAGPA achieves the best performance compared to all baseline methods.**

**Lemma 4.2** (Consistency of Multi-Task CI Losses). *Given a faithful data $\mathcal{D}$, as the sample size $n \to \infty$ and the LogMeanExp temperature $\alpha \to 0^+$ (Equation (8)), the optimal DAG $\boldsymbol{A}^*$ achieves the minimum value on each of $\mathcal{L}_{TP\text{-}0}, \mathcal{L}_{TP\text{-}1}, \mathcal{L}_{TN\text{-}0}, \mathcal{L}_{TN\text{-}1},$ and $\mathcal{L}_{DAG}$.*

**Lemma 4.3** (Mutual Independence of Low-order CI Statements). *All 0th- and 1st-order CI statements over the variable sets $\mathbb{I}_0 = \{(x,y) \mid x,y \in [d], x > y\}$ and $\mathbb{I}_1 = \{(x,y,z) \mid x,y,z \in [d], x > y, x \neq z, y \neq z\}$ are mutually independent. In other words, none of these CI statements can be implied from any other of these CI statements via the graphoid axioms [24].*

In Appendix C, we provide the complete algorithm for DAGPA along with detailed descriptions of additional heuristics employed to address optimization challenges, including: PCGrad [43] for resolving conflicting gradients in multi-task learning, Discrete Langevin Proposal (DLP) [50] for efficient exploration of the weight space, and an optimization-time DAG selection score based on CI statement measures from the data.

## 5 Experiments

This section presents an empirical evaluation of DAGPA's ability to discover DAGs whose d-separation statements are consistent with those derived from the underlying causal structure that generated the data. Our objective is not to test superiority over baselines in terms of exact structural recovery (that is left to Appendix F), but rather to demonstrate that DAGPA successfully achieves its intended purpose: learning DAGs that entail the same low-order conditional independence statements as the ground truth. The experimental results indicate that our differentiable d-separation framework offers a promising direction for causal discovery.

**Evaluation.** To evaluate causal discovery from a CI statement alignment perspective, we propose a metric that aligns with this objective: the **Low-order CI Statement Matthews Correlation Coefficient (CI-MCC)**, illustrated in Figure 2. Given a model's predicted causal graph (which may be a DAG, PDAG, CPDAG, or k-essential graph), we derive the set of all 0th- and 1st-order d-separation statements implied by the structure. We then compare these against the corresponding d-separation statements derived from the ground-truth DAG that generated the data. Treating this as a binary classification task, we compute the Matthews Correlation Coefficient, which ranges from $[-1, 1]$ with higher values indicating better alignment between model-predicted and ground-truth CI statements. We select MCC over alternatives like F1 score due to its robustness to class imbalance—a critical consideration since the true ratio of independence to dependence relationships varies widely across different causal structures and is typically unknown in real-world settings. For completeness, results using conventional causal discovery metrics, such as graph structure F1 and Structural Hamming Distance (SHD) scores, are provided in Appendix F.

**Baselines.** We compare DAGPA with representative causal discovery methods spanning different paradigms: constraint-based approaches including PC [34] and k-PC [18]; score-based methods such

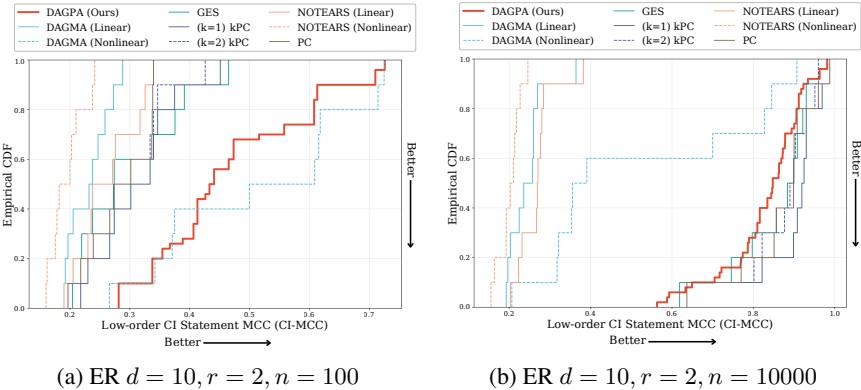

(a) ER $d = 10, r = 2, n = 100$        (b) ER $d = 10, r = 2, n = 10000$

Figure 4: The empirical cumulative distribution functions (CDFs) of the low-order CI Statement Matthews Correlation Coefficient (CI-MCC) on two synthetic binary settings. The closer the curves to the bottom-right ($\searrow$), the better the performance. **Our method achieves on-par performance to the constraint-based baselines in high-sample regime (Figure 4b) while demonstrating much better robustness in low-sample regime (Figure 4a).**

as GES [6]; and continuous optimization approaches including NOTEARS (linear and nonlinear) [36] and DAGMA (linear and nonlinear) [4]. This selection provides a comprehensive comparison landscape across methodological families. Detailed baseline configurations are provided in Appendix D.5.

**Experimental Settings.** We evaluate methods on both synthetic and real-world datasets. For synthetic data, we generate binary data requiring no normalization. We follow a protocol similar to Kocaoglu [18], generating binary datasets from randomly constructed DAGs. We use both Erdős–Rényi (ER) and Scale-Free (SF) graph models via the pyAgrum package [10], with varying complexity parameters: nodes $d \in \{10, 50\}$, edge-to-node ratio $r \in \{2, 4\}$, and sample sizes $n \in \{100, 1000, 10000, 100000\}$. For each configuration, we generate 10 datasets. All constraint-based methods employ Fisher-z tests [11] (with potential of other CI tests as alternative; details in appendix), for computing conditional independence p-values, ensuring a fair comparison. For real-world validation, we use the Sachs dataset [28], a benchmark protein signaling network with 11 variables and a known ground-truth causal structure derived from experimental interventions. This dataset represents a challenging real-world case where the underlying causal mechanisms are likely nonlinear and complex. We employ the standard discretized version preprocessed via the Hartemink discretization method, which converts continuous protein concentrations into 3-level categorical variables (low, average, high) while preserving dependence structure.

We note that our CI-constraint-based approach offers inherent robustness to preprocessing artifacts that can affect score-based methods: standard CI tests (e.g. Chi-squared for discrete data, Fisher-Z for continuous data) are invariant to scaling transformations, whereas likelihood-based objectives can exploit variance differences as spurious causal signals [27]. This makes our framework particularly robust across different preprocessing pipelines.

**Results on Sachs.** To evaluate performance on real-world data with potentially complex underlying mechanisms, we test DAGPA on the Sachs dataset [28], a benchmark protein signaling network with 11 variables and 853 samples. Figure 3 displays the empirical CDF of CI-MCC scores across methods. DAGPA demonstrates dramatically superior performance, achieving CI-MCC scores of 0.75-0.98, while all baseline methods cluster between 0.3-0.65. For our method, we select the 10 DAGs with highest DAG selection scores to reflect the uncertainty inherent in causal discovery from real data. For baselines that depend on random seeds, we run each method with 10 different initializations to ensure fair comparison. The substantial performance gap highlights DAGPA's ability to effectively discover causal structures in real-world data where the underlying data-generating process is complex and potentially nonlinear. This confirms that our differentiable d-separation framework, combined with gradient-informed discrete sampling, provides robust causal discovery capabilities that generalize well beyond synthetic settings to real-world applications where ground truth is available but unknown during learning.

**Results on synthetic binary data.** Figure 4 presents the empirical cumulative distribution functions (CDFs) of CI-MCC scores across methods for different sample sizes. For each baseline, we include

one result per data instance in the batch (10 results total), while for DAGPA, we select the 5 discovered DAGs with the highest DAG selection score (Appendix C.3) per instance, yielding 50 total evaluations.

In low-sample settings (n=100, Figure 4a), DAGPA demonstrates superior performance compared to most baselines and performs on-par with DAGMA-Nonlinear [4], with CDF curves positioned toward the bottom-right indicating consistently higher CI-MCC scores. The advantage of DAGPA is particularly pronounced when compared to constraint-based methods (PC, k-PC), which struggle in low-sample regimes due to their reliance on hard conditional independence statements derived from uncertain p-values. Most score-based methods, including DAGMA-linear and both NOTEARS variants [36], also underperform in this setting, likely due to overfitting to limited data.

As sample size increases to n=10000 (Figure 4b), most methods improve as expected, with DAGPA maintaining competitive performance alongside constraint-based approaches. This pattern holds consistently across different graph structures and dimensions (see Appendix F), confirming that DAGPA effectively addresses RQ1 by exhibiting strong robustness in low-sample settings while maintaining competitive performance when data is abundant.

# 6   Related work

**Constraint-based Methods.** These methods discover causal structures through conditional independence (CI) testing. The PC algorithm [33, 34] produces CPDAGs representing Markov equivalence classes but struggles with small samples due to unreliable high-order CI tests. Recent advances like LOCI [42] and k-PC [18] demonstrate that low-order CI statements can sufficiently identify causal structures in many practical settings. Our work builds on these insights but uniquely transforms discrete CI decisions into a differentiable learning framework, further enhancing robustness in the small sample regime.

**Score-based Methods.** These methods optimize a score function over the DAG space. Traditional approaches like GES [6] use greedy search with information-theoretic criteria but face scalability challenges due to the discrete DAG space. NOTEARS [51] introduced a breakthrough with differentiable acyclicity characterization, enabling gradient-based optimization on weighted adjacency matrices and spurring extensions like nonlinear variants [52] and improved acyclicity constraints in DAGMA [4]. Our framework builds upon these continuous optimization advances but redirects the objective toward maximizing agreement with CI constraints rather than fitting functional models. This creates a hybrid approach combining strengths from both paradigms and opening a new avenue for causal discovery.

A more comprehensive discussion on related work can be found in Appendix E.

# 7   Conclusions

We introduced a novel causal discovery framework that bridges constraint-based and gradient-based continuous-optimization methods through differentiable d-separation. By proposing percolation-based differentiable d-separation scores, we enabled gradient-based learning of causal DAG structures from conditional independence patterns in finite datasets. Our model instantiation, DAGPA, demonstrates competitive performance across synthetic and real-world settings, with strong robustness in low-sample regimes where traditional methods often struggle, while maintaining competitive performance as data becomes abundant.

**Limitations and Future Work.**   Several limitations in our current approach present opportunities for future research. The reliance of DAGPA on p-values to measure both conditional dependence and independence from data can be improved, as using a constant minus the p-value to assess dependence strength is a simple heuristic. Fortunately, our framework is measurement-agnostic and supports alternatives, which we recommend future work to investigate. In addition, our method focuses on only low-order conditioning sets and assumes no confounding variables in the data. Future directions include extending the framework to handle latent confounders, efficiently incorporating higher-order CI statements, and simultaneously improving scalability. Finally, as suggested by a reviewer, a particularly promising avenue is integrating our differentiable d-separation framework with score-based methods like NOTEARS and DAGMA, enabling causal structure learning from both data likelihood signals and CI statement fitness signals (case studies in Appendix G.1). These advancements would further bridge constraint-based and score-based paradigms in causal discovery.

## Acknowledgment

The authors thank Ruqi Zhang and Patrick Pynadath for the insightful discussions and valuable feedback on Bayesian sampling methods. This work was funded in part by the Ford Motor Company under the Purdue-Ford Alliance, the National Science Foundation (NSF) awards CAREER IIS-1943364, CNS-2212160 and CAREER 2239375, IIS-2348717, Amazon Research Award, AnalytiXIN, Adobe Research, Intuit, the Wabash Heartland Innovation Network (WHIN), Walther Cancer Foundation and Purdue Institute for Cancer Research (PICR) (Grant P30CA023168). Computing infrastructure supported in part by AMD under the AMD HPC Fund program and PICR. Any opinions, findings and conclusions or recommendations expressed in this material are those of the authors and do not necessarily reflect the views of the sponsors.

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

# A Proofs

**Definition 3.1.** (Low-order d-separation and d-connection FOL formulæ) Given a discrete directed graph $A \in \{0, 1\}^{d \times d}$, the 0th-order d-separation formula $S_A^{(0)} : [d]^2 \to \{0, 1\}$ and the 1st-order d-separation formula $S_A^{(1)} : [d]^3 \to \{0, 1\}$ are defined as:

$$S_A^{(0)}(x, y) := \forall a \in [d], \neg\left(R_A(a, x) \wedge R_A(a, y)\right), \tag{2}$$

$$S_A^{(1)}(x, y \mid z) := S_{A_{-z}}^{(0)}(x, y) \wedge \left(\left(\forall a \in [d] \setminus \{z\}, S_{A_{-z}}^{(0)}(x, a) \vee \neg R_A(a, z)\right)\right.$$
$$\left. \vee \left(\forall b \in [d] \setminus \{z\}, S_{A_{-z}}^{(0)}(y, b) \vee \neg R_A(b, z)\right)\right). \tag{3}$$

Equivalently, the 0-th and 1-st order d-connection statements $C_A^{(0)}$ and $C_A^{(1)}$ are:

$$C_A^{(0)}(x, y) := \exists a \in [d], R_A(a, x) \wedge R_A(a, y), \tag{4}$$

$$C_A^{(1)}(x, y \mid z) := C_{A_{-z}}^{(0)}(x, y) \vee \left(\left(\exists a \in [d] \setminus \{z\}, C_{A_{-z}}^{(0)}(x, a) \wedge R_A(a, z)\right)\right.$$
$$\left. \wedge \left(\exists b \in [d] \setminus \{z\}, C_{A_{-z}}^{(0)}(y, b) \wedge R_A(b, z)\right)\right). \tag{5}$$

**Theorem 3.2.** *For any DAG $A$ with $d$ nodes and any three nodes $x, y, z \in [d]$ that are distinct, $x \perp\!\!\!\perp_A y$ if and only if $S_A^{(0)}(x, y) = 1$, and $x \perp\!\!\!\perp_A y \mid z$ if and only if $S_A^{(1)}(x, y \mid z) = 1$. Similarly, $x \not\perp\!\!\!\perp_A y$ if and only if $C_A^{(0)}(x, y) = 1$, and $x \not\perp\!\!\!\perp_A y \mid z$ if and only if $C_A^{(1)}(x, y \mid z) = 1$.* [4]

*Proof.* First, we note that the d-separation formulæ $S_A^{(0)}, S_A^{(1)}$ and the d-connection formulæ $C_A^{(0)}, C_A^{(1)}$ are the negation of each other and can be obtained from each other via De Morgan's law. Thus, the statement "$x \perp\!\!\!\perp_A y$ if and only if $S_A^{(0)}(x, y) = 1$" is equivalent to "$x \not\perp\!\!\!\perp_A y$ if and only if $C_A^{(0)}(x, y) = 1$", and vice versa, the statement "$x \perp\!\!\!\perp_A y \mid z$ if and only if $S_A^{(1)}(x, y \mid z) = 1$" is equivalent to "$x \not\perp\!\!\!\perp_A y \mid z$ if and only if $C_A^{(1)}(x, y \mid z) = 1$." Thus, we proceed to prove the statements involving the d-connection formulæ $C_A^{(0)}, C_A^{(1)}$.

**Part 1: $x \not\perp\!\!\!\perp_A y$ if and only if $C_A^{(0)}(x, y) = 1$.**

- We show the direction $x \not\perp\!\!\!\perp_A y \implies C_A^{(0)}(x, y) = 1$.

  If $x \not\perp\!\!\!\perp_A y$, i.e., $x$ is d-connected to $y$ with an empty conditioning set, then there exists a path between $x$ and $y$ that does not contain any colliders, because otherwise the path would be blocked. Denote the sequence of nodes in this path $\mathbf{p} = (p_1, \dots, p_m)$ with $p_1 = x$ and $p_m = y$. In other words, for any three consecutive nodes $a, b, c \in \mathbf{p}$, we have either the chain structure $a \to b \to c$ or $a \leftarrow b \leftarrow c$, or the fork structure $a \leftarrow b \to c$.

  Because there is no collider structure in $\mathbf{p}$, we will show that there must be *at most one* fork structure in $\mathbf{p}$. To see this by contradiction, consider if $\mathbf{p}$ contains more than one fork, and let $p_{l-2} \leftarrow p_{l-1} \to p_l$ and $p_r \leftarrow p_{r+1} \to p_{r+2}$ be two forks ($l < r$) in $\mathbf{p}$ such that there is no other fork between $p_l$ and $p_r$. Then, either there are some colliders between $p_l$ and $p_r$, or $p_l$ and $p_r$ are connected by some directed paths, i.e., either $p_l \rightsquigarrow p_r$ or $p_r \rightsquigarrow p_l$. If $p_l \rightsquigarrow p_r$, then $p_r$ is a collider, and vice versa if $p_r \rightsquigarrow p_l$, then $p_l$ is a collider. Hence, in all scenarios, $\mathbf{p}$ will have at least one collider, contradicting the fact that $\mathbf{p}$ should not have any collider.

  Now, we discuss case by case based on whether $\mathbf{p}$ has no fork, or it has one fork. Figure 1a illustrates the structure such a path $\mathbf{p}$.

  **Case 1: Suppose there is no fork on $\mathbf{p}$:** If there is no fork at all, then $\mathbf{p}$ is a directed path. Without loss of generality, assume $\mathbf{p} = x \to \cdots \to y$. Then, let $a = x$, we have $a \rightsquigarrow x$ (a node is always

---

[4]The proofs of Theorem 3.2 and all following theorems and lemmas can be found in Appendix A.

naively reachable from itself) and $a \rightsquigarrow y$. Hence, $(R_{\boldsymbol{A}}(a, x) \wedge R_{\boldsymbol{A}}(a, y)) = 1$, and therefore the formulæ $C_{\boldsymbol{A}}^{(0)}(x, y) = \exists a \in [d], R_{\boldsymbol{A}}(a, x) \wedge R_{\boldsymbol{A}}(a, y)$ is 1.

**Case 2: Suppose there is one fork on** $p$ When there is one and only one fork, it means the center node of the fork is the common ancestor of both $x$ and $y$. In other words, let $a$ be this center node, then we have $a \rightsquigarrow x$ and $a \rightsquigarrow y$. Thus, similarly, we have $(R_{\boldsymbol{A}}(a, x) \wedge R_{\boldsymbol{A}}(a, y)) = 1$, and therefore the formulæ $C_{\boldsymbol{A}}^{(0)}(x, y) = \exists a \in [d], R_{\boldsymbol{A}}(a, x) \wedge R_{\boldsymbol{A}}(a, y)$ is 1.

Thus, we have shown that $x \not\perp_{\boldsymbol{A}} y \implies C_{\boldsymbol{A}}^{(0)}(x, y) = 1$

- Now we show the other direction $C_{\boldsymbol{A}}^{(0)}(x, y) = 1 \implies x \not\perp_{\boldsymbol{A}} y$.

  Given $C_{\boldsymbol{A}}^{(0)}(x, y) = 1$, we have $\exists a \in [d]$ such that $a \rightsquigarrow x$ and $a \rightsquigarrow y$. Let $a$ be such a node. Thus, $a*$ is a common ancestor of $x$ and $y$. Let $\mathbf{p}_x = a \rightarrow \cdots \rightarrow x$ be the directed path that starts at $a$ and ends at $x$, and similarly let $\mathbf{p}_y = a \rightarrow \cdots \rightarrow y$. Then, the path form by the sequence of nodes in $\mathbf{p}_x$ and $\mathbf{p}_y$, i.e., $\mathbf{p} = (x, \ldots, a, \ldots, y)$, is clearly a path consisting of at most one fork structure with other wise chain structures. When $a = x$ or $a = y$, $\mathbf{p}$ is a directed path without fork, and otherwise has one fork. In either case, $\mathbf{p}$ is a d-connecting path between $x$ and $y$, making $x \not\perp_{\boldsymbol{A}} y$.

**Part 2:** $x \not\perp_{\boldsymbol{A}} y \mid z$ **if and only if** $C_{\boldsymbol{A}}^{(1)}(x, y \mid z) = 1$**.**

- We show the direction $x \not\perp_{\boldsymbol{A}} y \mid z \implies C_{\boldsymbol{A}}^{(1)}(x, y \mid z) = 1$.

  We are given $x \not\perp_{\boldsymbol{A}} y \mid z$, i.e., $x$ and $y$ are d-connecting when conditioning on $z$. Thus, there must exist a non-empty set of d-connecting paths $\mathcal{P}$ between $x$ and $y$ that is not blocked given $z$. We proceed with a case-by-case discussion depending on whether all of these paths contain colliders.

  1. First, we consider the case where there is some path in $\mathcal{P}$ that does not contain any collider. Let $\mathbf{p} \in \mathcal{P}$ be such a path. Then, $z$ must *not* be on $\mathbf{p}$, because otherwise, since $z$ is not a collider and it is being conditioned, $\mathbf{p}$ would be blocked and therefore $\mathbf{p} \notin \mathcal{P}$.
     Since $\mathbf{p}$ does not include $z$, it remains unchanged in the subgraph $\boldsymbol{A}_{-z}$, in which node $z$ is removed from the graph $\boldsymbol{A}$, and thus it is still a d-connecting path in $\boldsymbol{A}_{-z}$. Hence, because of $\mathbf{p}$, we have $x$ and $y$ are d-connecting in the subgraph $\boldsymbol{A}_{-z}$. In other words, we have $C_{\boldsymbol{A}_{-z}}^{(0)}(x, y) = 1$. Since $C_{\boldsymbol{A}_{-z}}^{(0)}(x, y)$ is the one of two terms from the outmost disjunction operator $\vee$ in the formula for $C_{\boldsymbol{A}}^{(1)}(x, y \mid z)$, we have $C_{\boldsymbol{A}_{-z}}^{(0)}(x, y) = 1$ renders $C_{\boldsymbol{A}}^{(1)}(x, y \mid z) = 1$.
     In Figure 1b, this pattern is illustrated via the "backdoor path" structure.
  2. Otherwise, all paths in $\mathcal{P}$ contain colliders. We first show that this implies that we can find a path $\mathbf{p}^*$ in $\mathcal{P}$ *that has only one collider*.
     To see this, pick any path $\mathbf{p} \in \mathcal{P}$. If $\mathbf{p}$ indeed has only one collider, then we are done. Hence, we consider the non-trivial case where $\mathbf{p}$ has more than one collider. We further divide the cases depending on whether node $z$ is included in path $\mathbf{p}$.
     (a) If $z$ is included in $\mathbf{p}$, then $z$ must itself be a collider, because otherwise $\mathbf{p}$ would be blocked by $z$.
         Now, let $z'$ be another collider in $\mathbf{p}$. Then, $z$ must be a descendant of $z'$, because otherwise $\mathbf{p}$ would be blocked by $z'$ even though $z$ is in the conditioning set. In other words, $z$ is reachable from $z'$ with a directed path, i.e., $z' \rightsquigarrow z$.
         Using this knowledge, we can construct the following alternative path $\mathbf{p}_{z'}$ that still has $z$ as a collider but converts $z'$ to a non-collider. Specifically, without loss of generality, we assume the path $\mathbf{p}$ consists of the following sequence of nodes: $\mathbf{p} = (x, \ldots, p_{l'}, z', p_{r'}, \ldots, p_l, z, p_r, \ldots, y)$ for some indices $l, r, l', r'$ with collider structures $p_{l'} \rightarrow z' \leftarrow p_{r'}$ and $p_l \rightarrow z \leftarrow p_r$. We have shown that $z' \rightsquigarrow z$, and denote the corresponding directed path $z' \rightarrow q_1 \rightarrow \cdots \rightarrow q_m \rightarrow z$. We can obtain the new path as follows:
         $$\mathbf{p}_{z'} = (x, \ldots, p_{l'}, z', q_1, \ldots, q_m, z, p_r, \ldots, y),$$
         where we replace the middle section sub-path $(z', p_{r'}, \ldots, p_l, z)$ in $\mathbf{p}$ to $(z', q_1, \ldots, q_m, z)$ to form $\mathbf{p}_{z'}$.

Now, note that in $\mathbf{p}_{z'}$, $z'$ is no longer a collider but forms a chain structure, because $p_{l'} \to z' \to q_1$. Nevertheless, $z$ is still a collider, because $q_m \to z \leftarrow p_r$. Furthermore, $\mathbf{p}_{z'}$ is still a d-connecting path. This is because the sub-paths $(x, \ldots, p_{l'}, z')$ and $(z, p_r, \ldots, y)$ are d-connecting conditioning on $z$ because they are parts of the original $p$ which is a d-connecting path, and the "new" sub-path $(z', q_1, \ldots, q_m, z)$ by definition is d-connecting conditioning on $z$ because it is a directed path. Hence, taken together, $\mathbf{p}_{z'}$ is d-connecting given $z$, so $\mathbf{p}_{z'} \in \mathcal{P}$.

Thus, by iteratively constructing these alternative paths $\mathbf{p}_{z'}$ for any colliders $z'$ in $\mathbf{p}$ that is not $z$, we can arrive at a path $\mathbf{p}^* \in \mathcal{P}$ that has $z$ as the only collider.

(b) Now, suppose $z$ is not included in $\mathbf{p}$ and $\mathbf{p}$ has more than one collider. Let $z'$ and $z''$ be any two such colliders, and without loss of generality, let the path consists of the sequence of nodes $p = (x, \ldots, p_{l'}, z', p_{r'}, \ldots, p_{l''}, z'', p_{r''}, \ldots, y)$. Similarly, both $z'$ and $z''$ must be ancestors of $z$, because otherwise one of them would block the path. Hence, we know that $z' \rightsquigarrow z$ and $z'' \rightsquigarrow z$. Let the directed paths be $z' \to q_1 \to \cdots \to q_m \to z$ and $z'' \to o_1 \to \cdots \to o_s \to z$ respectively. We can construct the following new path:

$$\mathbf{p}_{z',z''} = (x, \ldots, p_{l'}, z', q_1, \ldots, q_m, z, o_s, \ldots, o_1, z'', p_r, \ldots, y).$$

Here, note that both $z'$ and $z''$ are no longer a collider, because $p_{l'} \to z' \to q_1$ and $o_1 \leftarrow z'' \leftarrow p_r$. Nevertheless, $z$ is a collider since $q_m \to z \leftarrow o_s$. Furthermore, $\mathbf{p}_{z',z''}$ is still a d-connecting path conditioned on $z$. As we have established in the discussion of the previous case, that the sub-paths $(x, \ldots, p_{l'}, z')$ and $(z'', p_r, \ldots, y)$ are d-connecting. Since $(z', q_1, \ldots, q_m, z)$ and $(z, o_s, \ldots, o_1, z'')$ are directed paths, we have that the sub-paths $(x, \ldots, q_m, z)$ and $(z, o_s, \ldots, y)$ are also d-connecting. Taken together, the whole path is d-connecting given $z$, i.e., $\mathbf{p}_{z',z''} \in \mathcal{P}$.

In other words, we have converted both colliders $z'$ and $z''$ in the original $\mathbf{p}$ into non-colliders while introducing $z$ as a new collider. This means that the resulting path $\mathbf{p}_{z',z''}$ satisfies the initial condition of a path $\mathbf{p}$ discussed in the previous case. Hence, by further following the procedure described in the previous case, we can eventually arrive at a path $\mathbf{p}^* \in \mathcal{P}$ that has $z$ as the only collider.

Now that we know $\mathcal{P}$ will necessarily include a path $\mathbf{p}^*$ with only one collider, denote this collider $z^*$, and we know either $z^* = z$ or $z^*$ is an ancestor of $z$. In other words, we have $z^* \rightsquigarrow z$. Hence, we can denote $\mathbf{p}^*$ as a sequence of nodes: $\mathbf{p}^* = (x, \ldots, p_l^*, z^*, p_r^* \ldots, y)$ for some indices $l, r$, where $p_l^* \to z^* \leftarrow p_r^*$.

Now, let $a = x$ if there is no fork structure in the sub-path $(x, \ldots, z^*)$ of $\mathbf{p}^*$, otherwise let $a$ be the *rightmost* fork in $(x, \ldots, z^*)$ (i.e. there is no other fork in the sub-path $(a, \ldots, z^*)$ of $\mathbf{p}^*$). Then, we have $a \rightsquigarrow z^*$ because there is neither any fork nor any collider in the sub-path $(a, \ldots, z^*)$, and we know it has a constituent edge $p_l^* \to z^*$ with an edge direction pointing towards $z^*$. Moreover, because $z^* \rightsquigarrow z$, we have $a \rightsquigarrow z$, and equivalently $R_{\boldsymbol{A}}(a, z)$.

On the other hand, the sub-path $(x, \ldots, a)$ is a d-connecting path conditioning on $z$ because its super-path $\mathbf{p}^* \in \mathcal{P}$ is a d-connecting path conditioning on $z$, does not have any collider because $z^*$ is the only collider in $\mathbf{p}^*$, and does not include $z$. Hence, in the subgraph $\boldsymbol{A}_{-z}$ where $z$ is removed from the graph $\boldsymbol{A}$, the sub-path $(x, \ldots, a)$ remains unchanged and is still a d-connecting path, and in this case without conditioning on $z$. Therefore, $x$ and $a$ is d-connecting in $\boldsymbol{A}_{-z}$, i.e., $x \not\perp_{\boldsymbol{A}_{-z}} a$. Hence, we have found a node $a$ such that $(x \not\perp_{\boldsymbol{A}_{-z}} a) \wedge (a \rightsquigarrow z)$, or equivalently, we have the following:

$$\left( \exists a \in [d] \setminus \{z\}, (x \not\perp_{\boldsymbol{A}_{-z}} a) \wedge R_{\boldsymbol{A}}(a, z) \right) = 1.$$

Similarly, let $b = y$ if there is no fork structure in the sub-path $(z, \ldots, y)$ of $\mathbf{p}^*$, otherwise let $b^*$ be the *leftmost* fork in $(z, \ldots, y)$. We then have also have the following:

$$\left( \exists b \in [d] \setminus \{z\}, (y \not\perp_{\boldsymbol{A}_{-z}} b) \wedge R_{\boldsymbol{A}}(b, z) \right) = 1.$$

Combining everything, we fulfill second term in the outermost disjunction operator in the FOL formula and thus have $C_{\boldsymbol{A}}^{(1)}(x, y \mid z) = 1$.

In Figure 1c, this pattern is illustrated as the "frontdoor path" structure.

- Now we show the other direction $C_{\boldsymbol{A}}^{(1)}(x, y \mid z) = 1 \implies x \not\perp_{\boldsymbol{A}} y \mid z$.

If $C_{\boldsymbol{A}}^{(1)}(x, y \mid z) = 1$, then we have

$$C_{\boldsymbol{A}_{-z}}^{(0)}(x, y) = 1,$$

or we have

$$\left(\left(\exists a \in [d] \setminus \{z\}, C_{\boldsymbol{A}_{-z}}^{(0)}(x, a) \wedge R_{\boldsymbol{A}}(a, z)\right) \wedge \left(\exists b \in [d] \setminus \{z\}, C_{\boldsymbol{A}_{-z}}^{(0)}(y, b) \wedge R_{\boldsymbol{A}}(b, z)\right)\right) = 1.$$

We discuss it case by case.

1. If it is the first case, $C_{\boldsymbol{A}_{-z}}^{(0)}(x, y) = 1$, then there exists a d-connecting path $\mathbf{p}$ between $x$ and $y$ in the subgraph $\boldsymbol{A}_{-z}$. $\mathbf{p}$ then does not include $z$, and must not include any colliders. Hence, back in the original graph $\boldsymbol{A}$, $\mathbf{p}$ is a "backdoor" path that does not involve $z$ that renders $x$ and $y$ d-connecting. Thus, we have $x \not\!\perp_{\boldsymbol{A}} y$ and also $x \not\!\perp_{\boldsymbol{A}} y \mid z$.

2. We now consider the other case. Since the left-hand-side term is a conjunction ($\wedge$), both term in the conjunction must be true, meaning there exists some nodes $a$ and $b$ satisfying either term respectively. Taking a step further, this means that all of the four terms – $C_{\boldsymbol{A}_{-z}}^{(0)}(x, a)$, $R_{\boldsymbol{A}}(a, z)$, $C_{\boldsymbol{A}_{-z}}^{(0)}(y, b)$, and $R_{\boldsymbol{A}}(b, z)$ – are true.

   Using a similar reasoning as mentioned in the previous case, $C_{\boldsymbol{A}_{-z}}^{(0)}(x, a)$ implies that we have a d-connecting path between $x$ and $a$ and that $x \not\!\perp_{\boldsymbol{A}} a \mid z$. Similarly, $C_{\boldsymbol{A}_{-z}}^{(0)}(y, b)$ implies that $y \not\!\perp_{\boldsymbol{A}} b \mid z$. Furthermore, because $R_{\boldsymbol{A}}(a, z) = 1$ and $R_{\boldsymbol{A}}(b, z) = 1$, that is, $a \rightsquigarrow z$ and $b \rightsquigarrow z$, the path between $a$ and $b$, formed by the corresponding directed paths that renders $a \rightsquigarrow z$ and $b \rightsquigarrow z$ respectively, is a d-connecting path conditioning on $z$, as $z$ serves as the only collider. Hence, we have $a \not\!\perp_{\boldsymbol{A}} b \mid z$.

   Hence, by transitivity of d-connection statements, we have that $x \not\!\perp_{\boldsymbol{A}} y \mid z$, finishing the entire proof.

$\square$

**Lemma 3.4.** *For any discrete graph $\boldsymbol{A} \in \{0, 1\}^{d \times d}$ with maximum directed path length $l$ and for all pair of nodes $x, y \in [d]$, $x \rightsquigarrow_{\boldsymbol{A}} y$ if and only if $R_{\boldsymbol{A}}^{(l)}(x, y) = 1$.*

*Proof.* We prove this lemma by induction on the maximum directed path length $l$.

**Basecase**: Suppose the maximum directed path length in $\boldsymbol{A}$ is 0. This means there is no path and $\boldsymbol{A}$ is an empty graph. Thus, for all $x \in [d]$, $x \rightsquigarrow_{\boldsymbol{A}} x$ and for all $x, y \in [d]$ with $x \neq z$, $x \not\rightsquigarrow_{\boldsymbol{A}} y$.

**Induction**: Suppose the statement in the lemma holds true with a maximum directed path length of $l - 1$. We now prove the statement for $l$.

- We prove the direction $x \rightsquigarrow_{\boldsymbol{A}} y \implies R_{\boldsymbol{A}}^{(l)}(x, y) = 1$.

  Given that $x \rightsquigarrow_{\boldsymbol{A}} y$, let $\mathbf{p}$ be such a directed path starting from $x$ and ending at $y$. Let $u$ be the second to last node in $\mathbf{p}$ with an edge $u \rightarrow y$, or equivalently, $\boldsymbol{A}_{u,y} = 1$. Then, we also have $x \rightsquigarrow_{\boldsymbol{A}} u$.

  Since the path from $x$ to $u$ will have a length smaller than or equal to $l - 1$, using the induction hypothesis, we have $R_{\boldsymbol{A}}^{(l-1)}(x, u) = 1$.

  Thus, we have $R_{\boldsymbol{A}}^{(l-1)}(x, u) \wedge \boldsymbol{A}_{u,y} = 1$, and consequently $\bigvee_{u \in d} R_{\boldsymbol{A}}^{(l-1)}(x, u) \wedge \boldsymbol{A}_{u,y} = 1$, satisfying the first term in the outmost disjunction in the recursive formula for $R_{\boldsymbol{A}}^{(l)}$, rendering $R_{\boldsymbol{A}}^{(l)}(x, y) = 1$.

- Now we prove the other direction $R_{\boldsymbol{A}}^{(l)}(x, y) = 1 \implies x \rightsquigarrow_{\boldsymbol{A}} y$.

  Given $R_{\boldsymbol{A}}^{(l)}(x, y) = 1$, there are possible scenarios: that $\bigvee_{u \in d} R_{\boldsymbol{A}}^{(l-1)}(x, u) \wedge \boldsymbol{A}_{u,y} = 1$ or that $R_{\boldsymbol{A}}^{(l-1)}(x, y) = 1$. we discuss case by case.

1. Suppose $\bigvee_{u \in d} R_{\boldsymbol{A}}^{(l-1)}(x, u) \wedge \boldsymbol{A}_{u,y} = 1$, then there exists at least one node $u$ such that $R_{\boldsymbol{A}}^{(l-1)}(x, u) \wedge \boldsymbol{A}_{u,y} = 1$, meaning both $R_{\boldsymbol{A}}^{(l-1)}(x, u) = 1$ and $\boldsymbol{A}_{u,y} = 1$. This means that, by the induction hypothesis, $x \rightsquigarrow_{\boldsymbol{A}} u$ with maximum path length $l - 1$. Furthermore, we have $u \to y$ is an directed edge. Thus, there exists a directed path from $x$ to $y$, i.e. $x \rightsquigarrow_{\boldsymbol{A}} y$ with maximum path length $l$.

2. Suppose $R_{\boldsymbol{A}}^{(l-1)}(x, y) = 1$, then by the inductive hypothesis, $x \rightsquigarrow_{\boldsymbol{A}} y$ with a maximum path length $l - 1$, which also makes the statement true under the setting of a maximum path length of $l$.

Thus, in either case, we have $x \rightsquigarrow_{\boldsymbol{A}} y$.

$\square$

The proof of Lemma 3.7 and subsequent Theorem 3.8 requires the following lemma showing the lower-bound nature of LogLTN's [2] t-norm and t-conorm logical operators. We state this lemma and its proof as follows.

**Lemma A.1.** *Given the t-norm and t-conorm operators of LogLTN [2], which we restate as follows,*

$$\tilde{T}_m(\{x'_i\}_{i \in [m]}) := \sum_{i=1}^{m} x'_i \quad , \tilde{O}_m(\{x'_i\}_{i \in [m]}) := \alpha \left( C + \log \left( \frac{\sum_{i=1}^{n} e^{x'_i/\alpha - C}}{m} \right) \right) \quad .$$

*Let $\mathbf{x}_1, \mathbf{x}_2, \ldots, \mathbf{x}_m$ be $m$ Bernoulli random variables that are mutually independent or positively correlated. Denote their probabilities as $p_1, p_2, \ldots, p_m$, respectively. Then, we have the following:*

$$\tilde{T}_m(\{\log(p_1), \ldots, \log(p_m)\}) \leq \log \mathbb{P}(\mathbf{x}_1 \wedge \cdots \wedge \mathbf{x}_m)$$
$$\tilde{O}_m(\{\log(p_1), \ldots, \log(p_m)\}) \leq \log \mathbb{P}(\mathbf{x}_1 \vee \cdots \vee \mathbf{x}_m)$$

*Proof.* We first show the inequality involving the t-norm $\tilde{T}_m$. We first observe that $\tilde{T}_m$ is the logarithm of the product t-norm, meaning

$$\exp(\tilde{T}_m(\{\log(p_1), \ldots, \log(p_m)\})) = \prod_{i=1}^{m} p_i.$$

Now, since $\mathbf{x}_i$'s are mutually independent or correlated, we naturally have $\prod_{i=1}^{m} p_i \leq \mathbb{P}(\mathbf{x}_1 \wedge \cdots \wedge \mathbf{x}_m)$. This is because for any two random variables $\mathbf{x}_i$ and $\mathbf{x}_j$,

$$\mathbb{P}(\mathbf{x}_i \wedge \mathbf{x}_j) = \mathbb{E}[\mathbf{x}_i \wedge \mathbf{x}_j] = \mathbb{E}[\mathbf{x}_i]\mathbb{E}[\mathbf{x}_j] + \text{Cov}(\mathbf{x}_i, \mathbf{x}_j) = p_i p_j + \text{Cov}(\mathbf{x}_i, \mathbf{x}_j) \geq p_i p_j,$$

where the last inequality sign holds because $\text{Cov}(\mathbf{x}_i, \mathbf{x}_j) \geq 0$ for mutually positively correlated or independent random variables.

Thus, we have

$$\exp(\tilde{T}_m(\{\log(p_1), \ldots, \log(p_m)\})) \leq \prod_{i=1}^{m} p_i \leq \mathbb{P}(\mathbf{x}_1 \wedge \cdots \wedge \mathbf{x}_m)$$
$$\tilde{T}_m(\{\log(p_1), \ldots, \log(p_m)\}) \leq \log \mathbb{P}(\mathbf{x}_1 \wedge \cdots \wedge \mathbf{x}_m).$$

For the t-conorm $\tilde{O}_m$, the inequality holds because $\tilde{O}_m$ is a lower bound of the logarithm of max, and max is then also a lower bound of the union of random variables. Specifically, as shown in the main text, $\tilde{O}_m$ as the LogMeanExp operation has the following property:

$$\exp(\tilde{O}_m(\{\log(p_1), \ldots, \log(p_m)\})) \leq \max\{p_1, \ldots, p_m\}.$$

On the other hand, max is in general the lower bound of the probability of the union of random variables,

$$\max\{\log(p_1), \ldots, p_m\} \leq \mathbb{P}(\mathbf{x}_1 \vee \cdots \vee \mathbf{x}_m).$$

Thus, taken together, we have

$$\exp(\tilde{O}_m(\{\log(p_1),\ldots,\log(p_m)\})) \leq \max\{p_1,\ldots,p_m\} \leq \mathbb{P}(\mathbf{x}_1 \vee \cdots \vee \mathbf{x}_m)$$
$$\tilde{O}_m(\{\log(p_1),\ldots,\log(p_m)\}) \leq \log \mathbb{P}(\mathbf{x}_1 \vee \cdots \vee \mathbf{x}_m).$$

$\square$

**Lemma 3.7** (Reachability Percolation Lower Bound). *Given a weighted adjacency matrix $\mathbf{W} \in [0,1]^{d \times d}$, for any $0 \leq l < d$, and for any pair of nodes $x, y \in [d]$, we have*

$$\tilde{R}^{(l)}_{\mathbf{W}}(x,y) \leq \log \mathbb{E}_{\mathbf{A}\sim\mathrm{Bern}(\mathbf{W})}\left[R^{(l)}_{\mathbf{A}}(x,y)\right] \quad \text{and} \quad \tilde{U}^{(l)}_{\mathbf{W}}(x,y) \leq \log \mathbb{E}_{\mathbf{A}\sim\mathrm{Bern}(\mathbf{W})}\left[U^{(l)}_{\mathbf{A}}(x,y)\right].$$

*Proof.* We first show that $\tilde{R}^{(l)}_{\mathbf{W}}(x,y) \leq \log \mathbb{E}_{\mathbf{A}\sim\mathrm{Bern}(\mathbf{W})}\left[R^{(l)}_{\mathbf{A}}(x,y)\right]$.

Intuitively, this inequality holds because in a probabilistic graph $\mathbf{W}$, where each edge $x \to y$ is a Bernoulli random variable parameterized by $\mathbf{W}_{x,y}$, the paths as random variables are either mutually independent, when they consist of distinct edges, or mutually positively correlated, when they share some common edges. Thus, using the results of Lemma A.1, the t-norm of the path probabilities yields a lower bound to their joint probability, and the t-conorm yields a lower bound to their union probability. Thus, the continuous reachability score $\tilde{R}^{(l)}_{\mathbf{W}}$, consisting of disjunctions and conjunctions of the path and edge probabilities, also gives lower bounds.

More specifically, we can observe this fact via induction. In the base case, we obviously have

$$\tilde{R}^{(0)}_{\mathbf{W}}(x,y) = \log(\mathbb{1}(x=y)) = \log \mathbb{P}_{\mathbf{A}\sim\mathrm{Bern}(\mathbf{W})}(R^{(0)}_{\mathbf{A}}(x,y)) = \mathbb{P}_{\mathbf{W}}(x \to y) = \log \mathbf{W}_{x,y}.$$

In addition, we can obviously see that the random variable $R^{(0)}_{\mathbf{A}}(x,u)$ is independent of the random variable $\mathbf{A}_{u,y}$ for any nodes $u \in [d]$.

Then, assuming the inequality holds for a graph with a maximum path length of $l-1$, and that the random variable $R^{(l-1)}_{\mathbf{A}}(x,u)$ is either independent of or positively correlated with the random variable $\mathbf{A}_{u,y}$ for any nodes $u \in [d]$, we now aim to show that these two statements also hold for $l$.

First, we can see that, for any node $u \in [d]$,

$$\tilde{T}_2(\tilde{R}^{(l-1)}_{\mathbf{W}}(x,u), \log(W_{uy})) \leq \log \mathbb{E}_{\mathbf{A}\sim\mathrm{Bern}(\mathbf{W})}\left[R^{(l-1)}_{\mathbf{A}}(x,u) \wedge \mathbf{A}_{u,y}\right],$$

from results in Lemma A.1 regarding the t-norm $\tilde{T}_2$ and the induction hypothesis that $R^{(l-1)}_{\mathbf{A}}(x,u)$ is either independent of or positively correlated with $\mathbf{A}_{u,y}$.

Then, again using the result in Lemma A.1 regarding the t-conorm $\tilde{O}_m$, we have

$$\tilde{R}^{(l)}_{\mathbf{W}}(x,y) = \tilde{O}_{d+1}\left(\{\tilde{T}_2(\tilde{R}^{(l-1)}_{\mathbf{W}}(x,u), \log(W_{uy}))\}_{u\in[d]} \cup \{\tilde{R}^{(l-1)}_{\mathbf{W}}(x,y)\}\right)$$

$$\leq \log \mathbb{E}_{\mathbf{A}\sim\mathrm{Bern}(\mathbf{W})}\left[\left(\bigvee_{u\in[d]}\left(R^{(l-1)}_{\mathbf{A}}(x,u) \wedge \mathbf{A}_{u,y}\right)\right) \vee R^{(l-1)}_{\mathbf{A}}(x,y)\right]$$

$$= \log \mathbb{E}_{\mathbf{A}\sim\mathrm{Bern}(\mathbf{W})}\left[R^{(l)}_{\mathbf{A}}(x,y)\right].$$

Regarding the relationship between $R^{(l)}_{\mathbf{A}}(x,u)$ and $\mathbf{A}_{u,y}$, we can see that the probability of $R^{(l)}_{\mathbf{A}}(x,u)$ increases if the probability of $\mathbf{A}_{u,y}$ also increases. This is because in the formula, there is no negation and $\mathbf{A}_{u,y}$ makes a possible contribution to the value of $R^{(l)}_{\mathbf{A}}(x,u)$. Thus, $R^{(l)}_{\mathbf{A}}(x,u)$ and $\mathbf{A}_{u,y}$ are positively correlated.

Now, we show for the unreachability inequality, $\tilde{U}^{(l)}_{\mathbf{W}}(x,y) \leq \log \mathbb{E}_{\mathbf{A}\sim\mathrm{Bern}(\mathbf{W})}\left[U^{(l)}_{\mathbf{A}}(x,y)\right].$

Similarly, we show by induction. In the base case, we have

$$\tilde{U}^{(0)}_{\mathbf{W}}(x,y) = \log(\mathbb{1}(x \neq y)) = \log \mathbb{P}_{\mathbf{A}\sim\mathrm{Bern}(\mathbf{W})}(\neg R^{(0)}_{\mathbf{A}}(x,y)) = \log(1 - \mathbf{W}_{x,y}).$$

In addition, we can see that the random variable $U_{\boldsymbol{A}}^{(0)}(x,u)$ is independent of the random variable $\neg R_{\boldsymbol{A}}^{(0)}(x,y)$.

Then, assuming the inequality holds for a graph with a maximum path length of $l-1$, and that the random variable $U_{\boldsymbol{A}}^{(l-1)}(x,u)$ is either independent of or positively correlated with the random variable $\neg R_{\boldsymbol{A}}^{(0)}(x,y)$. We aim to show that these two properties still hold for $l$.

First, using the result in Lemma A.1 regarding the t-conorm $\tilde{O}_m$, we have, for all nodes $u \in [d]$,

$$\tilde{O}_2(\tilde{U}_{\boldsymbol{W}}^{(l-1)}(x,u), \log(1-W_{uy})) \le \mathbb{E}_{\boldsymbol{A}\sim\mathrm{Bern}(\boldsymbol{W})}\left[U_{\boldsymbol{A}}^{(l-1)}(x,u) \vee \neg R_{\boldsymbol{A}}^{(0)}(u,y)\right].$$

Then, we note that the random variable $(U_{\boldsymbol{A}}^{(l-1)}(x,u) \vee \neg R_{\boldsymbol{A}}^{(0)}(u,y))$ for different $u$ are mutually independent or positively correlated. They are also independent of or positively correlated with $U_{\boldsymbol{A}}^{(l-1)}(x,y)$. This is similar to the mutual independence or positive correlation among paths, since for these "non-paths", if they consist of distinct "non-edges", then they are independent. Otherwise, they are positively correlated, since increasing the probability of the shared "non-edge" (or decreasing the probability of the shared edge) would simultaneously increase the probabilities of said "non-paths." Thus, again using the result in Lemma A.1 regarding the t-norm $\tilde{T}_{d+1}$, we have

$$\tilde{U}_{\boldsymbol{W}}^{(l)}(x,y) = \tilde{T}_{d+1}\left(\{\tilde{O}_2(\tilde{U}_{\boldsymbol{W}}^{(l-1)}(x,u), \log(1-W_{uy}))\}_{u\in[d]} \cup \{\tilde{U}_{\boldsymbol{W}}^{(l-1)}(x,y)\}\right)$$

$$\le \log \mathbb{E}_{\boldsymbol{A}\sim\mathrm{Bern}(\boldsymbol{W})}\left[\left(\bigwedge_{u\in[d]}\left(U_{\boldsymbol{A}}^{(l-1)}(x,u) \vee \neg\boldsymbol{A}_{u,y}\right)\right) \wedge U_{\boldsymbol{A}}^{(l-1)}(x,y)\right]$$

$$= \log \mathbb{E}_{\boldsymbol{A}\sim\mathrm{Bern}(\boldsymbol{W})}\left[U_{\boldsymbol{A}}^{(l)}(x,y)\right].$$

$\square$

**Theorem 3.8** (Lower Bound on Expected $d$-Separation Statements). *Given a weighted adjacency matrix $\boldsymbol{W} \in [0,1]^{d\times d}$, for any three nodes $x, y, z \in [d]$,*

$$\tilde{S}_{\boldsymbol{W}}^{(0)}(x,y) \le \log \mathbb{E}_{\boldsymbol{A}\sim Bern(\boldsymbol{W})}\left[S_{\boldsymbol{A}}^{(0)}(x,y)\right], \quad \tilde{S}_{\boldsymbol{W}}^{(1)}(x,y\mid z) \le \log \mathbb{E}_{\boldsymbol{A}\sim Bern(\boldsymbol{W})}\left[S_{\boldsymbol{A}}^{(1)}(x,y\mid z)\right],$$

$$\tilde{C}_{\boldsymbol{W}}^{(0)}(x,y) \le \log \mathbb{E}_{\boldsymbol{A}\sim Bern(\boldsymbol{W})}\left[C_{\boldsymbol{A}}^{(0)}(x,y)\right], \quad \tilde{C}_{\boldsymbol{W}}^{(1)}(x,y\mid z) \le \log \mathbb{E}_{\boldsymbol{A}\sim Bern(\boldsymbol{W})}\left[C_{\boldsymbol{A}}^{(1)}(x,y\mid z)\right].$$

*Proof.* First, we reiterate that, in the computation of differentiable d-separation $\tilde{S}_{\boldsymbol{W}}^{(0)}$ and $\tilde{S}_{\boldsymbol{W}}^{(1)}$, we use the continuous unreachability score $\tilde{U}_{\boldsymbol{W}}^{(d)}$ directly, rather than taking the negation of the continuous reachability score.

Using a similar reasoning as in the proof of the previous lemma, we first observe that, when treating $S_{\boldsymbol{A}}^{(0)}, S_{\boldsymbol{A}}^{(1)}, C_{\boldsymbol{A}}^{(0)}$, and $C_{\boldsymbol{A}}^{(1)}$ as random variables when $\boldsymbol{A} \sim \mathrm{Bern}(\boldsymbol{W})$, all terms in the d-separation/d-connection formulæ (Definition 3.1) are either mutually independent or positively correlated. In particular, this holds true for $S_{\boldsymbol{A}_{-z}}^{(0)}(x,a)$ and $\neg R_{\boldsymbol{A}}(a,z)$ for any $a \in [d]$, as well as for $C_{\boldsymbol{A}_{-z}}^{(0)}(y,b)$ and $R_{\boldsymbol{A}}(b,z)$ for any $b \in [d]$. Thus, using the results from Lemma A.1, we have the lower bounds as stated in the theorem. $\square$

**Lemma 4.2** (Consistency of Multi-Task CI Losses). *Given a faithful data $\mathcal{D}$, as the sample size $n \to \infty$ and the LogMeanExp temperature $\alpha \to 0^+$ (Equation (8)), the optimal DAG $\boldsymbol{A}^*$ achieves the minimum value on each of $\mathcal{L}_{TP\text{-}0}, \mathcal{L}_{TP\text{-}1}, \mathcal{L}_{TN\text{-}0}, \mathcal{L}_{TN\text{-}1}$, and $\mathcal{L}_{DAG}$.*

*Proof.* Let $\theta^* \in \mathbb{R}^{d\times d}$ be the parameter that approximates the optimal binary DAG $\boldsymbol{A}*$ of the given data $\mathcal{D}$ via $\sigma(\theta^*) \to \boldsymbol{A}^*$.

When the sample size $n \to \infty$ in a faithful data $\mathcal{D}$, we have $p_{\mathcal{D}}(x,y) \to 0$ and $p_{\mathcal{D}}(x,y\mid z) \to 0$ when $x \not\perp\!\!\!\perp_{\boldsymbol{A}^*} y$ and $x \not\perp\!\!\!\perp_{\boldsymbol{A}^*} y \mid z$, and conversely $p_{\mathcal{D}}(x,y) \to 1$ and $p_{\mathcal{D}}(x,y\mid z) \to 1$ when $x \perp\!\!\!\perp_{\boldsymbol{A}^*} y$ and $x \perp\!\!\!\perp_{\boldsymbol{A}^*} y \mid z$ for any $x,y,z \in [d]$, where $\boldsymbol{A}^*$ is the optimal causal graph of $\mathcal{D}$. When the LogMeanExp temperature $\alpha \to 0^+$, we have $\log \max\{x_i\}_{i=1}^m - \tilde{O}_m(\{\log(x_i)\}_{i=1}^m) \to 0^+$.

In addition, we note that when given a sequence of binary random variables $\mathbf{x}_1, \ldots, \mathbf{x}_m$, whose Bernoulli probabilities approach in the limit to either 0 or 1, the product t-norm $T_m$ and max t-conorm $O_m$ over $\mathbf{x}_i$'s probabilities approach in the limit to the probability of $\mathbb{P}(\mathbf{x}_1 \wedge \cdots \wedge \mathbf{x}_m)$ and $\mathbb{P}(\mathbf{x}_1 \vee \cdots \vee \mathbf{x}_m)$ respectively. This is because in the limit case, the random variables reduce to constant 0's and constant 1's, in which case the product t-norm and t-conorm reduce to the vanilla Boolean operations of conjunction and disjunction. Thus, combining the fact that LogLTN's [2] t-norm $\tilde{T}_m$ is equivalent to $T_m$ in the logarithm, and its t-conorm $\tilde{O}_m$ approaches in limit to $O_m$ in the logarithm, we have the following:

$$\log \mathbb{P}(\mathbf{x}_1 \wedge \cdots \wedge \mathbf{x}_m) - \tilde{T}_m(\{\log \mathbb{P}(\mathbf{x}_i)\}_{i=1}^m) \to 0^+$$

$$\log \mathbb{P}(\mathbf{x}_1 \vee \cdots \vee \mathbf{x}_m) - \tilde{O}_m(\{\log \mathbb{P}(\mathbf{x}_i)\}_{i=1}^m) \to 0^+.$$

Using this property, we can see that for the continuous reachability and unreachability, for any given maximum path length $l$ and any nodes $x, y \in [d]$, they satisfy

$$R_{\boldsymbol{A}^*}^{(l)}(x, y) - R_{\sigma(\theta^*)}^{(l)}(x, y) \to 0^+$$

$$U_{\boldsymbol{A}^*}^{(l)}(x, y) - U_{\sigma(\theta^*)}^{(l)}(x, y) \to 0^+.$$

Consequently, for the differentiable d-separation and d-connection scores, for any nodes $x, y, z \in [d]$, they satisfy

$$S_{\boldsymbol{A}^*}^{(0)}(x, y) - \tilde{S}_{\sigma(\theta^*)}^{(0)}(x, y) \to 0^+, \qquad S_{\boldsymbol{A}^*}^{(1)}(x, y \mid z) - \tilde{S}_{\sigma(\theta^*)}^{(1)}(x, y \mid z) \to 0^+,$$

$$C_{\boldsymbol{A}^*}^{(0)}(x, y) - \tilde{C}_{\sigma(\theta^*)}^{(0)}(x, y) \to 0^+, \qquad C_{\boldsymbol{A}^*}^{(1)}(x, y \mid z) - \tilde{C}_{\sigma(\theta^*)}^{(1)}(x, y \mid z) \to 0^+.$$

Furthermore, recall from Theorem 3.2, since $\boldsymbol{A}^*$ is a discrete DAG, $S_{\boldsymbol{A}^*}^{(0)}$, $S_{\boldsymbol{A}^*}^{(1)}$, $C_{\boldsymbol{A}^*}^{(0)}$, and $C_{\boldsymbol{A}^*}^{(1)}$ reduces exactly to the ground-truth d-separation and d-connection statement values in $\boldsymbol{A}^*$. Thus, combining with the result above, we have that

$$\tilde{S}_{\sigma(\theta^*)}^{(0)}(x, y) \to \begin{cases} 1 & \text{when } x \perp\!\!\!\perp_{\boldsymbol{A}^*} y \\ 0 & \text{otherwise} \end{cases}, \qquad \tilde{S}_{\sigma(\theta^*)}^{(1)}(x, y \mid z) \to \begin{cases} 1 & \text{when } x \perp\!\!\!\perp_{\boldsymbol{A}^*} y \mid z \\ 0 & \text{otherwise} \end{cases},$$

$$\tilde{C}_{\sigma(\theta^*)}^{(0)}(x, y) \to \begin{cases} 1 & \text{when } x \not\perp\!\!\!\perp_{\boldsymbol{A}^*} y \\ 0 & \text{otherwise} \end{cases}, \qquad \tilde{C}_{\sigma(\theta^*)}^{(1)}(x, y \mid z) \to \begin{cases} 1 & \text{when } x \not\perp\!\!\!\perp_{\boldsymbol{A}^*} y \mid z \\ 0 & \text{otherwise} \end{cases},$$

Therefore, combining with the values of the p-values $p_{\mathcal{D}}$, we have the following value matching between the model's predicted d-separation and d-connection score and the p-values. For all $(x, y) \in \mathbb{I}_0$ ($\mathbb{I}_0 = \{(x, y) \mid x, y \in [d], x > y\}$) and for all $(x, y, z) \in \mathbb{I}_1$ ($\mathbb{I}_1 = \{(x, y, z) \mid x, y, z \in [d], x > y, x \neq z, y \neq z\}$),

$$\tilde{S}_{\sigma(\theta^*)}^{(0)}(x, y) \to \begin{cases} 1 & \text{when } p_{\mathcal{D}}(x, y) \to 1 \\ 0 & \text{when } p_{\mathcal{D}}(x, y) \to 0 \end{cases}, \qquad \tilde{S}_{\sigma(\theta^*)}^{(1)}(x, y \mid z) \to \begin{cases} 1 & \text{when } p_{\mathcal{D}}(x, y \mid z) \to 1 \\ 0 & \text{when } p_{\mathcal{D}}(x, y \mid z) \to 0 \end{cases},$$

$$\tilde{C}_{\sigma(\theta^*)}^{(0)}(x, y) \to \begin{cases} 1 & \text{when } M_0 - p_{\mathcal{D}}(x, y) \to 1 \\ 0 & \text{when } M_0 - p_{\mathcal{D}}(x, y) \to 0 \end{cases}, \qquad \tilde{C}_{\sigma(\theta^*)}^{(1)}(x, y \mid z) \to \begin{cases} 1 & \text{when } M_1 - p_{\mathcal{D}}(x, y \mid z) \to 1 \\ 0 & \text{when } M_1 - p_{\mathcal{D}}(x, y \mid z) \to 0 \end{cases},$$

where $M_0 = \max_{\mathbb{I}_0} p_{\mathcal{D}}(x, y) \to 1$, and similarly $M_1 = \max_{\mathbb{I}_1} p_{\mathcal{D}}(x, y \mid z) \to 1$. In other words, the differentiable d-separation/d-connection scores over the optimal parameter $\theta^*$) and the p-values from the data have zero mismatch.

Thus, the TP and TN losses have the following values:

$$\mathcal{L}_{\text{TP-0}}(\theta, \mathcal{D}) \to -\sum_{(x,y) \in \mathbb{I}_0} \mathbb{1}[x \perp\!\!\!\perp_{\boldsymbol{A}^*} y], \qquad \mathcal{L}_{\text{TP-1}}(\theta, \mathcal{D}) \to -\sum_{(x,y,z) \in \mathbb{I}_1} \mathbb{1}[x \perp\!\!\!\perp_{\boldsymbol{A}^*} y \mid z],$$

$$\mathcal{L}_{\text{TN-0}}(\theta, \mathcal{D}) \to -\sum_{(x,y) \in \mathbb{I}_0} \mathbb{1}[x \not\perp\!\!\!\perp_{\boldsymbol{A}^*} y], \qquad \mathcal{L}_{\text{TN-1}}(\theta, \mathcal{D}) \to -\sum_{(x,y,z) \in \mathbb{I}_1} \mathbb{1}[x \not\perp\!\!\!\perp_{\boldsymbol{A}^*} y \mid z],$$

which are the lowest possible values that these loss functions can achieve, because any other configuration of the values for the differentiable d-separation/d-connection scores would result in a larger loss value due to possible mismatches with the p-values. Thus, the TP and TN losses are consistent.

Finally, since $\boldsymbol{A}^*$ is a DAG, the log-det acyclicity loss $\mathcal{L}_{\text{DAG}}$ also achieves the minimum value, for any value of the hyperparameter $s$. This property is proved in Bello et al. [4]. Thus, all loss functions in Definition 4.1 are consistent. □

**Lemma 4.3** (Mutual Independence of Low-order CI Statements). *All 0th- and 1st-order CI statements over the variable sets $\mathbb{I}_0 = \{(x,y) \mid x,y \in [d], x > y\}$ and $\mathbb{I}_1 = \{(x,y,z) \mid x,y,z \in [d], x > y, x \neq z, y \neq z\}$ are mutually independent. In other words, none of these CI statements can be implied from any other of these CI statements via the graphoid axioms [24].*

*Proof.* Let $\mathcal{X}, \mathcal{Y} \subseteq [d]$ be any non-empty node sets, and $\mathcal{Z}, \mathcal{W} \subseteq [d]$ be node sets that could be empty. The graphoid axiom [24] states the following five rules describing the relationship and dependencies between different CI statements in a graphoid dependency model:

1. Symmetry: $\mathcal{X} \perp\!\!\!\perp \mathcal{Y} \mid \mathcal{Z} \iff \mathcal{Y} \perp\!\!\!\perp \mathcal{X} \mid \mathcal{Z}$

2. Decomposition: $\mathcal{X} \perp\!\!\!\perp \mathcal{Y} \cup \mathcal{W} \mid \mathcal{Z} \implies (\mathcal{X} \perp\!\!\!\perp \mathcal{Y} \mid \mathcal{Z}) \wedge (\mathcal{X} \perp\!\!\!\perp \mathcal{W} \mid \mathcal{Z})$

3. Weak Union: $\mathcal{X} \perp\!\!\!\perp \mathcal{Y} \cup \mathcal{W} \mid \mathcal{Z} \implies (\mathcal{X} \perp\!\!\!\perp \mathcal{Y} \mid \mathcal{Z} \cup \mathcal{W}) \wedge (\mathcal{X} \perp\!\!\!\perp \mathcal{W} \mid \mathcal{Z} \cup \mathcal{Y})$

4. Contraction: $(\mathcal{X} \perp\!\!\!\perp \mathcal{Y} \mid \mathcal{Z}) \wedge (\mathcal{X} \perp\!\!\!\perp \mathcal{W} \mid \mathcal{Z} \cup \mathcal{Y}) \implies \mathcal{X} \perp\!\!\!\perp \mathcal{Y} \cup \mathcal{W} \mid \mathcal{Z}$

5. Intersection: $(\mathcal{X} \perp\!\!\!\perp \mathcal{Y} \mid \mathcal{Z} \cup \mathcal{W}) \wedge (\mathcal{X} \perp\!\!\!\perp \mathcal{W} \mid \mathcal{Z} \cup \mathcal{Y}) \implies \mathcal{X} \perp\!\!\!\perp \mathcal{Y} \cup \mathcal{W} \mid \mathcal{Z}$

We check rule by rule whether any of the low-order CI statements considered in $\mathbb{I}_0$ and $\mathbb{I}_1$ can be implied by any others.

1. Symmetry: No CI statement in $\mathbb{I}_0$ and $\mathbb{I}_1$ can be inferred from others via the symmetry rule. This is because $\mathbb{I}_0$ and $\mathbb{I}_1$ consists of only asymmetric statements where always $x > y$.

2. Decomposition: In the non-trivial case when $|\mathcal{Y}| \geq 1$ and $|\mathcal{W}| \geq 1$, this rule does not apply because its left-hand side (LHS) statement involves nodes set $\mathcal{Y} \cup \mathcal{W}$ with cardinality of at least 2. This type of statement is not considered in $\mathbb{I}_0$ or $\mathbb{I}_1$.

3. Weak Union: Similarly, in the non-trivial case when $|\mathcal{Y}| \geq 1$ and $|\mathcal{W}| \geq 1$, this rule does not apply because its LHS statement is not included in $\mathbb{I}_0$ or $\mathbb{I}_1$.

4. In the non-trivial case when $|\mathcal{Y}| \geq 1$ and $|\mathcal{W}| \geq 1$, the LHS applies, but the right-hand-side (RHS) involve node set $\mathcal{Y} \cup \mathcal{W}$ with cardinality of at least 2, which is not included in $\mathbb{I}_0$ or $\mathbb{I}_1$.

5. Intersection: Similarly, in the non-trivial case when $|\mathcal{Y}| \geq 1$ and $|\mathcal{W}| \geq 1$, the RHS involve node set $\mathcal{Y} \cup \mathcal{W}$ with cardinality of at least 2, which is not included in $\mathbb{I}_0$ or $\mathbb{I}_1$.

Thus, no CI statements in $\mathbb{I}_0$ and $\mathbb{I}_1$ can be inferred from other statements in $\mathbb{I}_0$ and $\mathbb{I}_1$ via the graphoid axioms. Thus, the CI statements we considered in our approach are mutually independent from the perspective of the graphoid dependency structure, justifying the use of the sum aggregation over these CI statements in the TP and TN losses. □

## B  Further Discussions on Percolation versus Diffusion

The distinction between graph percolation and graph diffusion is fundamental to understanding the probabilistic interpretation of our differentiable d-separation framework. Percolation theory studies graph properties (such as connectivity or reachability) when the graph structure itself is randomized [12, 16]. In contrast, diffusion theory typically assumes a fixed graph structure, with randomness arising from particles making probabilistic traversal decisions across this deterministic environment.

In the context of our weighted adjacency matrix $\mathbf{W} \in \mathbb{R}^{d \times d}$, these perspectives offer different interpretations. From a percolation standpoint, each entry $\mathbf{W}_{x,y}$ represents the probability that edge $x \to y$ exists in the graph. The reachability question then becomes: what is the probability that a directed path exists from one node to another, considering all possible graph configurations? From a diffusion perspective, $\mathbf{W}$ would instead represent transition probabilities on a complete graph, where

Random Graph $\mathbf{W}$ :

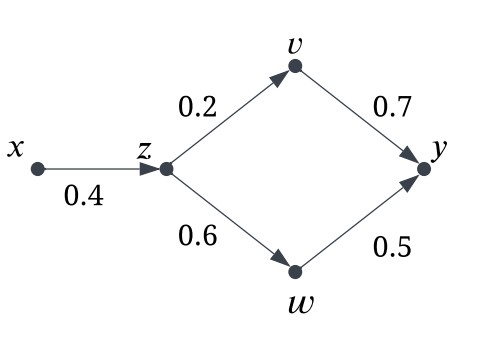

True Reachability Probability (Percolation):

$$\mathbb{P}(x \rightsquigarrow y) = 0.4 \cdot (0.2 \cdot 0.7 + 0.6 \cdot 0.5$$
$$-0.2 \cdot 0.7 \cdot 0.6 \cdot 0.5)$$
$$= 0.1592$$

Max-Product Bellman-Ford is Lower Bound:

$$R_{\mathbf{W}}^{(3)}(x, y) = 0.4 \cdot \max\{0.2 \cdot 0.7, 0.6 \cdot 0.5\}$$
$$= 0.12$$

Random Walk (Diffusion) is Upper Bound:

$$\mathbb{P}^{(\mathrm{RW})}(x \rightsquigarrow y) = 0.4 \cdot (0.2 \cdot 0.7 + 0.6 \cdot 0.5)$$
$$= 0.176$$

Figure 5: Example random graph to illustrate that the Max-Product Bellman-Ford reachability computes a lower bound of the true percolation-based reachability probability, whereas diffusion-based random walk computes an upper bound. Marked numbers are the edge probabilities. For the random walk computation, we assume a model with self-loop probabilities, e.g. $P(x \rightarrow x) = 0.6$ so that the transition probabilities to other nodes and the self-loop probabilities sum to one.

at node $x$, the weights $\{\mathbf{W}_{x,u}\}_{u \in [d]}$ (possibly normalized) determine the probability distribution of a particle's next position. The key distinction is that percolation randomizes the environment (the graph structure), while diffusion randomizes the trajectory through a fixed environment.

It is therefore most appropriate to frame our problem as a graph percolation problem. Given a weighted adjacency matrix $\mathbf{W}$, we interpret it as parametrizing a distribution $\mathrm{Bern}(\mathbf{W})$ from which random discrete graphs are sampled, $\mathbf{A} \sim \mathrm{Bern}(\mathbf{W})$. Our goal is to estimate the expected reachability (with path lengths up to $d$), $\mathbb{E}_{\mathbf{A} \sim \mathrm{Bern}(\mathbf{W})}\left[R_{\mathbf{A}}^{(d)}(x, y)\right]$, where the randomness comes from the graph structure itself.

This insight guides our estimation approach for the percolation-based reachability. We observe that our generalized Bellman-Ford-based differentiable reachability score (Definition 3.5) provides a lower bound on this expectation, while graph diffusion methods like Random Walk algorithms typically yield upper bounds. This distinction becomes clear when considering two potential paths $\mathbf{p}_1$ and $\mathbf{p}_2$ from node $x$ to node $y$ that share some edges. In percolation theory, reachability requires only that at least one path forms completely. Our Bellman-Ford approach with max operators captures this intuition—when one path's formation probability approaches 1, additional paths contribute minimally to the overall reachability probability. The lower bound property stems from the approximation gap between product-max t-norms/t-conorms and the true intersection/union of events, combined with the approximation in LogLTN's [2] LogMeanExp operator (Equation (8)). Formal proof of this lower bound property can be found in the proof for Lemma 3.7 in Appendix A. Consequently, our method will systematically *underestimate* true percolation-based reachability.

Conversely, in diffusion models, particles make independent decisions at each node. When two paths overlap, the probability of a particle reaching the target via either path is treated additively rather than as a union probability. This independence assumption fails to account for the correlation between overlapping paths in the percolation setting, causing diffusion methods to *overestimate* the true percolation-based reachability.

Figure 5 provides a concrete example illustrating the distinction between our max-product Bellman-Ford reachability and graph diffusion scores. The example shows a weighted adjacency matrix $\mathbf{W}$ with 5 nodes, where edge weights are as indicated in the figure (with zeros elsewhere). For the diffusion-based method, we choose the typical Random Walk algorithm, where we assume self-loops at each node so that the sum of self-loop probability and outgoing edge probabilities equals 1 (e.g., node $x$ has a self-loop probability of 0.6).

Consider two possible paths from $x$ to $y$: path $\mathbf{p}_1 = x \rightarrow z \rightarrow v \rightarrow y$ and path $\mathbf{p}_2 = x \rightarrow z \rightarrow w \rightarrow y$. $\mathbf{p}_1$ and $\mathbf{p}_2$ are two random variables under the graph distribution $\mathrm{Bern}(\boldsymbol{W})$. The true percolation-based reachability probability, denoted by $\mathbb{P}(x \rightsquigarrow y)$, is computed as the union of the

events of these two paths forming: $\mathbb{P}(\mathbf{p}_1 \vee \mathbf{p}_2)$. Since these paths share the edge $x \to z$ but diverge afterward, we must carefully apply probability theory on union of events. For the divergent segments, the probability is:

$$\mathbb{P}((z \to v \to y) \vee (z \to w \to y)) = \mathbb{P}(z \to v \to y) + \mathbb{P}(z \to w \to y) - \mathbb{P}((z \to v \to y) \wedge (z \to w \to y))$$

Because these two sub-paths share no edges, they are independent, giving $\mathbb{P}((z \to v \to y) \wedge (z \to w \to y)) = \mathbb{P}(z \to v \to y) \cdot \mathbb{P}(z \to w \to y)$. Calculating the full expression yields $\mathbb{P}(x \rightsquigarrow y) = 0.1592$.

Our max-product Bellman-Ford approach approximates this union using the max operator: for the divergent sub-paths from $z$ to $y$, we compute $R_{\mathbf{W}}^{(2)}(z, y) = \max\{\mathbb{P}(z \to v \to y), \mathbb{P}(z \to w \to y)\}$. Since the max operator always provides a lower bound for the union of events, our approach yields the final value $R_{\mathbf{W}}^{(3)}(x, y) = 0.12$, which underestimates the true reachability probability. In contrast, the diffusion-based Random Walk method, denoted by $\mathbb{P}^{(\mathrm{RW})}(x \rightsquigarrow y)$, adds probabilities additively across possible paths: $\mathbb{P}^{(\mathrm{RW})}(x \rightsquigarrow y) = \mathbb{P}(\mathbf{p}_1) + \mathbb{P}(\mathbf{p}_2)$. This leads to an overestimation of the true reachability probability, giving $\mathbb{P}^{(\mathrm{RW})}(x \rightsquigarrow y) = 0.176$. This example clearly demonstrates why our max-product approach provides a principled lower bound on percolation-based reachability, while diffusion-based methods typically yield upper bounds.

Our max-product Bellman-Ford approach is thus particularly well-suited for our proposed model instantiation, DAGPA, as it provides consistent lower bounds throughout computation, ensuring a coherent probabilistic interpretation. While we focus on lower-bound estimation, an alternative approach could theoretically estimate upper bounds on expected d-separation/d-connection statements. For such a method, diffusion-based approaches like Random Walk would be more appropriate for estimating graph reachability. However, this would require identifying a different pair of t-norm/t-conorm differentiable logical operators that yield upper bounds instead of lower bounds. Based on van Krieken et al. [38]'s comprehensive analysis of t-norm and t-conorm suitability for differentiable learning, we have not identified a configuration that simultaneously provides upper bounds and maintains gradient stability comparable to our max-product operators. This represents an interesting direction for future research, potentially expanding the theoretical foundations of differentiable d-separation frameworks, and we encourage future work to explore this avenue.

## C  Full Algorithm and Implementation Details of DAGPA

DAGPA, our proposed instantiation of the differentiable d-separation framework, employs multiple techniques to address optimization challenges and result validation. This section provides comprehensive implementation details for all techniques used in our approach. We first discuss the multi-task gradient conflict resolution technique (Section C.1) and the Bayesian sampling method for enhanced weight space exploration (Section C.2). Since DAGPA uses Bayesian sampling, each optimization step generates a new candidate causal graph. Consequently, we require a principled procedure to select the best DAGs from those sampled, using only information available from the input data. We present this DAG selection heuristic in Section C.3. Finally, we provide the complete algorithm pseudocode in Section C.4.

### C.1  Multitask optimizer to address conflicting gradients

The five distinct loss functions—0th and 1st-order TP/TN losses plus the DAG acyclicity constraint (Definition 4.1)—render our optimization a multi-objective problem. Recent research in multi-task learning has demonstrated that naively summing these loss functions leads to theoretically suboptimal convergence, as contradictory gradient directions among tasks can impede optimization progress [20, 22, 31, 40, 43]. This challenge is particularly pronounced in our framework, where the TP losses encouraging d-separation

---

**Algorithm 1** PCGrad [43] Gradient Projection

**Require:** Model parameters $\theta$, number of tasks $K$, loss functions $\{\mathcal{L}_k\}_{k=1}^K$
1: $\boldsymbol{g}_k \leftarrow \nabla_\theta \mathcal{L}_k(\theta) \quad \forall k \in [K]$
2: $\boldsymbol{g}_k^{\mathrm{PC}} \leftarrow \boldsymbol{g}_k \quad \forall k \in [K]$
3: **for** $i = 1 : K$ **do**
4:     **for** $j \overset{\text{uniformly}}{\sim} [K] \setminus i$ in random order **do**
5:         **if** $\boldsymbol{g}_i^{\mathrm{PC}} \cdot \boldsymbol{g}_j < 0$ **then**
6:             ▷ Subtract projection of $\boldsymbol{g}_i^{\mathrm{PC}}$ onto $\boldsymbol{g}_j$
7:             $\boldsymbol{g}_i^{\mathrm{PC}} \leftarrow \boldsymbol{g}_i^{\mathrm{PC}} - \frac{\boldsymbol{g}_i^{\mathrm{PC}} \cdot \boldsymbol{g}_j}{\|\boldsymbol{g}_j\|^2} \boldsymbol{g}_j$
8: **return** update $\Delta\theta^{(\mathrm{PC})} = \boldsymbol{g}^{\mathrm{PC}} = \sum_{i=1}^K \boldsymbol{g}_i^{\mathrm{PC}}$

naturally conflict with the TN losses that require d-connection. To address these inherent gradient conflicts, we adopt PCGrad [43], a gradient projection technique that resolves conflicting directions by projecting one gradient onto the normal plane of the other when they conflict. We reproduce the PCGrad algorithm [43] adopted to our setting in Algorithm 1.

## C.2 Gradient-informed discrete Bayesian sampling of $W$

Rather than using conventional stochastic gradient descent (SGD), we adopt Discrete Langevin Proposal (DLP) [50], a gradient-informed discrete Bayesian sampling technique, to sample parameters $\theta$ that yield low-loss configurations[5]. Our choice of DLP is motivated by several key factors. First, our empirical observations show that SGD often becomes trapped in local minima, possibly due to the highly non-convex loss landscape, in which case Bayesian sampling can naturally escape local minima and explores the weight space more effectively. Second, we favor discrete over continuous sampling because the most likely DAG $\mathbf{A}$ from a weighted adjacency matrix $\mathbf{W}$ is obtained via thresholding at 0.5. This threshold divides the parameter space into discrete regions, and continuous optimization or sampling approaches spend significant computational effort exploring within a single region, repeatedly yielding the same causal graph structure. In contrast, DLP's discrete support enables efficient exploration across different regions while leveraging gradient information from the continuous space to guide proposals.

To adopt DLP for DAGPA, we first restrict the parameter space from the real space $\mathbb{R}^{d \times d}$ to a space of discrete supports, $\mathbb{D}^{d \times d}$, where $\mathbb{D} = \{D_1, \ldots, D_m \mid D_i < D_j, \forall i < j\}$ is a finite set of values. Furthermore, we require $D_1 < 0$ and $D_m > 0$, so that for any pair of nodes $(x, y)$, if $\theta_{x,y} = D_1$, then $\mathbf{W}_{x,y} = \sigma(\theta_{x,y}) < 0.5$, which allows us to interpret $x, y$ as more likely to not having an edge. Similarly, if $\theta_{x,y} = D_m$, then $\mathbf{W}_{x,y} = \sigma(\theta_{x,y}) > 0.5$, which allows us to interpret $x, y$ as more likely to have an edge $x \to y$.

In reality through empirical experiments, we found that $\mathbb{D} = \{-2.0, 0.0, 2.0\}$ works quite well. We hypothesize this could be due to $-2.0$ and $2.0$ as sigmoid logits are not too extreme so that they enable smooth and meaningful gradients, while also sufficiently far apart for the differentiable d-separation formulæ. Furthermore, the "middle point" $0.0$ allows DAGPA to model uncertain edges, whose final values can only to be decided after other edges are settled.

The next step to adopt DLP [50] is to define the proposal sampling distribution and the Metrpolis-Hastings(MH) acceptance-rejection step [15, 21]. This is the discrete Metropolis-adjusted Langevin algorithm (DMALA) variant of DLP algorithm [50]. One of DLP's novel innovations is the parallel sampling of the parameters $\theta$ when its proposal distribution can be factorized along the dimensions, meaning at every step, multiple $\theta_i$ for different $i$'s may have their values updated. This technique significantly enhances DLP's efficiency compared to other discrete sampling methods. DAGPA fits this requirement as we can formulate a factorizable proposal distribution. Specifically, we adopt Equation (2) from Zhang et al. [50], but change the gradients to those obtained via PCGrad [43] from Algo-

---
**Algorithm 2** DLP (DMALA) [50] Sampling Step
---
**Require:** Model parameters $\theta$, step size $\beta$, gradient $\Delta\theta^{(\text{PC})}$, current energy $U(\theta)$
1: // Proposal Step
2: **for** i = 1 : d **do**      ▷ Can be done in parallel
3:     **construct** $q_i(\cdot \mid \theta)$ as in Equation (9)
4:     **sample** $\theta_i' \sim q_i(\cdot \mid \theta)$
5: // MH Acceptance Step
6: **compute** $U(\theta')$
7: **compute** $\Delta\theta'^{(\text{PC})}$ via PCGrad (Algorithm 1)
8: **compute** $q(\theta' \mid \theta) = \prod_i q_i(\theta_i' \mid \theta)$
9: **compute** $q(\theta \mid \theta') = \prod_i q_i(\theta_i \mid \theta')$
10: **set** $\theta \leftarrow \theta'$ with probability in Equation (10)
11: **return** new sample $\theta'$

---

rithm 1. That is, let $q(\theta' \mid \theta)$ be the proposal distribution. It can be factorized along the $d$ dimensions, $q(\theta' \mid \theta) = \prod_{i=1}^{d} q_i(\theta_i' \mid \theta)$, where $q_i(\theta_i' \mid \theta)$ is a categorical distribution of the form:

$$q_i(\theta_i' \mid \theta) := \text{Categorical}\left(\text{Softmax}\left(\frac{1}{2}\Delta\theta^{(\text{PC})}(\theta_i - \theta_i') - \frac{(\theta_i - \theta_i')^2}{2\beta}\right)\right), \qquad (9)$$

where $\Delta\theta^{(\text{PC})}$ is the projected gradients given by the PCGrad algorithm (Algorithm 1), and $\beta$ is a hyperparameter controlling the DLP step size.

---

[5]Note that this differs from the discrete DAG sampling $\mathbf{A} \sim \text{Bern}(\mathbf{W})$ in Section 3.2, which serves to theoretically establish the probabilistic interpretation of our continuous d-separation and d-connection values.

After a proposal $\theta'$ is sampled from $\theta$ according to Equation (9), we perform an MH acceptance-rejection step. Intuitively, this step is to ensure that we are not taking a step too far and landing into a "bad region" in the parameter space. Mathematically, this step is to ensure the Markov chain is reversible. Here, we adopt Equation (3) from Zhang et al. [50], but for the (negative) energy function, we simply adopt the heuristic of summing the multi-task losses (Definition 4.1). Thus, for DAGPA, we accepts the proposal $\theta'$ with probability

$$\min\left(1, \exp(U(\theta) - U(\theta'))\frac{q(\theta \mid \theta')}{q(\theta' \mid \theta)}\right), \tag{10}$$

where $U(\theta) = \mathcal{L}_{\text{TP-0}}(\theta, \mathcal{D}) + \mathcal{L}_{\text{TP-1}}(\theta, \mathcal{D}) + \mathcal{L}_{\text{TN-0}}(\theta, \mathcal{D}) + \mathcal{L}_{\text{TN-1}}(\theta, \mathcal{D}) + \mathcal{L}_{\text{DAG}}(\theta, s)$ with $\mathcal{D}$ the dataset.

We reproduce the DLP sampling algorithm adopted to DAGPA in Algorithm 2. We note, however, that the combination of PCGrad[43] projected gradients $\Delta\theta^{(\text{PC})}$ and the energy function $U(\theta)$ as the sum of multi-task losses may not form a mathematically well-defined and reversible Markov chain for the sampling, because the gradients are not directly derived from $U(\theta)$ but have been post-hoc modified via gradient projections. Nevertheless, we argue both theoretically and from empirical observations that the gradient modification step with either PCGrad or some alternative multi-task learning method is essential, as it addresses gradient conflicts and navigates the gradient landscape much more efficiently. Furthermore, we empirically found that the MH acceptance step using the energy function $U(\theta)$ obtained via summing the multi-task losses is sufficiently performative and, more importantly, provides the acceptance rate as an important indicator for adjusting the DLP step size $\beta$, which is one of the most important hyperparameters in DAGPA. We detail the hyperparameter choice in Appendix D.4. We invite future research to tackle this challenging of integrating multi-task gradients with MH acceptance step in a more principled way.

### C.3 Training-time DAG Selection

Finally, DAGPA employs a heuristic score to evaluate the quality of causal graphs obtained from sampled model parameters. This evaluation step is essential because DAGPA adopts DLP [50], and each DLP sampling iteration generates new model parameters and corresponding causal graphs, making it necessary to identify the highest-quality DAGs for final output. Importantly, this score is not a traditional "validation score" since it requires neither a separate validation dataset nor knowledge of the ground-truth causal structure. Instead, it operates solely on the input dataset $\mathcal{D}$ that is already used during optimization and sampling. This approach offers a practical advantage over model-based methods like NOTEARS [51] and DAGMA [4], which typically require splitting the dataset to create separate validation sets, thereby reducing the amount of data available for model training. Our score makes full use of all available data while providing a principled mechanism for DAG selection.

We name this score the "TPTN Ratio score", as it essentially checks the ratio of weighted true positive (TP) and true negative (TN) d-separation / CI statements predicted by the model, while treating the p-values from the dataset as the soft ground-truth labels. Specifically, given the current sampled model parameter $\theta$, we obtain the weighted adjacency matrix via $\boldsymbol{W} = \sigma(\theta)$, and then threshold it to convert it to a binary graph $\hat{\boldsymbol{A}}$, $\hat{\boldsymbol{A}}_{x,y} = \mathbb{1}[\boldsymbol{W}_{x,y} > 0.5], \forall x, y \in [d]$. Additionally, we would like to ensure that at every step of sampling, we are evaluating a DAG, as the graph $\hat{\boldsymbol{A}}$ converted from the model parameters $\theta$ found by the DLP sampling algorithm may not fully satisfy the acyclicity constraint. This requirement can be achieved by pruning $\hat{\boldsymbol{A}}$ to form an acyclic $\boldsymbol{A}$. In DAGPA, we proceed by finding a feedback arc set (ARC) from $\hat{\boldsymbol{A}}$ and remove all the edges in it to construct $\boldsymbol{A}$. We use the ARC method provided in the igraph package [7].

Then, setting the LogMeanExp temperature $\alpha$ (Equation (8)) to a very small value (e.g. $\alpha = 1e - 5$), we use the differentiable d-separation scores (Definition 3.6) on $\boldsymbol{A}$ to obtain $\tilde{S}_{\boldsymbol{A}}^{(0)}$ and $\tilde{S}_{\boldsymbol{A}}^{(1)}$. Since $\alpha$ is small but not exactly 0, $\exp(\tilde{S}_{\boldsymbol{A}}^{(0)})$ and $\exp(\tilde{S}_{\boldsymbol{A}}^{(0)})$ are close to but not exactly 0's or 1's. Thus, we threshold again to obtain the binary d-separation statements, $S_{\boldsymbol{A}}^{(0)}(x, y) = \mathbb{1}[\exp(\tilde{S}_{\boldsymbol{A}}^{(0)})(x, y) > 0.5]$ and $S_{\boldsymbol{A}}^{(1)}(x, y \mid z) = \mathbb{1}[\exp(\tilde{S}_{\boldsymbol{A}}^{(1)})(x, y \mid z) > 0.5]$. We treat these as the binary d-separation statements predicted by the model.

Then, given the p-values $p_{\mathcal{D}}$ from the data, we compute the true positives (TP), true negatives (TN), false positives (FP), and false negative (FN) scores as:

$$\text{TP}(\boldsymbol{A}, \mathcal{D}) = \sum_{x,y \in [d]} S_{\boldsymbol{A}}^{(0)}(x, y \mid z) p_{\mathcal{D}}(x, y) + \sum_{x,y,z \in [d]} S_{\boldsymbol{A}}^{(1)}(x, y \mid z) p_{\mathcal{D}}(x, y \mid z)$$

$$\text{TN}(\boldsymbol{A}, \mathcal{D}) = \sum_{x,y \in [d]} (1 - S_{\boldsymbol{A}}^{(0)}(x, y \mid z))(1 - p_{\mathcal{D}}(x, y)) + \sum_{x,y,z \in [d]} (1 - S_{\boldsymbol{A}}^{(1)}(x, y \mid z))(1 - p_{\mathcal{D}}(x, y \mid z))$$

$$\text{FP}(\boldsymbol{A}, \mathcal{D}) = \sum_{x,y \in [d]} S_{\boldsymbol{A}}^{(0)}(x, y \mid z)(1 - p_{\mathcal{D}}(x, y)) + \sum_{x,y,z \in [d]} S_{\boldsymbol{A}}^{(1)}(x, y \mid z)(1 - p_{\mathcal{D}}(x, y \mid z))$$

$$\text{FN}(\boldsymbol{A}, \mathcal{D}) = \sum_{x,y \in [d]} (1 - S_{\boldsymbol{A}}^{(0)}(x, y \mid z)) p_{\mathcal{D}}(x, y) + \sum_{x,y,z \in [d]} (1 - S_{\boldsymbol{A}}^{(1)}(x, y \mid z)) p_{\mathcal{D}}(x, y \mid z).$$

And then the TPTN Ratio score is computed as

$$\text{TPTN-Ratio}(\boldsymbol{A}, \mathcal{D}) = \frac{\text{TP}(\boldsymbol{A}, \mathcal{D}) + \text{TN}(\boldsymbol{A}, \mathcal{D})}{\text{TP}(\boldsymbol{A}, \mathcal{D}) + \text{TN}(\boldsymbol{A}, \mathcal{D}) + \text{FP}(\boldsymbol{A}, \mathcal{D}) + \text{FN}(\boldsymbol{A}, \mathcal{D})}, \tag{11}$$

which ranges in $[0, 1]$ and the higher the score, the better the causal graph in matching the low-order CI statements found in the data.

## C.4 Full algorithm of DAGPA

Combining all techniques together, DAGPA's full algorithm is given in Algorithm 3.

---

**Algorithm 3** DAGPA

---

**Require:** Data $\mathcal{D}$, initial parameter $\theta_0$, number of steps $T$, number of best DAGs $K$, step size $\beta$
1: **for** $t = 0 : T - 1$ **do**
2:     $\boldsymbol{W}_t \leftarrow \sigma(\theta_t)$
3:     **for** $(x, y) \in [d]^2$ **do**                                                          ▷ Can be done in parallel
4:         **compute** $\tilde{S}_{\boldsymbol{W}_t}^{(0)}$ and $\tilde{C}_{\boldsymbol{W}_t}^{(0)}$ as in Definition 3.6
5:     **for** $(x, y, z) \in [d]^3$ **do**                                                      ▷ Can be done in parallel
6:         **compute** $\tilde{S}_{\boldsymbol{W}_t}^{(1)}$ and $\tilde{C}_{\boldsymbol{W}_t}^{(1)}$ as in Definition 3.6
7:     **compute** $\mathcal{L}_{\text{TP-0}}, \mathcal{L}_{\text{TP-1}}, \mathcal{L}_{\text{TN-0}}, \mathcal{L}_{\text{TN-1}}, \mathcal{L}_{\text{DAG}}$ as in Definition 4.1
8:     $U(\theta_t) \leftarrow \mathcal{L}_{\text{TP-0}} + \mathcal{L}_{\text{TP-1}} + \mathcal{L}_{\text{TN-0}} + \mathcal{L}_{\text{TN-1}} + \mathcal{L}_{\text{DAG}}$
9:     **compute** $\Delta\theta_t^{(\text{PC})}$ via PCGrad [43] (Algorithm 1)
10:    **compute** $\theta_{t+1}$ via DLP [50] (Algorithm 2)
11:    **compute** $\boldsymbol{A}_{t+1}$ by converting $\theta_{t+1}$ to a discrete DAG (Appendix C.3)
12:    **compute** TPTN-Ratio$(\boldsymbol{A}_{t+1}, \mathcal{D})$
13: **return** $K$ DAGs from $\{\boldsymbol{A}_t\}_{t=1}^T$ with the Top-$K$ highest TPTN-Ratio$(\boldsymbol{A}_t, \mathcal{D})$ score

---

## C.5 Computation Complexity Analysis

The computational bottleneck of DAGPA is the p-values computation for all low-order CI statements and the reachability computation required by the differentiable d-separation formulae.

The complexity of p-values computation (which we GPU-accelerate) depends on the specific choice of statistical independence test. Take the Chi-squared test for example. Given $n$ data points, computing each single Chi-squared p-value takes $O(n)$ as this is the time required for iterating over all data points and building the contingency. Thus, the overall time required is $O(d^3 n)$ as we obtain a total of $d^2 + d^3$ number of p-values for the unconditional and first-order conditional statements.

For the differentiable d-separation formulae: the reachability subroutine (Definition 3.3) involves a maximum of $d$ recursive steps, and each all-pairs Bellman-Ford update step operates on $O(d^3)$ matrix entries (Equation (7)). Thus, the total computation time is $O(d^4)$. The resulting all-pairs reachability matrix will be cached and accessed by the d-connection/d-separation score computation. Now, since in the 1st-order d-connection/d-separation scores (Equations (3) and (5)), we need reachability on

the node-deleted subgraph $\mathbf{A}_{-z}$ for all conditioning node $z$, this requires a total of $d + 1$ all-pairs reachability matrices, thus rendering the total runtime of this part $O(d^5)$.

For the 0th-order d-separation/d-connection formulae (Equations (2) and (4)), computing $S^{(0)}\_\mathbf{A}(x, y)$ or $C^{(0)}\_\mathbf{A}(x, y)$ for each (x, y) takes $O(d)$ time, since it iterates over $d$ possible common ancestors and accessing the reachability cache takes $O(1)$. Thus, computing all-pairs 0th-order d-separation/d-connection scores take $O(d^3)$ time. Now, since later in the 1th-order formulae, we need all-pairs 0th-order d-separation/d-connection scores for the node-deleted subgraphs $A_{-z}$ for all conditioning node $z$, this adds another $d$ dimension. Thus, the total time is $O(d^4)$.

For the 1st-order d-separation/d-connection formulae (Equations (3) and (5)), similarly, computing the result for each query triple $(x, y \mid z)$ takes $O(d)$ time. Thus, getting the results for all triple of nodes takes a total of $O(d^4)$ computations.

Thus in summary, computing all-pairs p-values takes $O(d^3 n)$ computations, computing reachability matrix (and cache for later use) takes $O(d^5)$ computations, and computing 0th-order and 1st-order d-separation/d-connection scores each take $O(d^4)$ computations. The total will be dominated by the reachability computation, which is $O(d^5)$.

**Acceleration in Practice**     The total $O(d^5)$ time complexity notwithstanding, we seize many promising opportunities for acceleration in the code implementation of DAGPA.

First, we note that all computations of reachability and d-separation/d-connection scores are matrix-vector operations, which can benefit from *GPU vectorization*. In practice, we leverage PyTorch GPU tensor library for all such computations, avoiding any explicit for-loops and significantly improving the speed of optimization.

Second, in practice one can limit the reachability computation to only consider paths of a constant maximum length $k$, reducing the time complexity from $O(d^5)$ to $O(d^4 k)$, *rendering the entire pipeline* $O(d^4)$. In the experiments for larger graphs with $d = 50$ number of nodes (Appendix F.2), we limit the path length to $k = 25$. Observations during earlier method development suggest negligible performance degradation with significantly improved computational efficiency.

Third, the all-pairs p-values computation can be massively parallelized and then cached and reused. This is particularly useful in the case where one wishes to obtain multiple output causal graphs for the same input dataset to enhance solution diversity and exploration of the Markov equivalence class. In this case, the computation time for the all-pairs p-values step can be amortized as it can be reused in subsequent runs.

Finally, we expect future work to further examine how to incorporate sparsity assumptions into the graph to further reduce the time complexity. For example, instead of assuming each node can connect to all $d - 1$ other nodes, one reasonable assumption is to restrict to a maximum of constant $k$ degree. In that case, the reachability computation subroutine computation can drastically speed up.

## D    Experiment Details

### D.1    Code and dataset release

We release our code and data in

```
https://github.com/PurdueMINDS/DAGPA
```

### D.2    Dataset creation and conversion

We generated synthetic binary datasets following the approach introduced in the $k$-PC codebase [18]. We simulated DAGs using both Erdős–Rényi (ER) and Scale-Free (SF) graph models with $d \in \{10, 50\}$ nodes and expected edge-to-node ratios $r \in \{2, 4\}$. For ER graphs, we set edge probability $p = r/d$ to achieve the desired arc ratio. For SF graphs, we used the Barabási-Albert preferential attachment model with attachment parameter $m = \lfloor r/2 \rfloor$ to generate undirected graphs, then randomly oriented edges to form DAGs.

For each generated DAG adjacency matrix $A$, we constructed a binary Bayesian network using pyAgrum [10]. Each variable $X_i$ takes values in $\{0, 1\}$, and we randomly generated conditional

probability tables (CPTs) for all nodes. For root nodes (variables with no parents), we sampled marginal probabilities $P(X_i = 1) \sim \text{Uniform}(0.2, 0.8)$ to avoid extreme probabilities. For non-root nodes with parent set $\text{PA}(i)$, we sampled conditional probabilities $P(X_i = 1 \mid \text{PA}(i) = \mathbf{c}) \sim \text{Uniform}(0.2, 0.8)$ independently for each parent configuration $\mathbf{c} \in \{0, 1\}^{|\text{PA}(i)|}$. From these fully specified Bayesian networks, we drew $n \in \{100, 1000, 10000, 100000\}$ independent samples via ancestral sampling (forward sampling from the topological order). Since the data is binary, no normalization or scaling preprocessing was needed.

We also generated continuous data with simulated ER and SF DAG following the approach introduced in [4]. Specifically, for each DAG adjacency matrix $A$, we generated structural equation models of the form $X_i = \sum_{j \in \text{PA}(i)} w_{ji} f_j(X_j) + \epsilon_i$, where $\text{PA}(i)$ denotes the parent set of node $i$, edge weights $w_{ji}$ are sampled uniformly from $[-2.0, -0.5] \cup [0.5, 2.0]$, and noise terms $\epsilon_i \sim \mathcal{N}(0, 1)$ are independent Gaussian. For linear continuous data, we set $f_j(x) = x$. For nonlinear continuous data, we randomly assigned each $f_j$ to one of four nonlinear functions: $f_j(x) \in \{x^2, x^3, \sin(x), \cos(x)\}$ with equal probability. We generated datasets with the same sample sizes $n \in \{100, 1000, 10000, 100000\}$ and graph structures (ER and SF with $d \in \{10, 50\}$ nodes and arc ratios $r \in \{2, 4\}$) as the binary setting to enable direct comparison across data types.

For experiments on real-world dataset, we benchmarked our method and baselines over the Sachs dataset [28] and the LUCAS (LUng CAncer Simple set) dataset [14]. The Sachs dataset is a widely recognized benchmark in causal discovery research, consisting of protein signaling pathway data collected through flow cytometry experiments. It contains measurements of 11 phosphorylated proteins and phospholipids derived from thousands of individual primary immune system cells, gathered under various experimental conditions. What makes this dataset particularly valuable for benchmarking causal learning methods is that the pathways between these proteins are well-established in scientific literature, providing a reliable ground truth causal graph against which algorithms can be evaluated. The dataset has been extensively used in numerous studies to assess the performance of causal discovery algorithms, including recent work that demonstrates how less restrictive modeling approaches can capture complex causal relationships in this data that traditional methods assuming additive noise often fail to identify. We obtained and did benchmark on the subset of Sachs data containing approximately 800 samples with no perturbation. We used the standard discretized version preprocessed using the Hartemink discretization method, which converts the original continuous protein concentration measurements into 3-level categorical variables (low, average, high) while preserving the underlying dependence structure. The LUCAS dataset is an artificially generated benchmark specifically designed to evaluate causal discovery algorithms under different conditions. It features binary variables in a causal Bayesian network structure and comes in several variants that present increasing levels of challenge. Among all experiments introduced in LUCAS, we used LUCAS0 (baseline unmanipulated data) of size 2000.

### D.3 Evaluation metrics

The primary metric we adopt in this work is the Conditional Independence Matthews Correlation Coefficient (CI-MCC), which we describe in detail in Section 5 with a visual illustration in Figure 2.

Additionally, we evaluate our method and baselines using the following standard graph structure-based metric:

- **CPDAG Arrowhead F1:** We compare predicted causal graphs with the ground-truth CPDAG derived from the ground-truth DAG. Since different methods produce varying graph types, we standardize all outputs by converting them to CPDAGs. For methods like NOTEARS [51] and DAGMA [4] that return DAGs, we convert their outputs to CPDAGs using the utility method from the causal-learn package [53]. For k-PC [18], which returns k-essential graphs containing circle-marked edges (in addition to directed and undirected edges), we treat circle-marked edges as undirected edges in the CPDAG conversion. This treatment is appropriate because circle-marked edges in k-essential graphs denote edges whose directions cannot be determined from low-order CI statements alone. Once both predicted and ground-truth graphs are converted to CPDAGs, we compute the arrowhead F1 score exclusively over the directed edges in the CPDAGs.

- **CPDAG Skeleton F1:** Following the same standardization process as CPDAG Arrowhead F1, we convert all predicted causal graphs to CPDAGs. We then further convert all directed edges in both

predicted and ground-truth CPDAGs to undirected edges, creating graph skeletons. The F1 score is computed on these undirected skeleton graphs.

- **CPDAG Structural Hamming Distance (SHD):** After standardizing predicted causal graphs by converting them to CPDAGs, we compute the structural Hamming distance (SHD) between predicted and ground-truth CPDAGs. Each unit of SHD corresponds to a single edge difference between the two CPDAGs, including: missing edges, extra edges, incorrect edge orientations (directed vs. undirected), or incorrect edge directions.

- **DAG F1:** We also evaluate methods that directly output DAGs by comparing them with the ground-truth DAG. Since constraint-based methods like GES [6], PC [33], and k-PC [18] return CPDAGs or k-essential graphs rather than DAGs, this metric only applies to score-based methods: our approach, NOTEARS (linear and nonlinear) [51, 52], and DAGMA [4].

We note that these graph structure-based metrics may pose an inherent disadvantage to our method compared to CI-MCC. Through empirical observation, we have identified cases where DAGPA produces DAGs with similarly high CI-MCC scores but vastly different performance on structure-based metrics. For instance, one sampled DAG may achieve both high CI-MCC and high structure-based scores, while another DAG with comparable CI-MCC performance may score poorly on structure-based metrics. This suggests that while DAGPA consistently samples DAGs that align well with conditional independence patterns (as measured by CI-MCC), this alignment does not necessarily translate to high performance on traditional graph structure metrics. We provide detailed analysis of this phenomenon in Appendix F.1.

### D.4  DAGPA model details and hyperparameters

The most important and sensitive hyperparameter in DAGPA is the DLP sampling step size $\beta$ (Equation (9)). To this end, we first find values for all other hyperparameters through preliminary experimentations then fix them, and only vary in the step size for the experiments on the synthetic binary dataset and the real-world datasets.

Some of the other important hyperparameters and their values:

- **DLP support logit set** $\mathbb{D}$**:** This hyperparameter controls the support logits that the model parameter $\theta$ can take during sampling. We use $\mathbb{D} = [-2.0, 0.0, 2.0]$.

- **LogMeanExp temperature** $\alpha$**:** This hyperparameter controls the approximation accuracy of the t-conorm operator (Equation (8)) and thus the accuracy of differentiable d-separation scores. Setting this value too large will lose approximation accuracy, while setting it too low will induce unstable and unsmooth gradients. Thus, during training or parameter update cycles, we use a $\alpha_{\text{train}} = 0.01$, while during evaluation when computing the DAG selection score, we use a $\alpha_{\text{eval}} = 1e - 5$.

- **DAGMA's [4] log-det acyclicity constraint hyperparameter** $s$**:** This hyperparameter controls the valid region of M-matrices in which the log-det acyclicity loss is well-defined. Setting this value too large will risk model parameters stepping out of this region and causing undefined gradients, whereas setting this value too large will cause the gradient to have very small norm. In our experiments, for small graphs (include $n = 10$ synthetic binary dataset and both Sachs [28] and Lucas [14]) we use $s = 3.0$, while for large graphs ($n = 50$ synthetic binary dataset) we use $s = 8.0$.

Finally, for the DLP step size $\beta$, for each dataset we choose a different range to run hyperparameter search and choose the best sampled DAGs therein according to the DAG selection score (Appendix C.3). The specific value range of $\beta$ is chosen according to the acceptance rate in the DLP's Metropolis-Hastings acceptance-rejection step. Specifically, we choose the lowest $\beta$ value to be the one that can roughly achieve 0.8 acceptance rate, and the highest $\beta$ value to be the one that can roughly achieve 0.2 acceptance rate.

- $n = 10$ **graphs in synthetic binary dataset:** $\beta \in \{0.5, 0.6, 0.7, 0.8, 0.9, 1.0, 1.1, 1.2\}$.

- $n = 50$ **graphs in synthetic binary dataset:** $\beta \in \{0.31, 0.32, 0.33, 0.34, 0.35, 0.36, 0.37, 0.38\}$.

- **Sachs and Lucas:** $\beta \in \{0.76, 0.78, 0.80, 0.82, 0.84, 0.86, 0.88, 0.90\}$

### D.5 Baseline model details and hyperparameters

We tested our method against 5 methods from the table in the following section. For PC and kPC, we used chi-square independence test and chose significance level threshold at 0.05, and we tested with both first and second order conditional independence tests for kPC. We used the causal-learn implementation[53] for PC and GES algorithms. For GES we used the local BIC score. For NOTEARS and DAGMA linear mode, we used the l2 loss for non-linear mode, and we followed the default dimensions for the MLP layer as used in the authors' original code. Rest of parameters all followed default settings.

### D.6 Compute resources used

For the baselines, we ran PC, kPC, GES and linear version of NOTEARS, DAGMA on a 64-core AMD Epyc 7662 "Rome" processor with 16 CPU cores and 32 GB memory requested. The non-linear version of NOTEARS and DAGMA were run on one A30 with same CPU and memory requirement. Every experiment is completed in 4 hours.

For DAGPA, we run all experiments on an AMD GPU cluster, equipped with 32GB MI108 and 64GB MI210 and EPYC 7V13 cpu with 64 cores. Experiments on small graphs ($n = 10$ synthetic binary, Sachs, and Lucas) terminate within 1.5 hours, whereas experiments on large graphs ($n = 50$) terminates within 12 hours.

## E    Related Work

The methodological evolution of causal discovery has progressed through two primary paradigms—constraint-based and score-based approaches—with recent advances bridging their complementary strengths.

**Constraint-Based Methods** Pioneered by [34], constraint-based methods leverage conditional independence (CI) tests to reconstruct causal graphs. The PC algorithm [33] established core principles: (1) infer conditional independencies via statistical tests, (2) eliminate edges violating d-separation rules, and (3) orient edges using collider detection. While effective in principle, finite-sample reliability suffered from error propagation in high-order CI tests [9]. Extensions like FCI [46] addressed latent confounding through more sophisticated separation criteria, and Kernel CI Test (KCIT) [48] enabled nonparametric testing via Hilbert space embeddings. Modern variants like LOCI [42] demonstrated that low-order CI statements suffice for structure recovery under specific faithfulness conditions, while k-PC [18] provided theoretical guarantees for bounded-order testing.

**Score-Based Methods** Score-based methods reformulated causal discovery as a combinatorial optimization problem, maximizing score functions (*e.g.*, BIC [29]) over DAG spaces. These methods can be broadly divided into those that search the discrete graph space and those that leverage continuous optimization. Among discrete methods, GES [6] advanced this paradigm through greedy equivalence class search, Another major family consists of Bayesian methods, such as BayesDAG [1], which define a prior distribution over DAG structures and a likelihood function. By applying Bayes' rule, they compute a posterior distribution over DAGs, typically using sampling methods like MCMC to explore the structure space and average over models. While powerful for uncertainty quantification, these methods, like greedy search, are computationally intensive and constrained by the discrete search space. The field transformed with [51]'s NOTEARS framework, which redefined acyclicity through a differentiable constraint:

$$\mathrm{tr}(e^{W \circ W}) - d = 0,$$

where $W$ is the weighted adjacency matrix. This enabled continuous gradient-based optimization, spawning derivatives like DAGMA [4] with improved learning stability and enhanced gradient behavior via a log-determinant characterization,

$$- \log \det(s\boldsymbol{I} - \boldsymbol{W}) + d \log s = 0,$$

and GOLEM [23] using likelihood-based objectives. Nonlinear extensions [52] incorporated neural networks to model complex functional relationships while preserving acyclicity.

# F  Full Experiment Results

## F.1  Analysis of CI-MCC versus graph structure metrics

In this work we primarily showcase the conditional independence Matthews Correlation Coefficient (CI-MCC) metric, along with auxiliary graph-structure-based metrics like CPDAG Arrowhead F1, CPDAG Skeleton F1, CPDAG SHD, and DAG F1. We notice, however, that the graph-structure-based metrics may pose an unfair challenging to DAGPA. In particular, we found that the graph-structure metrics may not always align with DAGPA's objective of matching the causal DAG's predicted d-separation statements with the low-order CI statements found in the dataset. There are many cases where the DAGs returned by DAGPA made few mistakes in aligning the CI statements, yet are still scored badly by the graph-structure-based metrics. We provide concrete evidence in this section.

Here, we pick one experiment run of DAGPA on one of the synthetic binary dataset generated by an ER graph with $n = 10$ nodes with in total $N = 10000$ samples. This particular run performed 1000 sampling steps, thus giving us 1000 sampled DAGs. For each sampled DAG, we compute the DAG selection score (Appendix C.3) and the CI-MCC metric score, as well as the scores of all graph-structure-based metrics. To recall, the DAG selection score is used by DAGPA to select the best DAGs out of all that were sampled during training and requires only the input data to compute. It is *not* a test metric. Figure 8 shows the scatter plots of comparing CI-MCC to graph-structure-based metrics, Figure 7 shows the scatter plots of comparing the DAG selection score to graph-

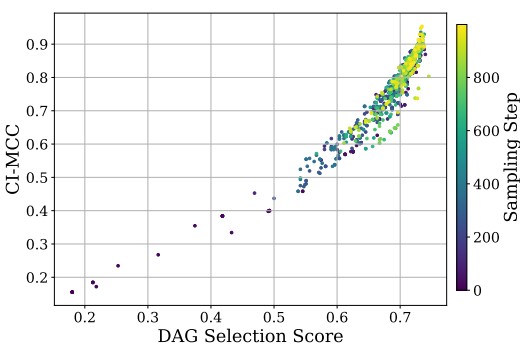

Figure 6: DAG selection score versus CI-MCC. There is a strong positive correlation.

structure-based method, and finally Figure 6 shows the relationship between the DAG selection score with CI-MCC.

From Figure 8, we can observe that, although the graph-structure-based metrics have generally positive correlation with CI-MCC, there are many samples concentrated in the bottom-right - the region where these sampled DAGs achieves high CI-MCC but low graph-structure-based metric values. Moreover, especially on regions of high CI-MCC values, the samples vary a lot on their graph-structure-based metric values. This implies that there is a misalignment between CI-MCC and the graph-structure-based metric, where performing well on matching model's d-separation with data's low-order CI statements do not translate to similarity on the graph structure.

Figure 7 reveals a similar story. In this case, the x-axis is the score actually used by DAGPA during training. The similar pattern shows that, even if DAGPA discovers a high-quality DAG with high DAG selection score, which in turn translates to making minimal false positive and false negative mistakes on the low-order CI statements (Appendix C.3), it may still have a drastically different structure than the ground-truth DAG or CPDAG, yielding a very low graph-structure-based metric value such as a low CPDAG Arrowhead F1 value. We leave the problem of proposing an alternative DAG selection score that may yield much more positive correlation with graph-structure-based metric to future research.

As a sanity check, Figure 6 shows the relationship between DAGPA's DAG selection score and the CI-MCC metric. Here, we can finally observe a strong correlation, demonstrating the validity of our proposed DAG selection score - it is correctly doing what it is designed to do, i.e., checking alignments of low-order CI statements between the model and input data.

Finally, throughout all three figures, we visualize the sampling step numbers in color scale, where the darker colors corresponds to early stages in the sampling, and lighter colors corresponds to later stages. We can observe that, in general, there exists a gradual transition from darker color to lighter color when going from bottom-left to top-right. This pattern suggests that DAGPA is indeed gradually approaching the regions in the weight space that have better and better performance, across all types of metrics.

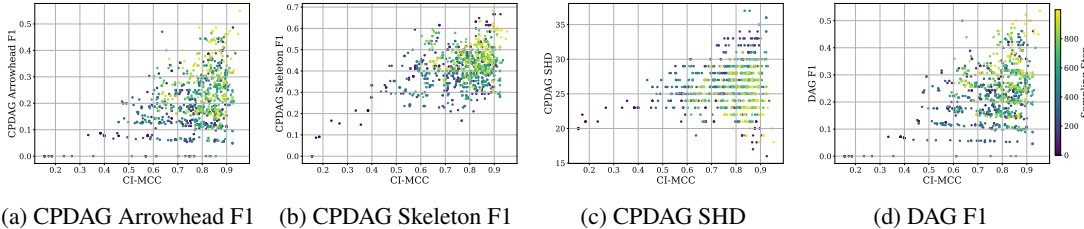

(a) CPDAG Arrowhead F1     (b) CPDAG Skeleton F1     (c) CPDAG SHD     (d) DAG F1

Figure 7: Scatter plots of CI-MCC metric versus standard graph-structure-based metrics of DAGs sampled by an exemplar DAGPA run on a $n = 10$ synthetic binary data. The graph-structure-based metrics are not strongly positively correlated with CI-MCC. Many DAGs have similarly good CI-MCC values, but vary significantly in their graph-structure-based metric values. Thus, the graph-structure-based metrics pose an unfair challenge to DAGPA that focuses on aligning low-order CI statements.

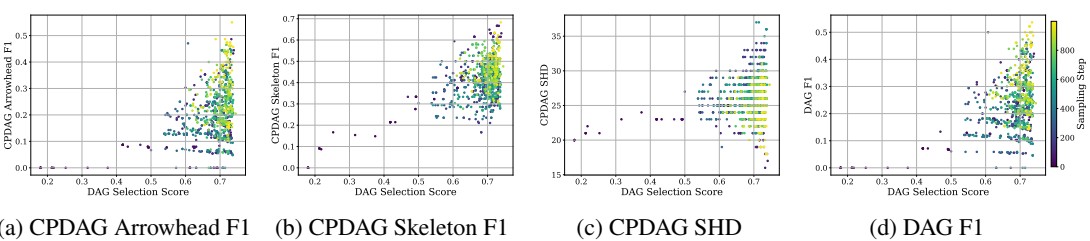

(a) CPDAG Arrowhead F1     (b) CPDAG Skeleton F1     (c) CPDAG SHD     (d) DAG F1

Figure 8: Scatter plots of DAGPA's DAG selection score (Appendix C.3) versus standard graph-structure-based metrics of DAGs sampled by an exemplar DAGPA run on a $n = 10$ synthetic binary data. Similar to the case of CI-MCC, many DAGs with similarly good DAG selection score may have vastly different values on the graph-strcture-based score.

## F.2 Results on synthetic binary data

We present the full suite of results on the synthetic binary dataset in Tables 1 to 4.

## F.3 Results on real-world data

We show the results with additional metrics on Sachs [28] dataset and on an additional real-world dataset Lucas [14] in Figure 9.

# G Future Work Discussions

## G.1 Integration with Score-Based Causal Discovery Methods

Our differentiable d-separation framework provides a novel optimization objective based on conditional independence constraints, which can potentially be integrated with existing score-based methods like NOTEARS and DAGMA as a structured regularization technique. This section briefly discusses potential strategies for integrating our approach with existing score-based methods, particularly NOTEARS [51] and DAGMA [4]. We focus on the key technical challenge: ensuring parameter compatibility between score-based and CI-constraint objectives.

The challenge is ensuring compatibility between the model parameters $\theta$ used in score-based methods and the probabilistic edge weights $W \in [0, 1]^{d \times d}$ required by our framework. For example, for NOTEARS linear models, applying sigmoid activation $\sigma(\theta)$ naturally produces valid probability measures. However, for nonlinear variants where the weighted adjacency matrix is derived from neural network parameters (e.g., $\sqrt{(\theta^{(1)})^T \theta^{(1)}}$ for the first MLP layer), additional normalization is needed to map unnormalized non-negative values to $[0, 1]$ while preserving meaningful probabilistic semantics. Once this parameterization is established, the score-based prediction loss (e.g., MSE reconstruction loss) can be added as an additional task in our multi-task optimization framework, combining functional relationship learning with explicit CI constraint enforcement. While this

## Table 1: Experimental results on Synthetic binary ER, $r = 2$

### (a) Synthetic binary ER, $r = 2$

| Method | Dataset Settings | | | | | | | |
|---|---|---|---|---|---|---|---|---|
| | d=10, n=100 | d=10, n=1k | d=10, n=10k | d=10, n=100k | d=50, n=100 | d=50, n=1k | d=50, n=10k | d=50, n=100k |
| DAGMA (Linear) [4]: | 0.2319 ±0.0351 | 0.2259 ±0.0370 | 0.2465 ±0.0502 | 0.2380 ±0.0305 | 0.0925 ±0.0198 | 0.0867 ±0.0144 | 0.0850 ±0.0264 | 0.0904 ±0.0150 |
| DAGMA (Nonlinear) [4]: | **0.5133** ±0.1653 | 0.3544 ±0.0682 | 0.5225 ±0.2650 | 0.4707 ±0.1433 | **0.2668** ±0.1312 | 0.1535 ±0.0466 | 0.1725 ±0.0652 | 0.1990 ±0.0713 |
| NOTEARS (Linear) [51]: | 0.2626 ±0.0517 | 0.2276 ±0.0333 | 0.2689 ±0.0483 | 0.2586 ±0.0363 | 0.1049 ±0.0215 | 0.0881 ±0.0211 | 0.0915 ±0.0273 | 0.0983 ±0.0178 |
| NOTEARS (Nonlinear) [52]: | 0.1949 ±0.0288 | 0.1810 ±0.0265 | 0.2023 ±0.0276 | 0.1901 ±0.0216 | 0.0758 ±0.0125 | 0.0727 ±0.0112 | 0.0718 ±0.0190 | 0.0734 ±0.0144 |
| GES [6]: | 0.2983 ±0.0899 | 0.6102 ±0.1428 | 0.8546 ±0.1076 | 0.7783 ±0.0864 | -0.1131 ±0.0794 | 0.0178 ±0.1155 | **0.1844** ±0.1689 | 0.0387 ±0.0235 |
| PC [33]: | 0.2847 ±0.0541 | 0.6227 ±0.1281 | 0.8666 ±0.1004 | 0.7922 ±0.0988 | -0.1931 ±0.0740 | 0.0081 ±0.1172 | 0.1073 ±0.1358 | 0.0325 ±0.0291 |
| kPC (k=1) [18]: | 0.3135 ±0.0706 | **0.7585** ±0.1540 | **0.8923** ±0.1001 | **0.8205** ±0.0906 | -0.1049 ±0.1208 | 0.0512 ±0.0925 | 0.0837 ±0.1375 | 0.0258 ±0.0224 |
| kPC (k=2) [18]: | 0.2941 ±0.0694 | 0.7100 ±0.1484 | 0.8667 ±0.0954 | 0.8109 ±0.0900 | -0.1679 ±0.0961 | 0.0590 ±0.0910 | 0.1139 ±0.1408 | 0.0292 ±0.0229 |
| **DAGPA (Ours)** | 0.4679 ±0.1285 | 0.6848 ±0.0817 | 0.8223 ±0.0970 | 0.7971 ±0.0849 | 0.0604 ±0.0163 | **0.2156** ±0.0555 | 0.0601 ±0.0199 | **0.3844** ±0.0974 |

### (b) Synthetic binary ER, $r = 2$, CPDAG F1 Arrowhead

| Method | Dataset Settings | | | | | | | |
|---|---|---|---|---|---|---|---|---|
| | d=10, n=100 | d=10, n=1k | d=10, n=10k | d=10, n=100k | d=50, n=100 | d=50, n=1k | d=50, n=10k | d=50, n=100k |
| DAGMA (Linear) [4]: | 0.0273 ±0.0862 | 0.0000 ±0.0000 | 0.0211 ±0.0666 | 0.0000 ±0.0000 | 0.0085 ±0.0179 | 0.0214 ±0.0348 | 0.0143 ±0.0214 | 0.0146 ±0.0241 |
| DAGMA (Nonlinear) [4]: | **0.2143** ±0.1470 | 0.2244 ±0.0624 | 0.1921 ±0.1522 | 0.3070 ±0.2069 | **0.1914** ±0.0334 | 0.1791 ±0.0781 | 0.2194 ±0.0972 | 0.2749 ±0.0581 |
| NOTEARS (Linear) [51]: | 0.0296 ±0.0655 | 0.0000 ±0.0000 | 0.0000 ±0.0000 | 0.0000 ±0.0000 | 0.0696 ±0.0705 | 0.0273 ±0.0355 | 0.0239 ±0.0437 | 0.0231 ±0.0250 |
| NOTEARS (Nonlinear) [52]: | 0.0424 ±0.0912 | 0.0000 ±0.0000 | 0.0133 ±0.0422 | 0.0000 ±0.0000 | 0.0214 ±0.0212 | 0.0387 ±0.0407 | 0.0519 ±0.0455 | 0.0102 ±0.0141 |
| GES [6]: | 0.0915 ±0.1322 | **0.3860** ±0.1894 | 0.3900 ±0.1013 | 0.4866 ±0.1802 | **0.2006** ±0.0870 | **0.5226** ±0.0646 | **0.6921** ±0.0943 | 0.7022 ±0.0897 |
| PC [33]: | 0.1025 ±0.1618 | 0.3607 ±0.2217 | **0.5673** ±0.1444 | **0.7620** ±0.1541 | 0.1242 ±0.0499 | 0.4011 ±0.0272 | 0.5938 ±0.0590 | **0.7325** ±0.0687 |
| kPC (k=1) [18]: | 0.0824 ±0.1370 | 0.1991 ±0.1077 | 0.2435 ±0.1380 | 0.3359 ±0.1664 | 0.1039 ±0.0453 | 0.2255 ±0.0575 | 0.2595 ±0.0419 | 0.2797 ±0.0435 |
| kPC (k=2) [18]: | 0.0942 ±0.1650 | 0.2298 ±0.1473 | 0.4042 ±0.1623 | 0.5329 ±0.2219 | 0.0923 ±0.0551 | 0.2788 ±0.0520 | 0.4137 ±0.0559 | 0.5561 ±0.0658 |
| **DAGPA (Ours)** | 0.1057 ±0.0766 | 0.1882 ±0.0786 | 0.2041 ±0.0912 | 0.2684 ±0.1081 | 0.0005 ±0.0034 | 0.0347 ±0.0267 | 0.0015 ±0.0071 | 0.0541 ±0.0156 |

### (c) Synthetic binary ER, $r = 2$, CPDAG F1 Skeleton

| Method | Dataset Settings | | | | | | | |
|---|---|---|---|---|---|---|---|---|
| | d=10, n=100 | d=10, n=1k | d=10, n=10k | d=10, n=100k | d=50, n=100 | d=50, n=1k | d=50, n=10k | d=50, n=100k |
| DAGMA (Linear) [4]: | 0.2921 ±0.1183 | 0.2629 ±0.1052 | 0.3034 ±0.1501 | 0.2682 ±0.1128 | 0.2953 ±0.0285 | 0.2417 ±0.0405 | 0.2431 ±0.0419 | 0.2428 ±0.0747 |
| DAGMA (Nonlinear) [4]: | **0.5418** ±0.0967 | 0.5362 ±0.0742 | 0.6114 ±0.1816 | 0.6341 ±0.1375 | 0.3667 ±0.0307 | 0.5542 ±0.0338 | 0.5701 ±0.0356 | 0.5989 ±0.0491 |
| NOTEARS (Linear) [51]: | 0.3535 ±0.1174 | 0.2791 ±0.0958 | 0.3692 ±0.1524 | 0.3252 ±0.1058 | 0.3574 ±0.0378 | 0.3135 ±0.0544 | 0.3057 ±0.0398 | 0.3036 ±0.0741 |
| NOTEARS (Nonlinear) [52]: | 0.1415 ±0.1327 | 0.0807 ±0.1433 | 0.1991 ±0.1278 | 0.0694 ±0.1124 | 0.1434 ±0.0518 | 0.1347 ±0.0482 | 0.1319 ±0.0812 | 0.1047 ±0.0501 |
| GES [6]: | 0.3845 ±0.1305 | 0.7013 ±0.0945 | 0.8038 ±0.0643 | 0.8271 ±0.0500 | **0.4303** ±0.0401 | 0.7213 ±0.0434 | 0.8588 ±0.0278 | 0.8627 ±0.0343 |
| PC [33]: | 0.4202 ±0.1054 | **0.8121** ±0.0899 | **0.9314** ±0.0420 | **0.9717** ±0.0322 | 0.4011 ±0.0272 | **0.8371** ±0.0168 | **0.8871** ±0.0184 | **0.9626** ±0.0161 |
| kPC (k=1) [18]: | 0.4270 ±0.0956 | 0.7831 ±0.0745 | 0.7793 ±0.0442 | 0.8004 ±0.0536 | 0.4278 ±0.0243 | 0.7488 ±0.0361 | 0.7860 ±0.0273 | 0.6889 ±0.0602 |
| kPC (k=2) [18]: | 0.4202 ±0.1054 | 0.8069 ±0.0806 | 0.8968 ±0.0332 | 0.8950 ±0.0670 | 0.4024 ±0.0276 | 0.7496 ±0.0411 | 0.8827 ±0.0154 | 0.9139 ±0.0292 |
| **DAGPA (Ours)** | 0.4062 ±0.0985 | 0.4273 ±0.0813 | 0.4442 ±0.0945 | 0.4452 ±0.0867 | 0.0062 ±0.0146 | 0.0826 ±0.0347 | 0.0079 ±0.0142 | 0.1043 ±0.0300 |

### (d) Synthetic binary ER, $r = 2$, CPDAG SHD

| Method | Dataset Settings | | | | | | | |
|---|---|---|---|---|---|---|---|---|
| | d=10, n=100 | d=10, n=1k | d=10, n=10k | d=10, n=100k | d=50, n=100 | d=50, n=1k | d=50, n=10k | d=50, n=100k |
| DAGMA (Linear) [4]: | 17.8000 ±1.8135 | 17.5000 ±1.0801 | 17.1000 ±2.0248 | 16.1000 ±1.5239 | 97.5000 ±1.7159 | 96.2000 ±2.6162 | 97.7000 ±1.8886 | 96.7000 ±2.4060 |
| DAGMA (Nonlinear) [4]: | 17.4000 ±2.2706 | 15.6000 ±0.9661 | 16.4000 ±2.6750 | 13.4000 ±2.4585 | 152.9000 ±15.3511 | 88.1000 ±4.9989 | 85.4000 ±6.2752 | 81.7000 ±3.4010 |
| NOTEARS (Linear) [51]: | 17.7000 ±1.5670 | 17.5000 ±1.0801 | 17.3000 ±1.8288 | 16.1000 ±1.4491 | 96.6000 ±3.6878 | 95.8000 ±2.8983 | 96.6000 ±2.9136 | 95.9000 ±2.2828 |
| NOTEARS (Nonlinear) [52]: | 18.1000 ±2.0248 | 18.1000 ±1.2867 | 18.3000 ±1.7029 | 17.2000 ±1.6193 | 99.8000 ±1.1353 | 98.0000 ±2.3570 | 97.9000 ±2.5144 | 98.6000 ±1.5055 |
| GES [6]: | 17.5000 ±2.0683 | 13.7000 ±3.9172 | 16.1000 ±2.8848 | 13.9000 ±5.8963 | 93.1000 ±5.7436 | 61.4000 ±6.2752 | 43.2000 ±11.2921 | 48.8000 ±13.9507 |
| PC [33]: | 17.5000 ±2.5495 | 12.8000 ±4.3153 | 9.6000 ±1.8974 | 4.9000 ±2.8848 | 95.0000 ±3.3333 | 70.1000 ±4.7011 | 49.1000 ±6.3675 | 31.0000 ±8.8066 |
| kPC (k=1) [18]: | 17.8000 ±2.1499 | 14.4000 ±2.8363 | 19.5000 ±4.2230 | 16.9000 ±4.1486 | 94.9000 ±4.1486 | 77.4000 ±5.3166 | 88.0000 ±8.2999 | 134.8000 ±28.9858 |
| kPC (k=2) [18]: | 17.4000 ±2.6331 | 12.5000 ±3.2059 | 12.0000 ±2.3094 | 10.9000 ±4.7947 | 94.5000 ±3.6286 | 69.2000 ±3.9101 | 55.5000 ±5.4416 | 50.3000 ±11.3534 |
| **DAGPA (Ours)** | 21.4800 ±2.6972 | 23.8200 ±2.2740 | 25.5600 ±3.3022 | 24.1400 ±2.8357 | 103.6200 ±10.4664 | 180.8400 ±8.3626 | 106.2000 ±15.9796 | 220.7800 ±21.7596 |

### (e) Synthetic binary ER, $r = 2$, DAG F1

| Method | Dataset Settings | | | | | | | |
|---|---|---|---|---|---|---|---|---|
| | d=10, n=100 | d=10, n=1k | d=10, n=10k | d=10, n=100k | d=50, n=100 | d=50, n=1k | d=50, n=10k | d=50, n=100k |
| DAGMA (Linear) [4]: | 0.1706 ±0.1091 | 0.1225 ±0.0616 | 0.2137 ±0.1476 | 0.1677 ±0.1313 | 0.2020 ±0.0328 | 0.1682 ±0.0327 | 0.1766 ±0.0490 | 0.1791 ±0.0485 |
| DAGMA (Nonlinear) [4]: | 0.3243 ±0.1575 | 0.2734 ±0.0819 | 0.3190 ±0.1602 | 0.4432 ±0.1806 | 0.2243 ±0.0269 | 0.3087 ±0.0649 | 0.3452 ±0.0670 | 0.3829 ±0.0546 |
| NOTEARS (Linear) [51]: | 0.2076 ±0.1263 | 0.1192 ±0.0676 | 0.2304 ±0.1460 | 0.1905 ±0.1282 | 0.2304 ±0.0583 | 0.1878 ±0.0441 | 0.2027 ±0.0489 | 0.2091 ±0.0354 |
| NOTEARS (Nonlinear) [52]: | 0.0688 ±0.0805 | 0.0179 ±0.0378 | 0.0564 ±0.0778 | 0.0327 ±0.0743 | 0.0583 ±0.0178 | 0.0665 ±0.0391 | 0.0725 ±0.0426 | 0.0507 ±0.0273 |
| **DAGPA (Ours)** | 0.1557 ±0.1036 | 0.2333 ±0.1026 | 0.2176 ±0.0971 | 0.2603 ±0.1145 | 0.0088 ±0.0117 | 0.0372 ±0.0230 | 0.0091 ±0.0143 | 0.0559 ±0.0163 |

## Table 2: Experimental results on Synthetic binary ER, $r = 4$

### (a) Synthetic binary ER, $r = 4$, CI-MC

| Method | Dataset Settings | | | | | | | |
|---|---|---|---|---|---|---|---|---|
| | d=10, n=100 | d=10, n=1k | d=10, n=10k | d=10, n=100k | d=50, n=100 | d=50, n=1k | d=50, n=10k | d=50, n=100k |
| DAGMA (Linear) [4]: | 0.2398 ± 0.0631 | 0.2085 ± 0.0347 | 0.2120 ± 0.0400 | 0.2226 ± 0.0584 | 0.0428 ± 0.0064 | 0.0398 ± 0.0039 | 0.0398 ± 0.0039 | 0.0395 ± 0.0051 |
| DAGMA (Nonlinear) [4]: | 0.5724 ± 0.1213 | 0.4450 ± 0.1860 | 0.5007 ± 0.1535 | 0.4624 ± 0.1775 | **0.2658** ± 0.0818 | 0.0591 ± 0.0169 | 0.0735 ± 0.0181 | 0.0623 ± 0.0132 |
| NOTEARS (Linear) [51]: | 0.2798 ± 0.0848 | 0.2197 ± 0.0391 | 0.2344 ± 0.0665 | 0.2294 ± 0.0273 | 0.0506 ± 0.0081 | 0.0419 ± 0.0049 | 0.0434 ± 0.0046 | 0.0423 ± 0.0057 |
| NOTEARS (Nonlinear) [52]: | 0.1882 ± 0.0371 | 0.1804 ± 0.0290 | 0.2091 ± 0.0469 | 0.1894 ± 0.0278 | 0.0369 ± 0.0045 | 0.0350 ± 0.0026 | 0.0342 ± 0.0040 | 0.0361 ± 0.0063 |
| GES [6]: | 0.3341 ± 0.1212 | 0.5932 ± 0.1362 | 0.9135 ± 0.1097 | 0.9251 ± 0.0568 | -0.3803 ± 0.1331 | -0.1514 ± 0.0513 | -0.0267 ± 0.0469 | 0.0000 ± 0.0000 |
| PC [33]: | 0.3282 ± 0.0842 | 0.7071 ± 0.1704 | 0.8897 ± 0.1385 | 0.9317 ± 0.0444 | -0.3514 ± 0.0755 | -0.0181 ± 0.0608 | -0.0049 ± 0.0138 | -0.0100 ± 0.0155 |
| kPC (k=1) [18]: | 0.4053 ± 0.1859 | **0.8818** ± 0.1073 | 0.9469 ± 0.0775 | 0.9372 ± 0.0443 | -0.1670 ± 0.0624 | -0.0027 ± 0.0514 | 0.0053 ± 0.0147 | 0.0000 ± 0.0000 |
| kPC (k=2) [18]: | 0.3307 ± 0.0878 | 0.8546 ± 0.1099 | 0.9469 ± 0.0775 | 0.9378 ± 0.0437 | -0.2243 ± 0.0834 | -0.0036 ± 0.0517 | 0.0054 ± 0.0153 | 0.0000 ± 0.0000 |
| **DAGPA (Ours)** | **0.6214** ± 0.1562 | 0.8299 ± 0.1328 | **0.9493** ± 0.0537 | **0.9484** ± 0.0357 | 0.2341 ± 0.0501 | **0.3292** ± 0.0720 | **0.2669** ± 0.1632 | **0.7209** ± 0.1333 |

### (b) Synthetic binary ER, $r = 4$, CPDAG F1 Arrowhead

| Method | Dataset Settings | | | | | | | |
|---|---|---|---|---|---|---|---|---|
| | d=10, n=100 | d=10, n=1k | d=10, n=10k | d=10, n=100k | d=50, n=100 | d=50, n=1k | d=50, n=10k | d=50, n=100k |
| DAGMA (Linear) [4]: | 0.0160 ± 0.0506 | 0.0000 ± 0.0000 | 0.0333 ± 0.1054 | 0.0000 ± 0.0000 | 0.0041 ± 0.0087 | 0.0051 ± 0.0086 | 0.0050 ± 0.0108 | 0.0040 ± 0.0085 |
| DAGMA (Nonlinear) [4]: | **0.1844** ± 0.1478 | 0.1668 ± 0.1707 | 0.0857 ± 0.1173 | 0.1994 ± 0.1346 | **0.1193** ± 0.0338 | 0.0643 ± 0.0368 | 0.1158 ± 0.0564 | 0.0976 ± 0.0409 |
| NOTEARS (Linear) [51]: | 0.0490 ± 0.0871 | 0.0000 ± 0.0000 | 0.0333 ± 0.1054 | 0.0000 ± 0.0000 | 0.0190 ± 0.0232 | 0.0061 ± 0.0109 | 0.0130 ± 0.0182 | 0.0020 ± 0.0063 |
| NOTEARS (Nonlinear) [52]: | 0.0000 ± 0.0000 | 0.0000 ± 0.0000 | 0.0847 ± 0.1208 | 0.0314 ± 0.0529 | 0.0109 ± 0.0145 | 0.0010 ± 0.0032 | 0.0101 ± 0.0157 | 0.0070 ± 0.0125 |
| GES [6]: | 0.0685 ± 0.0793 | 0.2782 ± 0.1766 | 0.3504 ± 0.2241 | **0.4993** ± 0.1762 | 0.0684 ± 0.0327 | **0.2896** ± 0.0465 | **0.5302** ± 0.0599 | 0.1253 ± 0.2657 |
| PC [33]: | 0.1049 ± 0.0729 | **0.3096** ± 0.1615 | **0.4223** ± 0.1233 | 0.4759 ± 0.1242 | **0.0819** ± 0.0301 | 0.2572 ± 0.0454 | 0.3632 ± 0.0439 | **0.4153** ± 0.0472 |
| kPC (k=1) [18]: | 0.1014 ± 0.0858 | 0.1965 ± 0.1176 | 0.2402 ± 0.1230 | 0.2588 ± 0.1050 | 0.0377 ± 0.0116 | 0.0262 ± 0.0136 | 0.0262 ± 0.0136 | 0.0200 ± 0.0115 |
| kPC (k=2) [18]: | 0.1034 ± 0.0874 | 0.1984 ± 0.1402 | 0.3077 ± 0.1590 | 0.3455 ± 0.1079 | 0.0385 ± 0.0132 | 0.0417 ± 0.0287 | 0.0532 ± 0.0236 | 0.0684 ± 0.0296 |
| **DAGPA (Ours)** | 0.1392 ± 0.0984 | 0.1788 ± 0.0976 | 0.2733 ± 0.0832 | 0.2589 ± 0.1219 | 0.0424 ± 0.0204 | 0.0572 ± 0.0228 | 0.0392 ± 0.0232 | 0.0648 ± 0.0131 |

### (c) Synthetic binary ER, $r = 4$, CPDAG F1 Skeleton

| Method | Dataset Settings | | | | | | | |
|---|---|---|---|---|---|---|---|---|
| | d=10, n=100 | d=10, n=1k | d=10, n=10k | d=10, n=100k | d=50, n=100 | d=50, n=1k | d=50, n=10k | d=50, n=100k |
| DAGMA (Linear) [4]: | 0.2706 ± 0.1085 | 0.2260 ± 0.0935 | 0.2053 ± 0.0892 | 0.2292 ± 0.0902 | 0.0982 ± 0.0344 | 0.0732 ± 0.0204 | 0.0952 ± 0.0135 | 0.0731 ± 0.0159 |
| DAGMA (Nonlinear) [4]: | **0.5460** ± 0.1325 | 0.5399 ± 0.0802 | 0.5137 ± 0.1327 | 0.5488 ± 0.1238 | 0.2296 ± 0.0323 | 0.1997 ± 0.0491 | 0.2653 ± 0.0517 | 0.2288 ± 0.0490 |
| NOTEARS (Linear) [51]: | 0.3598 ± 0.1345 | 0.2533 ± 0.0884 | 0.2489 ± 0.1027 | 0.2722 ± 0.0854 | 0.1388 ± 0.0521 | 0.0915 ± 0.0266 | 0.1193 ± 0.0138 | 0.0960 ± 0.0197 |
| NOTEARS (Nonlinear) [52]: | 0.1435 ± 0.1318 | 0.0848 ± 0.0995 | 0.1779 ± 0.1279 | 0.1492 ± 0.1253 | 0.0519 ± 0.0353 | 0.0350 ± 0.0258 | 0.0408 ± 0.0281 | 0.0352 ± 0.0154 |
| GES [6]: | 0.3903 ± 0.0991 | 0.6616 ± 0.0774 | 0.8165 ± 0.0861 | 0.7991 ± 0.0475 | 0.1820 ± 0.0589 | 0.4372 ± 0.0445 | 0.7155 ± 0.0350 | 0.1574 ± 0.3320 |
| PC [33]: | 0.4194 ± 0.0577 | 0.7178 ± 0.1084 | **0.8209** ± 0.0830 | **0.8080** ± 0.0550 | 0.2193 ± 0.0340 | 0.5469 ± 0.0404 | **0.7493** ± 0.0295 | **0.7816** ± 0.0204 |
| kPC (k=1) [18]: | 0.4421 ± 0.0710 | **0.7346** ± 0.1041 | 0.7659 ± 0.0614 | 0.7724 ± 0.0304 | **0.2317** ± 0.0301 | **0.5565** ± 0.0406 | 0.6894 ± 0.0297 | 0.6425 ± 0.0286 |
| kPC (k=2) [18]: | 0.4194 ± 0.0577 | 0.7178 ± 0.1084 | 0.8137 ± 0.0830 | 0.7958 ± 0.0502 | 0.2198 ± 0.0333 | 0.5530 ± 0.0403 | 0.7416 ± 0.0310 | 0.7458 ± 0.0208 |
| **DAGPA (Ours)** | 0.4217 ± 0.1182 | 0.4490 ± 0.0858 | 0.5309 ± 0.0943 | 0.5036 ± 0.0678 | 0.1057 ± 0.0277 | 0.1254 ± 0.0348 | 0.0837 ± 0.0329 | 0.1449 ± 0.0270 |

### (d) Synthetic binary ER, $r = 4$, CPDAG SHD

| Method | Dataset Settings | | | | | | | |
|---|---|---|---|---|---|---|---|---|
| | d=10, n=100 | d=10, n=1k | d=10, n=10k | d=10, n=100k | d=50, n=100 | d=50, n=1k | d=50, n=10k | d=50, n=100k |
| DAGMA (Linear) [4]: | 21.1000 ± 2.5144 | 20.4000 ± 3.9497 | 22.0000 ± 3.3665 | 21.2000 ± 3.3267 | 197.7000 ± 2.4518 | 197.7000 ± 1.6364 | 196.9000 ± 2.1318 | 197.9000 ± 1.2867 |
| DAGMA (Nonlinear) [4]: | 21.1000 ± 2.9231 | 18.3000 ± 3.8887 | 21.0000 ± 3.8006 | 19.0000 ± 3.2660 | 228.8000 ± 6.9889 | 191.7000 ± 5.8128 | 185.7000 ± 5.3552 | 188.4000 ± 4.0879 |
| NOTEARS (Linear) [51]: | 19.9000 ± 3.2128 | 20.4000 ± 3.9777 | 21.6000 ± 3.7476 | 21.1000 ± 3.3483 | 198.2000 ± 3.0111 | 197.6000 ± 2.1187 | 196.1000 ± 1.5239 | 197.9000 ± 1.2867 |
| NOTEARS (Nonlinear) [52]: | 21.9000 ± 2.3781 | 21.3000 ± 4.1379 | 22.1000 ± 3.4140 | 21.6000 ± 2.8752 | 198.5000 ± 1.7795 | 199.1000 ± 0.8756 | 198.6000 ± 1.8974 | 198.9000 ± 1.1972 |
| GES [6]: | 21.2000 ± 2.6998 | 18.3000 ± 5.0122 | 19.6000 ± 6.3281 | 18.4000 ± 7.2602 | 196.3000 ± 4.3218 | 164.5000 ± 7.7782 | 130.2000 ± 12.6474 | 24.6000 ± 52.7977 |
| PC [33]: | 20.7000 ± 2.7909 | 17.9000 ± 5.0870 | 17.3000 ± 4.9227 | 17.4000 ± 4.7656 | 205.0000 ± 5.8119 | 177.3000 ± 9.0437 | 164.4000 ± 8.8343 | 171.1000 ± 14.3717 |
| kPC (k=1) [18]: | 21.1000 ± 2.6854 | 17.7000 ± 5.1865 | 22.3000 ± 4.4234 | 22.4000 ± 5.1683 | 205.3000 ± 4.2701 | 184.5000 ± 7.7782 | 211.6000 ± 9.7091 | 282.7000 ± 26.8620 |
| kPC (k=2) [18]: | 20.5000 ± 2.8382 | 17.6000 ± 5.4610 | 18.0000 ± 5.9815 | 18.3000 ± 3.4657 | 204.4000 ± 4.4771 | 177.3000 ± 8.1384 | 173.2000 ± 8.4564 | 198.9000 ± 14.1142 |
| **DAGPA (Ours)** | 26.6400 ± 3.6854 | 27.2800 ± 4.2859 | 27.5600 ± 3.1826 | 28.2400 ± 3.6397 | 258.5600 ± 4.5273 | 268.3000 ± 10.5313 | 254.2000 ± 18.9737 | 307.2400 ± 24.4361 |

### (e) Synthetic binary ER, $r = 4$, DAG F1

| Method | Dataset Settings | | | | | | | |
|---|---|---|---|---|---|---|---|---|
| | d=10, n=100 | d=10, n=1k | d=10, n=10k | d=10, n=100k | d=50, n=100 | d=50, n=1k | d=50, n=10k | d=50, n=100k |
| DAGMA (Linear) [4]: | 0.1661 ± 0.1206 | 0.1406 ± 0.1301 | 0.1428 ± 0.0775 | 0.1365 ± 0.1200 | 0.0671 ± 0.0271 | 0.0569 ± 0.0181 | 0.0618 ± 0.0236 | 0.0452 ± 0.0191 |
| DAGMA (Nonlinear) [4]: | 0.2813 ± 0.1782 | 0.3242 ± 0.1300 | 0.2824 ± 0.1015 | 0.3234 ± 0.1329 | 0.1476 ± 0.0287 | 0.1280 ± 0.0440 | 0.1739 ± 0.0548 | 0.1441 ± 0.0466 |
| NOTEARS (Linear) [51]: | 0.2563 ± 0.0874 | 0.1505 ± 0.1279 | 0.1517 ± 0.0926 | 0.1492 ± 0.1050 | 0.0903 ± 0.0424 | 0.0668 ± 0.0261 | 0.0761 ± 0.0162 | 0.0512 ± 0.0224 |
| NOTEARS (Nonlinear) [52]: | 0.0772 ± 0.0767 | 0.0238 ± 0.0385 | 0.1082 ± 0.1006 | 0.0730 ± 0.0905 | 0.0279 ± 0.0208 | 0.0079 ± 0.0078 | 0.0195 ± 0.0156 | 0.0196 ± 0.0103 |
| **DAGPA (Ours)** | 0.2017 ± 0.1142 | 0.2056 ± 0.0526 | 0.2874 ± 0.0891 | 0.3427 ± 0.1043 | 0.0483 ± 0.0176 | 0.0608 ± 0.0245 | 0.0433 ± 0.0207 | 0.0681 ± 0.0153 |

## Table 3: Experimental results on Synthetic binary SF, $r = 2$

### (a) Synthetic binary SF, $r = 2$, CI-MC

| Method | Dataset Settings | | | | | | | |
|---|---|---|---|---|---|---|---|---|
| | d=10, n=100 | d=10, n=1k | d=10, n=10k | d=10, n=100k | d=50, n=100 | d=50, n=1k | d=50, n=10k | d=50, n=100k |
| DAGMA (Linear) [4]: | 0.2279 ± 0.0372 | 0.2247 ± 0.0472 | 0.2256 ± 0.0285 | 0.2167 ± 0.0328 | 0.1413 ± 0.0206 | 0.1483 ± 0.0215 | 0.1475 ± 0.0186 | 0.1560 ± 0.0178 |
| DAGMA (Nonlinear) [4]: | 0.4979 ± 0.1539 | 0.3630 ± 0.0893 | 0.4339 ± 0.1296 | 0.4195 ± 0.1210 | 0.1170 ± 0.0571 | **0.2208** ± 0.0521 | **0.2464** ± 0.0610 | **0.2867** ± 0.0398 |
| NOTEARS (Linear) [51]: | 0.2716 ± 0.0877 | 0.2422 ± 0.0590 | 0.2548 ± 0.0466 | 0.2556 ± 0.0563 | **0.1433** ± 0.0233 | 0.1551 ± 0.0168 | 0.1614 ± 0.0249 | 0.1659 ± 0.0188 |
| NOTEARS (Nonlinear) [52]: | 0.2018 ± 0.0362 | 0.1919 ± 0.0342 | 0.1960 ± 0.0298 | 0.1847 ± 0.0270 | 0.1263 ± 0.0190 | 0.1286 ± 0.0176 | 0.1263 ± 0.0161 | 0.1341 ± 0.0149 |
| GES [6]: | 0.3288 ± 0.1263 | 0.6257 ± 0.1922 | 0.7965 ± 0.1152 | 0.8394 ± 0.1061 | 0.0031 ± 0.0431 | 0.1955 ± 0.0926 | 0.2143 ± 0.1380 | 0.1430 ± 0.1201 |
| PC [33]: | 0.3359 ± 0.1207 | 0.7139 ± 0.1368 | 0.7956 ± 0.1502 | 0.8635 ± 0.0748 | -0.0206 ± 0.0445 | 0.1634 ± 0.0986 | 0.0462 ± 0.0724 | 0.0537 ± 0.0733 |
| kPC (k=1) [18]: | 0.3899 ± 0.1479 | **0.8032** ± 0.1781 | 0.8170 ± 0.1356 | **0.8687** ± 0.0740 | -0.0009 ± 0.0737 | 0.1036 ± 0.1189 | 0.0579 ± 0.0835 | 0.0392 ± 0.0693 |
| kPC (k=2) [18]: | 0.3435 ± 0.1270 | 0.7805 ± 0.1662 | 0.8199 ± 0.1342 | 0.8681 ± 0.0732 | -0.0261 ± 0.0723 | 0.1487 ± 0.1267 | 0.0495 ± 0.0990 | 0.0356 ± 0.0518 |
| **DAGPA (Ours)** | **0.5491** ± 0.1461 | 0.7731 ± 0.1272 | **0.8684** ± 0.1128 | 0.8659 ± 0.0624 | 0.0728 ± 0.0278 | 0.0906 ± 0.0262 | 0.1057 ± 0.0361 | 0.1938 ± 0.0784 |

### (b) Synthetic binary SF, $r = 2$, CPDAG F1 Arrowhead

| Method | Dataset Settings | | | | | | | |
|---|---|---|---|---|---|---|---|---|
| | d=10, n=100 | d=10, n=1k | d=10, n=10k | d=10, n=100k | d=50, n=100 | d=50, n=1k | d=50, n=10k | d=50, n=100k |
| DAGMA (Linear) [4]: | 0.0000 ± 0.0000 | 0.0125 ± 0.0395 | 0.0000 ± 0.0000 | 0.0426 ± 0.0904 | 0.0087 ± 0.0183 | 0.0000 ± 0.0000 | 0.0044 ± 0.0139 | 0.0084 ± 0.0178 |
| DAGMA (Nonlinear) [4]: | **0.1950** ± 0.1364 | 0.2072 ± 0.1209 | 0.2012 ± 0.1235 | 0.1725 ± 0.1522 | **0.1465** ± 0.0501 | 0.1891 ± 0.0760 | 0.2219 ± 0.0531 | 0.2210 ± 0.0975 |
| NOTEARS (Linear) [51]: | 0.0495 ± 0.1267 | 0.0211 ± 0.0666 | 0.0000 ± 0.0000 | 0.0000 ± 0.0000 | 0.0273 ± 0.0312 | 0.0148 ± 0.0248 | 0.0132 ± 0.0212 | 0.0149 ± 0.0202 |
| NOTEARS (Nonlinear) [52]: | 0.0451 ± 0.0832 | 0.0000 ± 0.0000 | 0.0000 ± 0.0000 | 0.0190 ± 0.0602 | 0.0284 ± 0.0367 | 0.0210 ± 0.0262 | 0.0106 ± 0.0150 | 0.0167 ± 0.0214 |
| GES [6]: | 0.0686 ± 0.0922 | 0.3097 ± 0.1616 | **0.5025** ± 0.1347 | **0.5590** ± 0.2118 | 0.0801 ± 0.0622 | **0.4695** ± 0.0836 | **0.7163** ± 0.0728 | **0.7741** ± 0.0545 |
| PC [33]: | 0.1330 ± 0.1174 | **0.3398** ± 0.0895 | 0.4687 ± 0.1326 | 0.4592 ± 0.1445 | 0.1093 ± 0.0673 | 0.3793 ± 0.0854 | 0.4883 ± 0.0582 | 0.5647 ± 0.1108 |
| kPC (k=1) [18]: | 0.1301 ± 0.0959 | 0.1593 ± 0.1036 | 0.2905 ± 0.1309 | 0.2411 ± 0.1627 | 0.0757 ± 0.0320 | 0.1542 ± 0.0634 | 0.2068 ± 0.0310 | 0.2871 ± 0.0657 |
| kPC (k=2) [18]: | 0.1131 ± 0.0897 | 0.1612 ± 0.1402 | 0.3378 ± 0.1372 | 0.2496 ± 0.1216 | 0.0729 ± 0.0341 | 0.1670 ± 0.0556 | 0.2671 ± 0.0434 | 0.4013 ± 0.0787 |
| **DAGPA (Ours)** | 0.1363 ± 0.0942 | 0.1919 ± 0.0815 | 0.2266 ± 0.1236 | 0.2374 ± 0.1035 | 0.0037 ± 0.0076 | 0.0186 ± 0.0185 | 0.0078 ± 0.0100 | 0.0387 ± 0.0178 |

### (c) Synthetic binary SF, $r = 2$, CPDAG F1 Skeleton

| Method | Dataset Settings | | | | | | | |
|---|---|---|---|---|---|---|---|---|
| | d=10, n=100 | d=10, n=1k | d=10, n=10k | d=10, n=100k | d=50, n=100 | d=50, n=1k | d=50, n=10k | d=50, n=100k |
| DAGMA (Linear) [4]: | 0.2159 ± 0.1017 | 0.1933 ± 0.0984 | 0.2072 ± 0.1170 | 0.2296 ± 0.1284 | 0.2210 ± 0.0413 | 0.2187 ± 0.0249 | 0.2148 ± 0.0467 | 0.1912 ± 0.0305 |
| DAGMA (Nonlinear) [4]: | 0.5808 ± 0.0988 | 0.4768 ± 0.1131 | 0.5342 ± 0.1081 | 0.5834 ± 0.1504 | 0.3076 ± 0.0528 | 0.5308 ± 0.0508 | 0.5412 ± 0.0557 | 0.5433 ± 0.0681 |
| NOTEARS (Linear) [51]: | 0.2836 ± 0.1490 | 0.2358 ± 0.1330 | 0.2683 ± 0.1199 | 0.3141 ± 0.1574 | 0.2857 ± 0.0257 | 0.2748 ± 0.0333 | 0.2745 ± 0.0516 | 0.2630 ± 0.0405 |
| NOTEARS (Nonlinear) [52]: | 0.1153 ± 0.1130 | 0.0641 ± 0.0741 | 0.0815 ± 0.0977 | 0.0776 ± 0.1300 | 0.1159 ± 0.0571 | 0.1063 ± 0.0485 | 0.0860 ± 0.0648 | 0.0817 ± 0.0381 |
| GES [6]: | 0.3601 ± 0.1194 | 0.6713 ± 0.0849 | 0.8113 ± 0.0427 | 0.8196 ± 0.0875 | 0.2684 ± 0.1460 | 0.6875 ± 0.0449 | 0.8637 ± 0.0173 | 0.8839 ± 0.0322 |
| PC [33]: | 0.4474 ± 0.0868 | 0.7446 ± 0.0904 | 0.8168 ± 0.0510 | 0.8004 ± 0.0827 | 0.2942 ± 0.1534 | 0.6830 ± 0.0399 | 0.8277 ± 0.0306 | 0.8505 ± 0.0269 |
| kPC (k=1) [18]: | 0.4685 ± 0.0839 | 0.7454 ± 0.0918 | 0.7774 ± 0.0494 | 0.7737 ± 0.0593 | 0.3689 ± 0.0323 | 0.6872 ± 0.0304 | 0.7782 ± 0.0238 | 0.7501 ± 0.0268 |
| kPC (k=2) [18]: | 0.4474 ± 0.0788 | 0.7404 ± 0.0882 | 0.8120 ± 0.0508 | 0.7894 ± 0.0759 | 0.3612 ± 0.0344 | 0.7025 ± 0.0374 | 0.8231 ± 0.0281 | 0.8361 ± 0.0281 |
| **DAGPA (Ours)** | 0.4003 ± 0.0897 | 0.4746 ± 0.0788 | 0.4839 ± 0.1013 | 0.4846 ± 0.0975 | 0.0282 ± 0.0194 | 0.0389 ± 0.0183 | 0.0357 ± 0.0277 | 0.0851 ± 0.0264 |

### (d) Synthetic binary SF, $r = 2$, CPDAG SHD

| Method | Dataset Settings | | | | | | | |
|---|---|---|---|---|---|---|---|---|
| | d=10, n=100 | d=10, n=1k | d=10, n=10k | d=10, n=100k | d=50, n=100 | d=50, n=1k | d=50, n=10k | d=50, n=100k |
| DAGMA (Linear) [4]: | 19.1000 ± 1.1972 | 19.2000 ± 0.6325 | 19.2000 ± 1.0328 | 18.7000 ± 1.4181 | 97.2000 ± 1.3166 | 96.5000 ± 1.2693 | 95.9000 ± 2.2336 | 96.7000 ± 1.1595 |
| DAGMA (Nonlinear) [4]: | 18.2000 ± 2.0976 | 17.2000 ± 1.1353 | 17.1000 ± 1.9692 | 16.9000 ± 2.1833 | 147.7000 ± 15.9865 | 86.1000 ± 4.8408 | 83.6000 ± 5.1251 | 85.2000 ± 6.9889 |
| NOTEARS (Linear) [51]: | 18.6000 ± 1.7764 | 19.2000 ± 0.6325 | 18.9000 ± 0.9944 | 18.7000 ± 0.8233 | 98.9000 ± 1.6633 | 95.2000 ± 1.9889 | 94.7000 ± 2.9458 | 95.9000 ± 1.6633 |
| NOTEARS (Nonlinear) [52]: | 19.3000 ± 0.8233 | 19.8000 ± 0.4216 | 19.8000 ± 0.4216 | 19.6000 ± 0.6992 | 98.8000 ± 2.1499 | 98.6000 ± 2.4129 | 99.0000 ± 0.8165 | 98.5000 ± 1.4337 |
| GES [6]: | 18.8000 ± 1.6193 | 15.9000 ± 2.7264 | 13.8000 ± 3.2249 | 14.0000 ± 6.0736 | 79.4000 ± 41.9952 | 66.3000 ± 7.2885 | 40.5000 ± 8.3832 | 36.7000 ± 8.9821 |
| PC [33]: | 18.3000 ± 1.9465 | 15.7000 ± 1.3375 | 14.2000 ± 3.1552 | 15.6000 ± 3.4383 | 84.7000 ± 44.7314 | 82.8000 ± 9.8070 | 71.8000 ± 10.0421 | 65.1000 ± 14.4795 |
| kPC (k=1) [18]: | 18.0000 ± 1.8257 | 16.6000 ± 1.7127 | 17.7000 ± 5.3759 | 20.1000 ± 3.0714 | 120.0000 ± 26.6041 | 88.6000 ± 9.2999 | 90.5000 ± 7.7496 | 106.1000 ± 10.4823 |
| kPC (k=2) [18]: | 18.1000 ± 1.7288 | 16.1000 ± 1.5951 | 15.3000 ± 4.2701 | 17.5000 ± 3.3082 | 118.0000 ± 26.5330 | 83.1000 ± 8.5434 | 75.7000 ± 7.2732 | 74.3000 ± 12.7632 |
| **DAGPA (Ours)** | 24.7800 ± 2.4436 | 25.7400 ± 2.4562 | 25.6400 ± 3.6576 | 25.8600 ± 3.3382 | 119.7500 ± 5.5366 | 124.3200 ± 6.9765 | 130.7000 ± 4.4366 | 192.3400 ± 31.3106 |

### (e) Synthetic binary SF, $r = 2$, DAG F1

| Method | Dataset Settings | | | | | | | |
|---|---|---|---|---|---|---|---|---|
| | d=10, n=100 | d=10, n=1k | d=10, n=10k | d=10, n=100k | d=50, n=100 | d=50, n=1k | d=50, n=10k | d=50, n=100k |
| DAGMA (Linear) [4]: | 0.1383 ± 0.0889 | 0.0978 ± 0.0642 | 0.1394 ± 0.0736 | 0.1697 ± 0.1112 | 0.1275 ± 0.0364 | 0.1529 ± 0.0301 | 0.1451 ± 0.0566 | 0.1372 ± 0.0341 |
| DAGMA (Nonlinear) [4]: | 0.3603 ± 0.1203 | 0.2724 ± 0.0810 | 0.3587 ± 0.1040 | 0.3766 ± 0.1507 | 0.1845 ± 0.0362 | 0.3204 ± 0.0741 | 0.3446 ± 0.0670 | 0.3594 ± 0.0663 |
| NOTEARS (Linear) [51]: | 0.2089 ± 0.1474 | 0.1266 ± 0.1081 | 0.1872 ± 0.0802 | 0.2016 ± 0.1147 | 0.1648 ± 0.0436 | 0.1801 ± 0.0434 | 0.1849 ± 0.0516 | 0.1733 ± 0.0365 |
| NOTEARS (Nonlinear) [52]: | 0.0788 ± 0.1022 | 0.0281 ± 0.0453 | 0.0451 ± 0.0628 | 0.0265 ± 0.0605 | 0.0402 ± 0.0375 | 0.0450 ± 0.0233 | 0.0281 ± 0.0306 | 0.0438 ± 0.0268 |
| **DAGPA (Ours)** | 0.1689 ± 0.0987 | 0.2409 ± 0.0858 | 0.2662 ± 0.1244 | 0.2682 ± 0.0969 | 0.0061 ± 0.0080 | 0.0205 ± 0.0187 | 0.0163 ± 0.0195 | 0.0410 ± 0.0116 |

## Table 4: Experimental results on Synthetic binary SF, $r = 4$

### (a) Synthetic binary SF, $r = 4$, CI-MC

| Method | Dataset Settings | | | | | | | |
|---|---|---|---|---|---|---|---|---|
| | $d=10, n=100$ | $d=10, n=1k$ | $d=10, n=10k$ | $d=10, n=100k$ | $d=50, n=100$ | $d=50, n=1k$ | $d=50, n=10k$ | $d=50, n=100k$ |
| DAGMA (Linear) [4]: | 0.2029 ±0.0485 | 0.1665 ±0.0130 | 0.1764 ±0.0243 | 0.1886 ±0.0233 | 0.0527 ±0.0119 | 0.0516 ±0.0093 | 0.0454 ±0.0075 | 0.0543 ±0.0063 |
| DAGMA (Nonlinear) [4]: | 0.4897 ±0.1854 | 0.3381 ±0.1350 | 0.3735 ±0.1415 | 0.3296 ±0.0864 | **0.2058** ±0.0892 | 0.0811 ±0.0204 | 0.0708 ±0.0374 | 0.0781 ±0.0315 |
| NOTEARS (Linear) [51]: | 0.2459 ±0.0949 | 0.1940 ±0.0335 | 0.1831 ±0.0328 | 0.2024 ±0.0238 | 0.0613 ±0.0173 | 0.0548 ±0.0068 | 0.0494 ±0.0080 | 0.0577 ±0.0100 |
| NOTEARS (Nonlinear) [52]: | 0.1762 ±0.0327 | 0.1559 ±0.0209 | 0.1734 ±0.0326 | 0.1763 ±0.0394 | 0.0494 ±0.0096 | 0.0456 ±0.0058 | 0.0417 ±0.0069 | 0.0462 ±0.0042 |
| GES [6]: | 0.2012 ±0.1185 | 0.7465 ±0.1809 | 0.9963 ±0.0078 | 0.9938 ±0.0111 | -0.3315 ±0.0820 | -0.1133 ±0.0631 | 0.0046 ±0.1675 | -0.0089 ±0.0175 |
| PC [33]: | 0.4374 ±0.2841 | 0.8618 ±0.1168 | 0.9601 ±0.0637 | 0.9950 ±0.0065 | -0.2749 ±0.0535 | -0.0404 ±0.0358 | -0.0052 ±0.0131 | -0.0013 ±0.0023 |
| kPC (k=1) [18]: | 0.4708 ±0.2792 | **0.9846** ±0.0166 | **0.9987** ±0.0027 | **0.9975** ±0.0033 | -0.1443 ±0.0624 | -0.0175 ±0.0295 | -0.0014 ±0.0031 | 0.0000 ±0.0000 |
| kPC (k=2) [18]: | 0.4528 ±0.2777 | 0.9731 ±0.0295 | 0.9987 ±0.0027 | 0.9975 ±0.0033 | -0.1893 ±0.0730 | -0.0266 ±0.0386 | -0.0014 ±0.0031 | 0.0000 ±0.0000 |
| **DAGPA (Ours)** | **0.7190** ±0.1381 | 0.9629 ±0.0264 | 0.9899 ±0.0079 | 0.9927 ±0.0051 | 0.0907 ±0.0570 | **0.1031** ±0.0610 | **0.1851** ±0.0615 | **0.3492** ±0.1561 |

### (b) Synthetic binary SF, $r = 4$, CPDAG F1 Arrowhead

| Method | Dataset Settings | | | | | | | |
|---|---|---|---|---|---|---|---|---|
| | $d=10, n=100$ | $d=10, n=1k$ | $d=10, n=10k$ | $d=10, n=100k$ | $d=50, n=100$ | $d=50, n=1k$ | $d=50, n=10k$ | $d=50, n=100k$ |
| DAGMA (Linear) [4]: | 0.0000 ±0.0000 | 0.0000 ±0.0000 | 0.0000 ±0.0000 | 0.0000 ±0.0000 | 0.0091 ±0.0111 | 0.0020 ±0.0063 | 0.0000 ±0.0000 | 0.0010 ±0.0032 |
| DAGMA (Nonlinear) [4]: | 0.1100 ±0.1109 | 0.1141 ±0.0839 | 0.0759 ±0.1125 | 0.0713 ±0.0985 | **0.1244** ±0.0284 | 0.0727 ±0.0527 | 0.0943 ±0.0360 | 0.0954 ±0.0361 |
| NOTEARS (Linear) [51]: | 0.0125 ±0.0395 | 0.0000 ±0.0000 | 0.0000 ±0.0000 | 0.0000 ±0.0000 | 0.0191 ±0.0120 | 0.0040 ±0.0084 | 0.0041 ±0.0085 | 0.0051 ±0.0087 |
| NOTEARS (Nonlinear) [52]: | 0.0105 ±0.0333 | 0.0061 ±0.0192 | 0.0000 ±0.0000 | 0.0053 ±0.0166 | 0.0100 ±0.0114 | 0.0080 ±0.0103 | 0.0051 ±0.0086 | 0.0000 ±0.0000 |
| GES [6]: | 0.0253 ±0.0544 | 0.1724 ±0.1392 | 0.2427 ±0.0730 | **0.4408** ±0.1339 | 0.0663 ±0.0216 | **0.2716** ±0.0383 | **0.5297** ±0.0529 | 0.3611 ±0.3114 |
| PC [33]: | 0.0223 ±0.0300 | **0.2349** ±0.1248 | **0.3093** ±0.1109 | 0.3147 ±0.1203 | 0.0730 ±0.0231 | 0.2456 ±0.0320 | 0.3968 ±0.0569 | **0.4005** ±0.0362 |
| kPC (k=1) [18]: | 0.0211 ±0.0361 | 0.0528 ±0.0543 | 0.1813 ±0.1249 | 0.1299 ±0.0942 | 0.0360 ±0.0192 | 0.0333 ±0.0136 | 0.0169 ±0.0154 | 0.0211 ±0.0137 |
| kPC (k=2) [18]: | 0.0159 ±0.0349 | 0.0452 ±0.0656 | 0.1128 ±0.0828 | 0.1637 ±0.1004 | 0.0400 ±0.0157 | 0.0427 ±0.0150 | 0.0430 ±0.0229 | 0.0590 ±0.0226 |
| **DAGPA (Ours)** | **0.1558** ±0.0800 | 0.2338 ±0.0763 | 0.2589 ±0.1168 | 0.2731 ±0.1252 | 0.0172 ±0.0164 | 0.0276 ±0.0155 | 0.0331 ±0.0118 | 0.0476 ±0.0197 |

### (c) Synthetic binary SF, $r = 4$, CPDAG F1 Skeleton

| Method | Dataset Settings | | | | | | | |
|---|---|---|---|---|---|---|---|---|
| | $d=10, n=100$ | $d=10, n=1k$ | $d=10, n=10k$ | $d=10, n=100k$ | $d=50, n=100$ | $d=50, n=1k$ | $d=50, n=10k$ | $d=50, n=100k$ |
| DAGMA (Linear) [4]: | 0.1182 ±0.0801 | 0.0625 ±0.0319 | 0.0846 ±0.0569 | 0.1074 ±0.0528 | 0.0957 ±0.0156 | 0.0722 ±0.0154 | 0.0637 ±0.0199 | 0.0785 ±0.0248 |
| DAGMA (Nonlinear) [4]: | 0.3830 ±0.0892 | 0.2930 ±0.1131 | 0.2995 ±0.0944 | 0.3001 ±0.0924 | **0.2290** ±0.0240 | 0.2179 ±0.0538 | 0.2149 ±0.0313 | 0.2347 ±0.0506 |
| NOTEARS (Linear) [51]: | 0.1631 ±0.1014 | 0.1159 ±0.0605 | 0.0980 ±0.0618 | 0.1341 ±0.0477 | 0.1305 ±0.0223 | 0.1040 ±0.0249 | 0.0922 ±0.0249 | 0.1112 ±0.0195 |
| NOTEARS (Nonlinear) [52]: | 0.0739 ±0.0750 | 0.0284 ±0.0504 | 0.0791 ±0.0714 | 0.0817 ±0.0957 | 0.0405 ±0.0246 | 0.0293 ±0.0205 | 0.0294 ±0.0171 | 0.0245 ±0.0206 |
| GES [6]: | 0.1416 ±0.1132 | 0.4808 ±0.0780 | 0.7679 ±0.0314 | 0.8424 ±0.0302 | 0.1747 ±0.0148 | 0.4210 ±0.0358 | 0.7082 ±0.0507 | 0.4682 ±0.4031 |
| PC [33]: | 0.3141 ±0.1136 | 0.5696 ±0.0531 | 0.7348 ±0.0409 | 0.7840 ±0.0457 | 0.1984 ±0.0180 | 0.5263 ±0.0269 | **0.7387** ±0.0342 | **0.7675** ±0.0187 |
| kPC (k=1) [18]: | 0.3237 ±0.1185 | **0.6298** ±0.0675 | **0.8036** ±0.0174 | **0.8551** ±0.0264 | 0.2114 ±0.0244 | **0.5395** ±0.0290 | 0.6921 ±0.0280 | 0.6553 ±0.0214 |
| kPC (k=2) [18]: | 0.3141 ±0.1136 | 0.5836 ±0.0662 | 0.7538 ±0.0483 | 0.8036 ±0.0361 | 0.2005 ±0.0206 | 0.5344 ±0.0268 | 0.7307 ±0.0329 | 0.7388 ±0.0135 |
| **DAGPA (Ours)** | **0.5007** ±0.0512 | 0.5889 ±0.0564 | 0.6445 ±0.0586 | 0.6512 ±0.0560 | 0.0435 ±0.0249 | 0.0677 ±0.0178 | 0.0726 ±0.0265 | 0.0956 ±0.0358 |

### (d) Synthetic binary SF, $r = 4$, CPDAG SHD

| Method | Dataset Settings | | | | | | | |
|---|---|---|---|---|---|---|---|---|
| | $d=10, n=100$ | $d=10, n=1k$ | $d=10, n=10k$ | $d=10, n=100k$ | $d=50, n=100$ | $d=50, n=1k$ | $d=50, n=10k$ | $d=50, n=100k$ |
| DAGMA (Linear) [4]: | 38.6000 ±0.8433 | 39.1000 ±0.7379 | 38.6000 ±1.1738 | 38.3000 ±1.0593 | 197.9000 ±1.6633 | 198.0000 ±1.2472 | 198.6000 ±1.7764 | 197.8000 ±1.4757 |
| DAGMA (Nonlinear) [4]: | 36.4000 ±2.0656 | 37.1000 ±1.8529 | 36.8000 ±3.0478 | 36.1000 ±2.7669 | 221.4000 ±10.6999 | 190.5000 ±4.3525 | 189.6000 ±3.6878 | 188.4000 ±3.9497 |
| NOTEARS (Linear) [51]: | 38.5000 ±1.5092 | 38.4000 ±1.4298 | 38.3000 ±1.3375 | 38.1000 ±1.3703 | 198.9000 ±2.4244 | 197.4000 ±1.3499 | 198.0000 ±2.1602 | 197.1000 ±1.5239 |
| NOTEARS (Nonlinear) [52]: | 39.5000 ±1.0801 | 39.8000 ±0.4216 | 39.5000 ±1.2693 | 39.5000 ±0.7071 | 199.8000 ±1.8135 | 199.1000 ±1.3703 | 199.4000 ±0.8433 | 199.4000 ±1.2649 |
| GES [6]: | 31.2000 ±16.4776 | 35.9000 ±2.8067 | 35.4000 ±2.9136 | 28.1000 ±5.0870 | 197.6000 ±3.4705 | 167.4000 ±5.8916 | 127.0000 ±12.9013 | 75.6000 ±65.6374 |
| PC [33]: | 39.6000 ±1.3499 | 34.7000 ±2.8304 | 33.0000 ±4.1366 | 32.0000 ±4.7376 | 206.9000 ±2.3310 | 181.4000 ±8.2892 | 158.6000 ±15.1526 | 177.7000 ±11.6433 |
| kPC (k=1) [18]: | 39.8000 ±1.3166 | 33.9000 ±3.6040 | 32.7000 ±4.7152 | 34.1000 ±4.3063 | 207.8000 ±2.1499 | 188.9000 ±11.5609 | 196.5000 ±13.1085 | 270.3000 ±20.3309 |
| kPC (k=2) [18]: | 39.9000 ±1.2867 | 34.2000 ±2.5734 | 33.5000 ±3.4721 | 32.2000 ±4.8944 | 205.7000 ±2.5841 | 181.6000 ±9.2280 | 167.9000 ±14.4564 | 203.5000 ±7.9757 |
| **DAGPA (Ours)** | 37.1250 ±2.8482 | 36.0000 ±2.2315 | 35.1600 ±3.6609 | 34.7400 ±3.7022 | 233.0000 ±19.1066 | 244.4800 ±10.0249 | 250.1000 ±8.9972 | 272.6444 ±21.6671 |

### (e) Synthetic binary SF, $r = 4$, DAG F1

| Method | Dataset Settings | | | | | | | |
|---|---|---|---|---|---|---|---|---|
| | $d=10, n=100$ | $d=10, n=1k$ | $d=10, n=10k$ | $d=10, n=100k$ | $d=50, n=100$ | $d=50, n=1k$ | $d=50, n=10k$ | $d=50, n=100k$ |
| DAGMA (Linear) [4]: | 0.0682 ±0.0555 | 0.0433 ±0.0271 | 0.0517 ±0.0409 | 0.0651 ±0.0493 | 0.0673 ±0.0142 | 0.0530 ±0.0128 | 0.0406 ±0.0086 | 0.0603 ±0.0218 |
| DAGMA (Nonlinear) [4]: | 0.2486 ±0.0855 | 0.1928 ±0.1048 | 0.1950 ±0.0860 | 0.1987 ±0.0994 | 0.1565 ±0.0358 | 0.1362 ±0.0558 | 0.1259 ±0.0300 | 0.1475 ±0.0555 |
| NOTEARS (Linear) [51]: | 0.1008 ±0.0867 | 0.0882 ±0.0547 | 0.0608 ±0.0435 | 0.0879 ±0.0382 | 0.0845 ±0.0229 | 0.0738 ±0.0156 | 0.0628 ±0.0174 | 0.0763 ±0.0203 |
| NOTEARS (Nonlinear) [52]: | 0.0369 ±0.0560 | 0.0096 ±0.0203 | 0.0276 ±0.0385 | 0.0497 ±0.0714 | 0.0144 ±0.0149 | 0.0127 ±0.0131 | 0.0118 ±0.0077 | 0.0098 ±0.0080 |
| **DAGPA (Ours)** | 0.2219 ±0.0624 | 0.2752 ±0.0778 | 0.3358 ±0.1050 | 0.3581 ±0.0739 | 0.0235 ±0.0144 | 0.0334 ±0.0174 | 0.0416 ±0.0110 | 0.0532 ±0.0150 |

integration direction is promising and could leverage complementary strengths of both paradigms, the normalization challenge for nonlinear models and comprehensive empirical evaluation remain important future work.

## G.2 Alternative Conditional Independence Measurements and Tradeoffs

Our framework uses p-values from statistical conditional independence (CI) tests as soft measures of independence strength. This section discusses the motivation for CI-based measurements and compares different CI measurement approaches.

### G.2.1 Motivation: Robustness and Interpretability

**Robustness to Data Preprocessing.** Many statistical CI tests offer inherent robustness due to their scale-invariance properties. Standard CI tests remain reliable across different preprocessing pipelines: Fisher-z test statistics are invariant under linear transformations since partial correlations $\rho_{XY|Z}$ are preserved when variables are standardized; chi-square tests operate on frequency counts which are unaffected by scaling; kernel-based tests use normalized embeddings providing scale-invariance by design. We empirically verified this: DAGPA maintains CI-MCC scores with low variance of $\approx 0.12$ across unnormalized, standardized, min-max normalized, and robust-scaled versions of the same dataset ($n = 100, d = 10$ ER-2 synthetic data), demonstrating stability that is valuable when data sources have heterogeneous scales or different preprocessing conventions.

**Interpretability via Direct Hypothesis Testing.** CI-based measurements provide direct interpretability through testable hypotheses. Each structural decision corresponds to specific CI tests with p-values, enabling transparent explanations. For example, when removing edge $X \rightarrow Y$, we can justify: "Edge removed because unconditional test shows $X \not\perp\!\!\!\perp Y$ with p-value 0.001 and conditional test shows $X \perp\!\!\!\perp Y|Z$ with p-value 0.82, indicating $X$ influences $Y$ likely through $Z$ but does not have direct causal link to $Y$." Domain experts can independently verify these claims by examining raw data, running alternative tests, or checking whether distributional assumptions hold. This transparency is particularly valuable in high-stakes applications (medical diagnosis, policy making) where algorithmic decisions require human oversight and regulatory approval.

### G.2.2 Overview of CI Measurement Methods

Different CI tests present tradeoffs across multiple dimensions. Table 5 summarizes key characteristics:

Table 5: Comparison of CI measurement methods. "Small-$n$" indicates performance with $n < 200$.

| Method | Assumptions | Complexity | Small-$n$ | Nonlinearity | p-value |
|--------|-------------|------------|-----------|--------------|---------|
| Fisher-z [11] | Gaussian, linear | $O(n)$ | Excellent | Poor | Native |
| Chi-square | Categorical | $O(n)$ | Good | Arbitrary | Native |
| KCIT [48] | Nonparametric | $O(n^2)$ | Poor | Excellent | Approximate |
| RCIT [35] | Nonparametric | $O(n \log n)$ | Moderate | Good | Approximate |
| CMI (k-NN) | Nonparametric | $O(n \log n)$ | Poor | Good | Requires transform |

**Fisher-z Test:** Tests whether partial correlation $\rho_{XY|Z}$ equals zero using transformation $z = \frac{1}{2}\sqrt{n - |Z| - 3}\log\frac{1+\hat{\rho}}{1-\hat{\rho}} \sim \mathcal{N}(0, 1)$. Strengths include excellent sample efficiency (reliable with $n \geq 50$ for $|Z| \leq 1$), $O(n)$ computational cost, and native p-values. Limitations: designed for Gaussian data and primarily detects linear relationships. Our default choice for continuous data in low-sample regime.

**Chi-square Test:** Tests independence via contingency table using $\chi^2 = \sum_{i,j} \frac{(O_{ij} - E_{ij})^2}{E_{ij}}$. Strengths include no distributional assumptions, detection of arbitrary associations, and good small-sample performance. Limitations: only applicable to categorical variables, requires sufficient cell counts ($E_{ij} \geq 5$), and exponential complexity with $|Z|$. Our default for binary/categorical data.

**Kernel CI Test (KCIT):** Tests independence via kernel embeddings using Hilbert-Schmidt Independence Criterion. Strengths include detecting arbitrary nonlinear dependencies and no parametric

assumptions. Limitations: $O(n^2)$ complexity, requires $n > 500$ for reliable results, and kernel bandwidth selection affects performance. Recommended for large datasets ($n > 1000$) with suspected nonlinear relationships.

**Randomized CI Test (RCIT):** Approximates KCIT using random Fourier features, reducing complexity to $O(n \log n)$. Provides middle ground between Fisher-z (fast, linear) and KCIT (slow, nonlinear) for moderate samples ($500 < n < 5000$).

**Conditional Mutual Information (CMI):** Measures $I(X; Y|Z) = \mathbb{E}\left[\log \frac{p(X,Y|Z)}{p(X|Z)p(Y|Z)}\right]$ via k-NN or kernel density estimation. Provides information-theoretic interpretation but lacks native p-values (requires transformation like $p = \exp(-\beta \cdot \hat{I})$) and needs $n > 1000$ for reliable density estimation.

**Tradeoffs and Selection Guidance.** For causal discovery with $d$ variables, total CI testing requires $O(d^2)$ order-0 tests and $O(d^3)$ order-1 tests. With $d = 50, n = 1000$: Fisher-z completes in $\sim$30 seconds, RCIT in $\sim$5 minutes, KCIT in $\sim$45 minutes. Sample efficiency also varies: Fisher-z achieves high power with $n < 200$ for linear relationships, while kernel methods require $n > 500$ but eventually match or exceed Fisher-z for nonlinear cases. Our framework is measurement-agnostic—any function producing $[0, 1]$ scores can be substituted via our API:

```
model = DAGPA(ci_test='fisherz')      # Default
model = DAGPA(ci_test='kcit')         # For nonlinear data
model = DAGPA(ci_test=MyCustomTest()) # Custom implementation
```

We recommend Fisher-z or chi-square for most applications due to sample efficiency and computational scalability, with kernel methods reserved for large datasets with confirmed nonlinear relationships. Practitioners should conduct sensitivity analysis across multiple tests when possible, as edges appearing consistently provide higher confidence regardless of specific test assumptions.

## H   Licenses

In this work, we evaluated our method on two publicly available causal discovery benchmark datasets:

**Sachs Dataset**: The Sachs dataset contains simultaneous measurements of 11 phosphorylated proteins and phospholipids derived from thousands of individual primary immune system cells, subjected to both general and specific molecular interventions. This dataset was originally published by Sachs et al. (2005) and is widely used as a benchmark in causal discovery research. The dataset is publicly available through multiple repositories including bnlearn [30], other causal discovery toolboxes.

**Availability**: The dataset is publicly accessible for research purposes through various causal discovery software packages and repositories.

**LUCAS Dataset**: The LUCAS (LUng CAncer Simple set) dataset [14] is a synthetic benchmark dataset consisting of 12 binary variables and 2000 instances, representing 12 different causal relationships in a medical diagnosis problem for identifying patients with lung cancer. This dataset was created as part of the Causality Workbench project to provide standardized benchmarks for testing causal discovery algorithms.

**Availability**: The dataset is publicly available for research and educational purposes through the Causality Workbench repository.

Usage Declaration: Both datasets were used in accordance with their respective terms of use for academic research purposes. No additional permissions were required for their use in this study.

## I   Societal Impact Statement

Our differentiable d-separation framework for causal discovery has potential positive impacts in healthcare (identifying causal factors in disease), public policy (evaluating intervention effectiveness), scientific discovery (understanding complex systems), and algorithmic fairness (distinguishing causation from correlation in decision systems). However, several risks must be acknowledged: (1) incorrect causal discoveries could lead to harmful interventions if implemented without domain expert validation; (2) causal discovery in sensitive domains may raise privacy concerns, necessitating differential privacy techniques; (3) when applied to historically biased datasets, discovered

relationships might reflect and perpetuate these biases rather than ground truth; and (4) computational requirements might limit accessibility to well-resourced institutions. We recommend multiple mitigations: returning diverse candidate structures rather than single models, requiring expert validation before implementation, implementing privacy-preserving techniques with sensitive data, examining discovered relationships for bias, and developing more efficient implementations to improve accessibility. Causal discovery tools require responsible application and domain expertise, especially in high-stakes domains where incorrect inferences could lead to harmful consequences.

# NeurIPS Paper Checklist

1. **Claims**

   Question: Do the main claims made in the abstract and introduction accurately reflect the paper's contributions and scope?

   Answer: [Yes]

   Justification: The abstract and introduction claim that we contribute a novel framework for causal discovery that bridges constraint-based and continuous-optimization score-based methods with promising empirical results. These claims accurately reflect our work's scope and contributions as evidenced in: (1) Section 3, where we develop the novel differentiable d-separation framework with theoretical guarantees and percolation-based probabilistic interpretations; (2) Section 4, where we demonstrate one practical instantiation of this framework into the model we named DAGPA; and (3) Section 5, where we empirically validate that our approach achieves competitive performance compared to established methods across multiple datasets and settings.

   Guidelines:

   - The answer NA means that the abstract and introduction do not include the claims made in the paper.
   - The abstract and/or introduction should clearly state the claims made, including the contributions made in the paper and important assumptions and limitations. A No or NA answer to this question will not be perceived well by the reviewers.
   - The claims made should match theoretical and experimental results, and reflect how much the results can be expected to generalize to other settings.
   - It is fine to include aspirational goals as motivation as long as it is clear that these goals are not attained by the paper.

2. **Limitations**

   Question: Does the paper discuss the limitations of the work performed by the authors?

   Answer: [Yes]

   Justification: We provide a dedicated Limitations paragraph in Section 7 that thoroughly discusses multiple aspects of our work's constraints and outlines specific future research directions to address them. Throughout the paper, we are transparent about the scope of our empirical validation and the provisional nature of our current implementation compared to the theoretical framework's potential.

   Guidelines:

   - The answer NA means that the paper has no limitation while the answer No means that the paper has limitations, but those are not discussed in the paper.
   - The authors are encouraged to create a separate "Limitations" section in their paper.
   - The paper should point out any strong assumptions and how robust the results are to violations of these assumptions (e.g., independence assumptions, noiseless settings, model well-specification, asymptotic approximations only holding locally). The authors should reflect on how these assumptions might be violated in practice and what the implications would be.
   - The authors should reflect on the scope of the claims made, e.g., if the approach was only tested on a few datasets or with a few runs. In general, empirical results often depend on implicit assumptions, which should be articulated.
   - The authors should reflect on the factors that influence the performance of the approach. For example, a facial recognition algorithm may perform poorly when image resolution is low or images are taken in low lighting. Or a speech-to-text system might not be used reliably to provide closed captions for online lectures because it fails to handle technical jargon.
   - The authors should discuss the computational efficiency of the proposed algorithms and how they scale with dataset size.
   - If applicable, the authors should discuss possible limitations of their approach to address problems of privacy and fairness.
   - While the authors might fear that complete honesty about limitations might be used by reviewers as grounds for rejection, a worse outcome might be that reviewers discover limitations that aren't acknowledged in the paper. The authors should use their best judgment and recognize

that individual actions in favor of transparency play an important role in developing norms that preserve the integrity of the community. Reviewers will be specifically instructed to not penalize honesty concerning limitations.

3. **Theory assumptions and proofs**

   Question: For each theoretical result, does the paper provide the full set of assumptions and a complete (and correct) proof?

   Answer: [Yes]

   Justification: In our paper, all assumptions are explicitly stated before each theorem and lemma, and the complete proofs are provided in Appendix A, with every item properly numbered and consistently cross-referenced throughout the paper. In addition, we precede each theoretical result with intuitive explanations connecting the formal statements to the broader framework, and for the most important results, such as Theorem 3.2, we provide proof sketches in the main text that highlight the critical steps and insights.

   Guidelines:

   - The answer NA means that the paper does not include theoretical results.
   - All the theorems, formulas, and proofs in the paper should be numbered and cross-referenced.
   - All assumptions should be clearly stated or referenced in the statement of any theorems.
   - The proofs can either appear in the main paper or the supplemental material, but if they appear in the supplemental material, the authors are encouraged to provide a short proof sketch to provide intuition.
   - Inversely, any informal proof provided in the core of the paper should be complemented by formal proofs provided in appendix or supplemental material.
   - Theorems and Lemmas that the proof relies upon should be properly referenced.

4. **Experimental result reproducibility**

   Question: Does the paper fully disclose all the information needed to reproduce the main experimental results of the paper to the extent that it affects the main claims and/or conclusions of the paper (regardless of whether the code and data are provided or not)?

   Answer: [Yes]

   Justification: We provide complete reproducibility through detailed algorithm pseudocode for DAGPA (Appendix C), code and data release (Appendix D.1), dataset creation and preprocessing procedures (Appendix D.2), and comprehensive hyperparameter specifications for both our method (Appendix D.4) and all baselines (Appendix D.5). Additionally, we document the computational resources used (Appendix D.6) and fully explain our evaluation metrics (Appendix D.3), ensuring all aspects of our experimental pipeline can be independently reproduced.

   Guidelines:

   - The answer NA means that the paper does not include experiments.
   - If the paper includes experiments, a No answer to this question will not be perceived well by the reviewers: Making the paper reproducible is important, regardless of whether the code and data are provided or not.
   - If the contribution is a dataset and/or model, the authors should describe the steps taken to make their results reproducible or verifiable.
   - Depending on the contribution, reproducibility can be accomplished in various ways. For example, if the contribution is a novel architecture, describing the architecture fully might suffice, or if the contribution is a specific model and empirical evaluation, it may be necessary to either make it possible for others to replicate the model with the same dataset, or provide access to the model. In general. releasing code and data is often one good way to accomplish this, but reproducibility can also be provided via detailed instructions for how to replicate the results, access to a hosted model (e.g., in the case of a large language model), releasing of a model checkpoint, or other means that are appropriate to the research performed.
   - While NeurIPS does not require releasing code, the conference does require all submissions to provide some reasonable avenue for reproducibility, which may depend on the nature of the contribution. For example
     (a) If the contribution is primarily a new algorithm, the paper should make it clear how to reproduce that algorithm.

(b) If the contribution is primarily a new model architecture, the paper should describe the architecture clearly and fully.

(c) If the contribution is a new model (e.g., a large language model), then there should either be a way to access this model for reproducing the results or a way to reproduce the model (e.g., with an open-source dataset or instructions for how to construct the dataset).

(d) We recognize that reproducibility may be tricky in some cases, in which case authors are welcome to describe the particular way they provide for reproducibility. In the case of closed-source models, it may be that access to the model is limited in some way (e.g., to registered users), but it should be possible for other researchers to have some path to reproducing or verifying the results.

5. **Open access to data and code**

Question: Does the paper provide open access to the data and code, with sufficient instructions to faithfully reproduce the main experimental results, as described in supplemental material?

Answer: [Yes]

Justification: We provide complete code and data access through an anonymous GitHub repository (linked in Appendix D.1) that includes all implementation files, datasets (both synthetic generators and real-world data), and evaluation scripts necessary to reproduce our results. The repository contains detailed documentation with exact environment setup instructions (via Conda environment files), step-by-step commands to run all experiments, data preprocessing pipelines, and scripts to generate all figures and tables presented in the paper.

Guidelines:

- The answer NA means that paper does not include experiments requiring code.
- Please see the NeurIPS code and data submission guidelines (`https://nips.cc/public/guides/CodeSubmissionPolicy`) for more details.
- While we encourage the release of code and data, we understand that this might not be possible, so "No" is an acceptable answer. Papers cannot be rejected simply for not including code, unless this is central to the contribution (e.g., for a new open-source benchmark).
- The instructions should contain the exact command and environment needed to run to reproduce the results. See the NeurIPS code and data submission guidelines (`https://nips.cc/public/guides/CodeSubmissionPolicy`) for more details.
- The authors should provide instructions on data access and preparation, including how to access the raw data, preprocessed data, intermediate data, and generated data, etc.
- The authors should provide scripts to reproduce all experimental results for the new proposed method and baselines. If only a subset of experiments are reproducible, they should state which ones are omitted from the script and why.
- At submission time, to preserve anonymity, the authors should release anonymized versions (if applicable).
- Providing as much information as possible in supplemental material (appended to the paper) is recommended, but including URLs to data and code is permitted.

6. **Experimental setting/details**

Question: Does the paper specify all the training and test details (e.g., data splits, hyperparameters, how they were chosen, type of optimizer, etc.) necessary to understand the results?

Answer: [Yes]

Justification: The main text (Section 5) provides essential experimental details including dataset characteristics, evaluation metrics, baselines, and the overall experimental design, while Appendix D contains comprehensive information on hyperparameters, model selection criteria, data creation and preprocessing, and computational resources. We explicitly describe our parameter selection process and provide justification for key design choices to ensure results can be fully understood and contextualized.

Guidelines:

- The answer NA means that the paper does not include experiments.
- The experimental setting should be presented in the core of the paper to a level of detail that is necessary to appreciate the results and make sense of them.

- The full details can be provided either with the code, in appendix, or as supplemental material.

7. **Experiment statistical significance**

    Question: Does the paper report error bars suitably and correctly defined or other appropriate information about the statistical significance of the experiments?

    Answer: [Yes]

    Justification: Our primary results are presented as empirical cumulative distribution functions (CDFs) over multiple (10) independent dataset instances per configuration, fully displaying performance variability across different data realizations and allowing for direct statistical comparison between methods. For tabular results, we report mean values with standard deviation error bars (1-sigma), clearly labeled as such, and calculated using standard statistical formulas across the independent trials. The sources of variability (different random DAG structures and data samples) are explicitly described in Section 5, and all statistical comparisons are appropriately referenced in the text.

    Guidelines:

    - The answer NA means that the paper does not include experiments.
    - The authors should answer "Yes" if the results are accompanied by error bars, confidence intervals, or statistical significance tests, at least for the experiments that support the main claims of the paper.
    - The factors of variability that the error bars are capturing should be clearly stated (for example, train/test split, initialization, random drawing of some parameter, or overall run with given experimental conditions).
    - The method for calculating the error bars should be explained (closed form formula, call to a library function, bootstrap, etc.)
    - The assumptions made should be given (e.g., Normally distributed errors).
    - It should be clear whether the error bar is the standard deviation or the standard error of the mean.
    - It is OK to report 1-sigma error bars, but one should state it. The authors should preferably report a 2-sigma error bar than state that they have a 96% CI, if the hypothesis of Normality of errors is not verified.
    - For asymmetric distributions, the authors should be careful not to show in tables or figures symmetric error bars that would yield results that are out of range (e.g. negative error rates).
    - If error bars are reported in tables or plots, The authors should explain in the text how they were calculated and reference the corresponding figures or tables in the text.

8. **Experiments compute resources**

    Question: For each experiment, does the paper provide sufficient information on the computer resources (type of compute workers, memory, time of execution) needed to reproduce the experiments?

    Answer: [Yes]

    Justification: Yes. We detail the computation resources used for each experiment settings in Appendix D.6.

    Guidelines:

    - The answer NA means that the paper does not include experiments.
    - The paper should indicate the type of compute workers CPU or GPU, internal cluster, or cloud provider, including relevant memory and storage.
    - The paper should provide the amount of compute required for each of the individual experimental runs as well as estimate the total compute.
    - The paper should disclose whether the full research project required more compute than the experiments reported in the paper (e.g., preliminary or failed experiments that didn't make it into the paper).

9. **Code of ethics**

    Question: Does the research conducted in the paper conform, in every respect, with the NeurIPS Code of Ethics https://neurips.cc/public/EthicsGuidelines?

Answer: [Yes]

Justification: Yes. This research fully conforms to NeurIPS Code of Ethics.

Guidelines:

- The answer NA means that the authors have not reviewed the NeurIPS Code of Ethics.
- If the authors answer No, they should explain the special circumstances that require a deviation from the Code of Ethics.
- The authors should make sure to preserve anonymity (e.g., if there is a special consideration due to laws or regulations in their jurisdiction).

10. **Broader impacts**

Question: Does the paper discuss both potential positive societal impacts and negative societal impacts of the work performed?

Answer: [Yes]

Justification: We provide a comprehensive Broader Impact Statement in Appendix I that discusses both potential positive impacts (in healthcare, public policy, scientific discovery, and algorithmic fairness) and potential negative impacts (misplaced trust in discovered causal relationships, privacy concerns, potential for reinforcing biases, and accessibility challenges). For each potential negative impact, we also suggest specific mitigation strategies that practitioners can implement when applying our method.

Guidelines:

- The answer NA means that there is no societal impact of the work performed.
- If the authors answer NA or No, they should explain why their work has no societal impact or why the paper does not address societal impact.
- Examples of negative societal impacts include potential malicious or unintended uses (e.g., disinformation, generating fake profiles, surveillance), fairness considerations (e.g., deployment of technologies that could make decisions that unfairly impact specific groups), privacy considerations, and security considerations.
- The conference expects that many papers will be foundational research and not tied to particular applications, let alone deployments. However, if there is a direct path to any negative applications, the authors should point it out. For example, it is legitimate to point out that an improvement in the quality of generative models could be used to generate deepfakes for disinformation. On the other hand, it is not needed to point out that a generic algorithm for optimizing neural networks could enable people to train models that generate Deepfakes faster.
- The authors should consider possible harms that could arise when the technology is being used as intended and functioning correctly, harms that could arise when the technology is being used as intended but gives incorrect results, and harms following from (intentional or unintentional) misuse of the technology.
- If there are negative societal impacts, the authors could also discuss possible mitigation strategies (e.g., gated release of models, providing defenses in addition to attacks, mechanisms for monitoring misuse, mechanisms to monitor how a system learns from feedback over time, improving the efficiency and accessibility of ML).

11. **Safeguards**

Question: Does the paper describe safeguards that have been put in place for responsible release of data or models that have a high risk for misuse (e.g., pretrained language models, image generators, or scraped datasets)?

Answer: [NA]

Justification: Our work presents a methodological framework for causal discovery that does not pose the high-risk misuse concerns typically associated with generative models or scraped datasets. We only use synthetic datasets (generated with provided code) and well-established, properly cited benchmark datasets (e.g. Sachs) from trusted scientific sources. Our method does not learn representations that could be misused for generating harmful content, creating fake profiles, or similar high-risk applications that would necessitate special safeguards beyond standard research best practices.

Guidelines:

- The answer NA means that the paper poses no such risks.
- Released models that have a high risk for misuse or dual-use should be released with necessary safeguards to allow for controlled use of the model, for example by requiring that users adhere to usage guidelines or restrictions to access the model or implementing safety filters.
- Datasets that have been scraped from the Internet could pose safety risks. The authors should describe how they avoided releasing unsafe images.
- We recognize that providing effective safeguards is challenging, and many papers do not require this, but we encourage authors to take this into account and make a best faith effort.

12. **Licenses for existing assets**

Question: Are the creators or original owners of assets (e.g., code, data, models), used in the paper, properly credited and are the license and terms of use explicitly mentioned and properly respected?

Answer: [Yes]

Justification: We properly cite all original papers for datasets and code libraries used in our work. In Appendix H, we provide detailed license information for the Sachs dataset (the only external real-world dataset used) and all third-party libraries/packages employed in our implementation, including version numbers and URLs to their repositories. For synthetic datasets, we specify that they are generated by our code which is released under an MIT license.

Guidelines:

- The answer NA means that the paper does not use existing assets.
- The authors should cite the original paper that produced the code package or dataset.
- The authors should state which version of the asset is used and, if possible, include a URL.
- The name of the license (e.g., CC-BY 4.0) should be included for each asset.
- For scraped data from a particular source (e.g., website), the copyright and terms of service of that source should be provided.
- If assets are released, the license, copyright information, and terms of use in the package should be provided. For popular datasets, `paperswithcode.com/datasets` has curated licenses for some datasets. Their licensing guide can help determine the license of a dataset.
- For existing datasets that are re-packaged, both the original license and the license of the derived asset (if it has changed) should be provided.
- If this information is not available online, the authors are encouraged to reach out to the asset's creators.

13. **New assets**

Question: Are new assets introduced in the paper well documented and is the documentation provided alongside the assets?

Answer: [Yes]

Justification: We release our implementation of DAGPA via an anonymized GitHub repository (linked in Appendix D.1) with comprehensive documentation including: installation instructions, usage examples, API documentation, parameter descriptions, training/optimization/sampling procedures, and known limitations. The repository includes a structured README file, detailed code comments, and a clear MIT license statement.

Guidelines:

- The answer NA means that the paper does not release new assets.
- Researchers should communicate the details of the dataset/code/model as part of their submissions via structured templates. This includes details about training, license, limitations, etc.
- The paper should discuss whether and how consent was obtained from people whose asset is used.
- At submission time, remember to anonymize your assets (if applicable). You can either create an anonymized URL or include an anonymized zip file.

14. **Crowdsourcing and research with human subjects**

Question: For crowdsourcing experiments and research with human subjects, does the paper include the full text of instructions given to participants and screenshots, if applicable, as well as details about compensation (if any)?

Answer: [NA]

Justification: This work does not involve crowdsourcing nor research with human subjects.

Guidelines:

- The answer NA means that the paper does not involve crowdsourcing nor research with human subjects.
- Including this information in the supplemental material is fine, but if the main contribution of the paper involves human subjects, then as much detail as possible should be included in the main paper.
- According to the NeurIPS Code of Ethics, workers involved in data collection, curation, or other labor should be paid at least the minimum wage in the country of the data collector.

15. **Institutional review board (IRB) approvals or equivalent for research with human subjects**

Question: Does the paper describe potential risks incurred by study participants, whether such risks were disclosed to the subjects, and whether Institutional Review Board (IRB) approvals (or an equivalent approval/review based on the requirements of your country or institution) were obtained?

Answer: [NA]

Justification: This work does not involve crowdsourcing nor research with human subjects.

Guidelines:

- The answer NA means that the paper does not involve crowdsourcing nor research with human subjects.
- Depending on the country in which research is conducted, IRB approval (or equivalent) may be required for any human subjects research. If you obtained IRB approval, you should clearly state this in the paper.
- We recognize that the procedures for this may vary significantly between institutions and locations, and we expect authors to adhere to the NeurIPS Code of Ethics and the guidelines for their institution.
- For initial submissions, do not include any information that would break anonymity (if applicable), such as the institution conducting the review.

16. **Declaration of LLM usage**

Question: Does the paper describe the usage of LLMs if it is an important, original, or non-standard component of the core methods in this research? Note that if the LLM is used only for writing, editing, or formatting purposes and does not impact the core methodology, scientific rigorousness, or originality of the research, declaration is not required.

Answer: [NA]

Justification: This work does not involve LLMs as any component of our core methodology, algorithms, or experiments. LLMs were used solely for writing assistance, editing, and LaTeX formatting, which per the NeurIPS LLM policy does not impact the scientific methodology or originality of our work and therefore does not require declaration beyond this checklist.

Guidelines:

- The answer NA means that the core method development in this research does not involve LLMs as any important, original, or non-standard components.
- Please refer to our LLM policy (`https://neurips.cc/Conferences/2025/LLM`) for what should or should not be described.

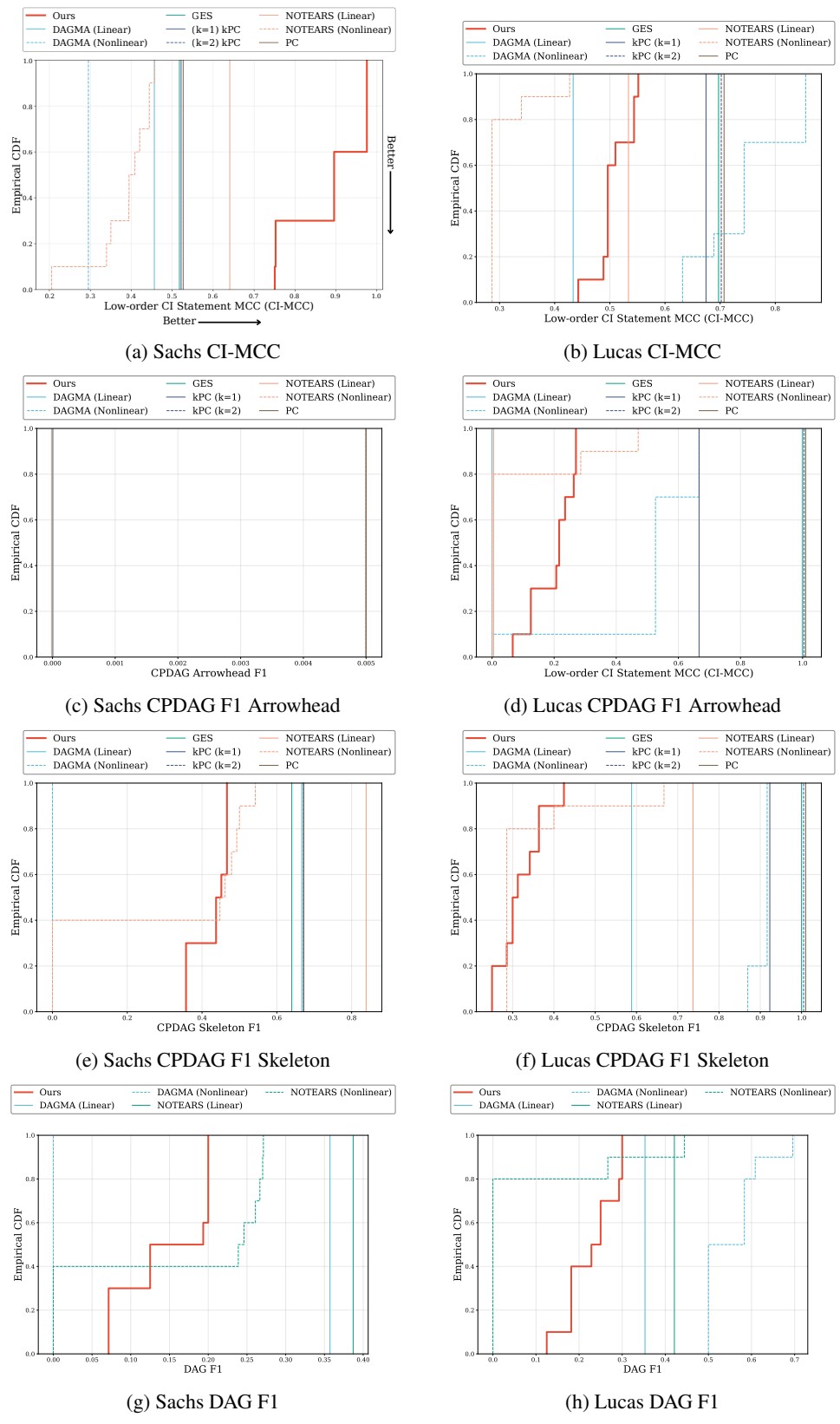

Figure 9: Full results on Sachs [28] and Lucas [14] real-world dataset.

