# OpenReview forum: "Differentiable Constraint-Based Causal Discovery"
_NeurIPS.cc/2025/Conference — NeurIPS 2025 poster_

### Official Review · Reviewer_523A · 2025-06-24

**Clarity:** 3
**Significance:** 2
**Originality:** 3
**Rating:** 4
**Confidence:** 4

**Summary:**

In this paper, the authors introduce DAGPA (DAG Percolation Apartness), a novel causal discovery method formulated as an optimization problem, which directly employs results from conditional independence tests. In detail, DAGPA computes the $p$-value of marginal independence tests between any two distinct variables $(x, y)$ and of conditional independence tests between any three distinct variables $(x, y, z)$. Then, the authors train the model by relating conditional independence statements with $d$-separation conditions. They inscribe this matching in a differentiable optimization problem by (i.) formulating $d$-separation as a first-order-logic formula and (ii.) applying a soft logic framework to optimize the DAG edges. The authors test empirically their work against differentiable score-based baselines (NOTEARS, DAGMA) and constraint-based baselines (PC).

**Questions:**

- In Appendix, the method underperforms in terms of F1 and SHD. Can the authors justify this behaviour? While I agree that MCC is an interesting measure, the paper should also discuss more in detail its relation with more widely adopted metrics in the main body.
- It seems that to compute the loss, you need to precompute all possible conditional independence tests. Is this correct? How would this compare with the number of tests performed by PC? How does DAGPA compare to PC and score-based methods in terms of computational time?
- Differentiable score-based methods, such as NOTEARS and DAGMA, are proven to work remarkably well on non-normalized data, while struggling to recover the DAG after rescaling data ("Beware of the Simulated DAG! Causal Discovery Benchmarks May Be Easy To Game.", Reisach et al., 2021). How is the pre-processing handled in your experimental pipeline?
- Can this approach be integrated as a sort of "regularization" technique to existing score-based differentiable methods? For instance, in the linear case, I would have imagined to also minimize the reconstruction MSE loss $\|\|\mathbf{W}^\top\mathbf{X}-\mathbf{X}\|\|_2^2$ adopted by NOTEARS et similia. Is there some fundamental reason to avoid doing so?

**Ethical Concerns:**

["NO or VERY MINOR ethics concerns only"]

**Final Justification:**

The authors addressed my comments concerning the evaluation metric, the pre-processing pipeline, and the integration with linear causal structure learning algorithms. I am concerned by the computational time required by the method. However, the paper opens up an interesting novel direction and might be worth accepting.

**Limitations:**

yes

**Quality:**

2

**Strengths And Weaknesses:**

Strengths.

- The method is designed to improve causal discovery performance in the small sample size regime, and experimental results hint at better results in terms of Matthews Correlation Coefficient (MCC) compared to baselines in this particular scenario.
- The overall method is novel, and it is interesting the idea of mixing conditional independence-based methods with optimization approaches.
- While the use of first-order-logic formulas lead to a quite convoluted presentation, the authors' effort in describing step-by-step most pivotal passages is appreciable.

Weaknesses.

- The method requires computing conditional independence among all possible pairs and triples of distinct variables to define the loss function of the model. This might significantly affect performance in terms of computational time.
- Widely accepted metrics for causal discovery, such as F1 and SHD, are relegated to the Appendix where (i.) the method underperforms compared to the baselines (ii.) I would have expected the score-based baselines to have better results. However, this second point might not be a problem depending on the data-generating process adopted by the authors — please refer to the question.
- Not really a weakness, but please notice two typos "pioneerd" (line 96) and "Hammindg" (line 295).

---

> ### Author Rebuttal · Authors · 2025-07-31
>
> We thank Reviewer 523A for their detailed feedback and constructive criticism. We appreciate that the reviewer found our method "novel" and the core idea "interesting." The questions raised are important, and they will help us significantly improve the paper's discussion on scalability, evaluation, and its relationship with existing methods. We will also correct the typos noted.
>
> Below, we address the specific weaknesses and questions.
>
>
> > W2 & Q1. The method underperforms in terms of F1 and SHD. Can the authors justify this behaviour? The paper should also discuss more in detail its relation with more widely adopted metrics in the main body.
>
>
> A1. We appreciate the reviewer's thoughtful question. To address this concern, we direct the reviewer to **Appendix E.1**, where we provide a detailed analysis of the relationship between CI-MCC and traditional metrics such as F1 and SHD. This section explicitly motivates our choice of CI-MCC as the primary evaluation metric.
>
> The core objective of our method is to learn a DAG whose **low-order conditional independence (CI) statements** align with those observed in the data. CI-MCC directly measures this alignment, whereas F1 and SHD assess structural similarity, which is not always equivalent to CI accuracy. As discussed in **Appendix E.1**, structurally distinct DAGs can imply identical sets of low-order CI statements. Our empirical analysis demonstrates numerous cases where DAGs sampled by our method achieve high CI-MCC scores (indicating few CI errors) but exhibit widely varying F1/SHD scores. This underscores that while our method successfully achieves its intended objective, it does not necessarily optimize for structural metrics like SHD or F1.
>
> This behavior is inherent to causal discovery, as different graphs within the same *k*-Markov equivalence class can yield identical CI statements despite structural differences. **Furthermore, recent work by Petersen (2025) highlights limitations of F1 and SHD, showing they often fail to distinguish model performance with sufficient statistical significance.** Given these challenges, we prioritized CI-MCC as the main evaluation metric while retaining F1/SHD results in the Appendix for completeness.
>
> We acknowledge that our discussion of these metrics was confined to the Appendix in the initial submission. Thanks to the reviewer’s feedback, we will incorporate a concise paragraph in the main experimental section (Section 5) to explicitly highlight this distinction and direct readers to the detailed analysis in **Appendix E.1**.
>
> Petersen, A. H. (2025). *Are You Doing Better Than Random Guessing? A Call for Using Negative Controls When Evaluating Causal Discovery Algorithms*. UAI 2025.
>
>
> > W1 & Q2: The method requires computing conditional independence among all possible pairs and triples of distinct variables to define the loss function of the model. This might significantly affect performance in terms of computational time. How does DAGPA compare to PC and score-based methods in terms of computational time?
>
> A2. The reviewer raises an important point about scalability, and it is correct that the DAGPA implementation requires pre-computing a bounded number of conditional independence tests $O(d^3)$. While this can still be computationally intensive, using parallel computation we observe in practice that our method has manageable runtime in our experiment setting. Below, we present the wallclock empirical comparison of our method against the baselines.
>
> We note that the wall-clock time of our method DAGPA shown below is an **overestimate** of the actual time it took DAGPA to reach the best resulting solution. This is because DAGPA uses Bayesian sampling (specifically Discrete Langevian Proposal (DLP) sampler) to continue exploring the DAG space. Rather than taking the last sampled DAG as the algorithmic output, we kept the algorithm running, kept track of all DAGs sampled at each step, evaluated them using the metric using only the input data (Appendix C3), and returned the best one. In almost all cases, the algorithm could have been terminated earlier to return DAGs of equal quality to those shown in the paper’s experimental results.
>
>
> GPU: 24GB A10
> |  | PC | kPC (k=1) | kPC (k=2) | GES | NOTEARS (Linear) | NOTEARS (Nonlinear) | DAGMA (Linear) | DAGMA (Nonlinear) | **DAGPA (Ours)** |
> |---|---|---|---|---|---|---|---|---|---|
> | $d=10$ | 0.60 | 0.15 s | 0.32 s | 1.80 s | 1.36 s | 10.98 s | 1.26 s | 172.92 s | 1.5 h (1000 steps) |
> | $d=50$ | 2.98 s | 2.60 s | 1.97 s | 921.14 s | 19.66 s | 44.23 s | 4.35 s | 801.98 s | 10.5 h (500 steps) |
>
> In addition, we would like to highlight several practical techniques that can greatly enhance the computation efficiency in real-world deployment:
>
> - First, our differentiable d-separation framework is amenable to efficient matrix-vector operators that can benefit from **vectorizations**. This is what we did for the DAGPA implementation, where we leverage PyTorch tensor library to implement all forward passes (i.e. computation of differentiable d-separation scores) as tensor operations without explicit for-loops. This allows us to utilize CUDA GPUs to significantly improve the computation speed, similar to how the nonlinear version of NOTEARs may leverage GPUs.
>
>
> - Second, the all-pairs p-values computation can be massively parallelized and then cached and reused. This is particularly useful in the case where one wants to obtain multiple output causal graphs for the same input dataset to enhance solution robustness and exploration of the Markov equivalence class. In this case, the computation time for the all-pairs p-values step can be amortized as it can be reused in subsequent runs.
>
>
> - For future work, another possibility is to use minibatching. That is, for very large graphs where computing all CI statements is infeasible, our framework can be extended so that at each step, we could sample a random subset of CI statements to estimate the gradients, trading a small amount of gradient accuracy for a significant reduction in per-step cost.
>
>
> In the paper revision, we will add the empirical runtime analysis to the Appendix and in the main paper discussion section, emphasize which directions can yield scalability improvement worthy of future work to explore.
>
>
> > W2 & Q3. How is the pre-processing handled in your experimental pipeline?
>
>
> A3. Thank you for this interesting question regarding the work of Reisach et al. (2021). This allows us to highlight a key strength of our framework.
>
> 1. **Experimental Pipeline:** For our synthetic experiments, we used binary data, where normalization/scaling is not needed. For the real-world Sachs dataset, we used the standard discretized version preprocessed using the Hartemink discretization method, which converts the original continuous protein concentration measurements into 3-level categorical variables (low, average, high) while preserving the underlying dependence structure.
>
> 2. **Robustness of Our Method:** Our framework should be inherently more robust to pre-processing issues than other score-based methods. The issue highlighted by Reisach et al. is that methods like NOTEARS can "game" the benchmark by exploiting variance differences as a proxy for causality, a trick that fails when data is normalized. Our approach, however, relies on CI tests. Standard CI testers (like the Chi-squared test for binary data or Fisher's Z-test for continuous data) are robust to these kinds of scaling transformations. Therefore, DAGPA does not rely on such data artifacts and is expected to be more reliable across different pre-processing pipelines.
>
> We will add a paragraph to the experimental setup section clarifying our pre-processing steps and explaining why our constraint-based approach is less susceptible to these known pitfalls of score-based methods.
>
>
> > Q4. Can this approach be integrated as a sort of "regularization" technique to existing score-based differentiable methods?
>
>
> A4. Yes, absolutely. Our formulation of the differentiable d-separation framework should be general enough to support such a potential hybrid method, combining the strength of both constraint-based scores and likelihood score (such as NOTEARS’ reconstruction MSE loss). Below we provide a more detailed explanation of how such a hybrid method might be implemented.
>
>
> The main challenge to address is defining how a weighted adjacency matrix $\mathbf{W} \in [0, 1]^{d \times d}$ with edge weights in the range $[0, 1]$ can be derived from the model parameter $\theta$ in a way that is compatible with both our differentiable d-separation framework and the score-based method. Particularly to our method, we require the edge weights to fall in $[0, 1]$ as it needs to be valid probability measures.
>
> For instance, to convert the model parameter to our framework’s probabilistic edge weights, it should be possible to apply a sigmoid activation $\sigma(\cdot)$ for the NOTEARS linear model’s parameter $\theta$. For the nonlinear version of NOTEARS though, its graph weighted adjacency matrix, obtained by $\text{sqrt}({\theta^{(1)}\_j}^T \theta^{(1)}\_j)$ ($\theta^{(1)}$ the first MLP layer weight) consists of unnormalized, nonnegative values. Thus, one needs to find a proper way to normalize such a weight into range $[0, 1]$ to satisfy the probabilistic semantics. The rest is to simply add the prediction loss, such as the MSE loss $\mathcal{L}_{\text{MSE}}(\theta, X) = \frac{1}{2n} \| X - XW \|^2_2$ for the linear NOTEARS model, as an additional task loss and apply the same multi-task optimization routine of DAGPA.
>
>
>
>
> It is our belief that our work opens up the opportunity of an interesting research direction, and merits further study on possible ways to integrate with various existing score-based methods.

---

> > ### Comment · Reviewer_523A · 2025-08-05
> >
> > I thank the authors for their detailed response to my review, which answered most of my concerns.
> >
> > Even if it is an overestimate, the difference in terms of computational time is significant compared to the remaining methods. However, despite its limited applicability, the methodology is novel and interesting and encourages future works in the area. I would strongly recommend highlighting these results as a limitation of the proposal in a next revision of the paper.
> >
> > Overall, I am positively adjusting my evaluation according to this discussion.

---

> ### Author Response · Authors · 2025-08-06
>
> We sincerely thank Reviewer 523A for their detailed and constructive feedback throughout the review process, and for positively adjusting their evaluation.
>
> We completely agree that scalability is a critical aspect of our method's applicability. As requested, we will be sure to highlight the computational complexity as a limitation in the final version of the paper. In the revised Appendix, we will carefully present the complexity analysis and runtime comparison, detail the various engineering techniques we implemented to speed up the method (e.g. GPU acceleration w/ pytorch), and discuss potential future directions—such as sampling 1st-order CI statements to use in the loss function to reduce the number to at most O(d^2), alternative faster reachability computation or estimation—that can lead to greater scalability. We hope this transparent discussion will encourage future work to explore this fruitful avenue.
>
> We are very grateful for the reviewer's valuable insights, which have helped us improve the paper significantly.
>
> Thank you!

---

### Official Review · Reviewer_JM3s · 2025-06-30

**Clarity:** 2
**Significance:** 3
**Originality:** 3
**Rating:** 5
**Confidence:** 4

**Summary:**

This paper proposes a novel method to incorporate "(conditional) independence statements" into a differentiable causal discovery framework. The authors first express d-separation $S_A(x,y|z)$ using logical notation (Definition 3.1) and then introduce a continuous relaxation of this definition  $\tilde{S}_W(x,y|z)$ through soft logic. They establish a connection between the soft version and the "expected" d-separation statements, enabling the integration of causal inference with differentiable optimization techniques.

**Questions:**

How many CI statements are needed? If you need test all pairs, PC would be much faster? I can’t imagine a practical scenario where testing all pairs is feasible, especially when the number of nodes is large. This seems to contradict the motivation behind methods like NOTEARS, which aim to reduce computational complexity.


Comments:
1. The notation $\independent_D$ is somewhat unfamiliar to me. Typically, independence is discussed in the context of a joint distribution $P$, so this notation may need further clarification.
2. Is Theorem 3.2 a theorem or a definition？As far as I know， d-separation can be directly read off from the graph $A$ and is denoted by $\independent_A$. From my perspective, Thm.3.2 and Def.3.1  are the same thing.
3. Lemma 3.4 appear to be redundant in this draft and could be omitted to improve clarity and conciseness.
4. It took me some time to understand the exact method for learning the graph. It would be helpful to highlight the key method more clearly at the beginning of Section 3.
5. The mathematical notation is dense and may be difficult to follow for readers unfamiliar with the specific formalism used. Simplifying or breaking down the notation could improve readability.
6. Despite these points, I appreciate the idea of bridging constraint-based and score-based methods, which is a promising direction in causal discovery.

**Ethical Concerns:**

["NO or VERY MINOR ethics concerns only"]

**Final Justification:**

I've read the authors' response and will keep my score.

**Limitations:**

yes

**Quality:**

3

**Strengths And Weaknesses:**

Strengths
- Novelty: This is the first time I’ve seen a natural way to combine constraint-based and score-based methods, which is a significant contribution to the field.

Weaknesses
- The method description could be presented in a more straightforward way.
- Learning Time: The paper does not report or compare the learning time, which is crucial for evaluating the method's efficiency, especially in large-scale settings.
- Computational Cost: The method requires performing CI tests on all pairs of variables, which can be computationally expensive and may not scale well to large networks.
- Code is not available

---

> ### Author Rebuttal · Authors · 2025-07-31
>
> We thank Reviewer JM3s for their positive evaluation and for recognizing the novelty of our work as a "significant contribution to the field." We are grateful for the detailed and constructive comments on how to improve the paper's clarity and presentation. We agree with the reviewer's suggestions and will incorporate them into our revision.
> Below, we address the specific weaknesses and questions raised.
>
> **On Computational Cost, Scalability, and Learning Time (Weaknesses 2, 3 & Question 1)**
>
> A1. The reviewer raises an important point about the computational cost and scalability of our method, particularly the need to handle all low-order CI statements.
>
> 1. **Empirical Runtime Comparison**: The table below presents the wall-clock comparison of DAGPA against the baseline methods on the synthetic $d=10$ and $d=50$ experiment settings, which was the time these methods took to yield the results shown in the paper. We note that the wall-clock time of our method DAGPA shown below is an **overestimate** of the actual time it took DAGPA to reach the best resulting solution. This is because DAGPA uses Bayesian sampling (specifically Discrete Langevian Proposal (DLP) sampler) to continue exploring the DAG space. Rather than taking the last sampled DAG as the algorithmic output, we kept the algorithm running, kept track of all DAGs sampled at each step, evaluated them using the metric using only the input data (Appendix C3), and returned the best one. In almost all cases, the algorithm could have been terminated earlier to return DAGs of equal quality to those shown in the paper’s experimental results.
>
> GPU: 24GB A10
> |  | PC | kPC (k=1) | kPC (k=2) | GES | NOTEARS (Linear) | NOTEARS (Nonlinear) | DAGMA (Linear) | DAGMA (Nonlinear) | **DAGPA (Ours)** |
> |---|---|---|---|---|---|---|---|---|---|
> | $d=10$ | 0.60 | 0.15 s | 0.32 s | 1.80 s | 1.36 s | 10.98 s | 1.26 s | 172.92 s | 1.5 h (1000 steps) |
> | $d=50$ | 2.98 s | 2.60 s | 1.97 s | 921.14 s | 19.66 s | 44.23 s | 4.35 s | 801.98 s | 10.5 h (500 steps) |
>
> 2. Number of CI Statements & Comparison to PC: Our method considers all 0th-order ($O(d^2)$) and 1st-order ($O(d^3)$) CI statements, for a total of $O(d^3)$ constraints. While the PC algorithm may be faster on sparse graphs by adaptively pruning its search, our goal is different. DAGPA is designed for robustness in low-sample regimes, where PC's reliance on binary CI test results, obtained via hard-thresholding p-values, can be challenging. Our method's strength lies in using "soft" CI information from all low-order statements to guide a differentiable search, which, as our experiments show, leads to superior accuracy when data is limited. The trade-off is a higher computational cost for greater robustness.
>
> 3. Motivation vs. NOTEARS: We thank the reviewer for the opportunity to clarify our method’s motivations versus NOTEARS. While we do not claim to address the computational complexity of causal discovery problems as directly as NOTEARS did, we believe our method indeed has the following practical advantage in real-world settings.
>
>     -  **Differentiable Learning of k-Markov Equivalence Class**: Currently, there is no differentiable-learning-based approach similar to NOTEARS that can learn the k-Markov equivalence class in the low-sample regime, when we can only rely on low-degree CI tests since high-order ones are extremely unreliable. We argue that this is a significant shortcoming, since real-world tasks often have a limited number of samples, and the k-Markov equivalence class theoretically quantifies the ceiling of what can be learned under this scenario.
>
>     - **Interpretability via Conditional Independence (CI) Tests**: Our method leverages CI tests as its core mechanism, which can offer greater interpretability in some scenarios when compared to other score-based methods that rely on likelihoods. This could be particularly valuable in real-world applications where interpretability is critical.
>
>     - **Robustness in Low-Sample Regimes**: Unlike traditional score-based approaches, which often struggle with overfitting or underfitting when fitting functional models to scarce data, our method (DAGPA) incorporates "soft" CI information derived from p-values. This design choice enhances robustness in low-sample settings, as evidenced by its superior performance at $n = 100$ compared to baselines (Figure 3a). This highlights a practical advantage of DAGPA in data-limited scenarios.
>
>
> 4. Code Availability: The code is available. Please refer to Appendix D.1 in the supplementary zip file, in which we provide a link to an Anonymous repository containing all the code for DAGPA. Upon acceptance, we will replace this repository with a Github link.
>
>
>
> **On Method Description and Clarity (Weakness 1 & Comments)**
>
> A2. We appreciate the reviewer's detailed feedback on how to make our method easier to understand. We will implement the following changes in our revision:
>
> - **Comment 1: Notation for Independence:** Thank you for pointing this out. For simplicity of presentation, we used the graphical independence notation $\perp\perp$, which is used in works such as Pearl, (2009); Koller & Friedman (2009), to also express independence observed from the dataset $\mathcal{D}$. In order to improve readability, we will add a note on the equivalence between $P(x \mid y, z) = P(x \mid z)$ and the notation in the paper.
>
> - **Comment 2: Theorem 3.2 vs Definition 3.1:** We thank the reviewer for highlighting this important aspect of the presentation. Theorem 3.2 serves to establish the correctness of our newly proposed first-order logical characterization of low-order d-separation (Definition 3.1), which is a foundational contribution of this work. The proof demonstrates that the output values derived from the logical formulae in Definition 3.1 precisely match those obtained via traditional procedural combinatorial algorithms for d-separation. This equivalence is important, as it enables the continuous, differentiable relaxation introduced in Section 3.2—a capability not readily achievable with traditional, non-differentiable combinatorial methods.
>
>
> - **Comment 3: Redundancy of Lemma 3.4:** We agree that it may interrupt the flow of the text, so we will move it to Appendix in the revised paper. Although the correctness of the reachability FOL formulae in Definition 3.3 may seem intuitively straightforward, Lemma 3.4 is still a necessary building block for a mathematically rigorous argument, since we believe it formally justifies the adoption of the reachability predicate $R_{\mathbf{A}}$​ in our d-separation formulae.
>
> - **Comment 4 & 5: Highlighting the Method and Simplifying Notation:** We agree that the flow of Section 3 and a description and break-down of the key mathematical notation can be further improved. We will improve the "roadmap" paragraph at the beginning of Section 3 to make the method description more intuitive, and also review the entire section to add more intuitive explanations for our notation, such as a concrete running example, and break down complex formulae to improve readability.
>
> We are confident that these revisions will improve the clarity and accessibility of our paper. We thank the reviewer again for their supportive assessment and valuable suggestions.

---

### Official Review · Reviewer_pEMd · 2025-06-30

**Clarity:** 3
**Significance:** 4
**Originality:** 4
**Rating:** 6
**Confidence:** 4

**Summary:**

Differentiable Constraint‐Based Causal Discovery introduces a novel hybrid framework that marries the rigorous conditional‐independence testing of constraint‐based methods with the flexibility of gradient‐based optimization. The authors first show how low‐order d‐separation and d‐connection statements can be encoded as first‐order logical formula over graph reachability, then relax these discrete predicates into continuously differentiable “percolation” measures using probabilistic soft logic (LogLTN operators). This yields differentiable lower‐bounds on expected d‐separation/d‐connection that serve as losses alongside a log‐determinant acyclicity regularizer, forming the DAGPA algorithm. Empirical results on synthetic Erdős–Rényi and Scale‐Free graphs and the real‐world Sachs protein‐signaling dataset demonstrate DAGPA’s strong robustness in small‐sample regimes—surpassing both traditional constraint‐based (PC, k‐PC) and score‐based (NOTEARS, DAGMA) baselines—and competitive performance when ample data are available.

**Questions:**

I recommend adding the final optimization objective to the main text—showing concretely how the differentiable d-separation/d-connection losses and the acyclicity regularizer combine into a single formula will greatly improve readability.

It would be helpful if the authors could analyze the computational complexity of DAGPA and report empirical runtimes on representative benchmarks in the main paper. Are there opportunities to accelerate the method (e.g.\ parallelizing percolation calculations, exploiting sparsity in the graph, or using approximate reachability)?

The experimental metric plotted in Figures 2 and 3 (“Low-order CI Statement MCC vs. Empirical CDF”) is not self-explanatory. Could you share more details on this part.

**Ethical Concerns:**

["NO or VERY MINOR ethics concerns only"]

**Final Justification:**

I find the paper presents a very interesting idea—personally, it caught my eye. The authors’ comprehensive rebuttal has addressed my concerns. This paper is solid 5, but I’m raising my score to a 6 as encouragement.

**Limitations:**

YES

**Quality:**

4

**Strengths And Weaknesses:**

**Strengths**

1. The paper is exceptionally well written. Despite introducing many novel concepts and intricate notation, the authors clearly explain each idea and present their results in a coherent manner. To further aid comprehension, I suggest including a concrete example alongside each formal definition—particularly Definition 3.1—which currently feels under‐explained.

2. The motivation and core idea are both compelling and original. The authors demonstrate how to embed the d-separation/d-connection criterion as a differentiable scoring mechanism within a gradient-based optimization framework. This novel integration clearly merits an accept.

**Weaknesses**

As the authors acknowledge, the current approach only handles d-separation/d-connection for singleton conditioning sets. Extending this to larger conditioning sets would require exponentially many additional constraints, leading to severe computational challenges both in computing each individual score and in managing the objectives within the multi-task optimization.

---

> ### Author Rebuttal · Authors · 2025-07-31
>
> We are grateful to Reviewer pEMd for their positive and thorough review. We are delighted that the reviewer found our work "exceptionally well written" and the core idea "compelling and original," and we sincerely appreciate their championing of the paper. The reviewer suggestions are all constructive and will certainly help us improve the paper's clarity.
> Below, we address the reviewer's questions and suggestions.
>
> > Q1. I recommend adding the final optimization objective to the main text—showing concretely how the differentiable d-separation/d-connection losses and the acyclicity regularizer combined into a single formula will greatly improve readability.
>
> A1. We appreciate the constructive suggestion to enhance readability. In the revised manuscript, we will introduce a dedicated paragraph at the end of Section 4 to explicitly summarize all components of the multi-task loss function. While our method does not rely on a single unified formula due to its multi-objective nature, we will consolidate the individual loss terms into a centralized location for clarity. If space constraints prevent full inclusion in the main paper, we will provide explicit pointers to the detailed equations in the Appendix.
>
> > Q2. It would be helpful if the authors could analyze the computational complexity of DAGPA and report empirical runtimes... Are there opportunities to accelerate the method?
>
> A2. Thank you for this valuable suggestion. We will add a detailed analysis of complexity, runtimes, and acceleration opportunities in the Appendix. We will also emphasize the scalability limitation and point to promising avenues for future work in the revision.
>
> **Empirical Runtime Comparison**: The table below presents the wall-clock comparison of DAGPA against the baseline methods on the synthetic $d=10$ and $d=50$ experiment settings, which was the time these methods took to yield the results shown in the paper. We note that the wall-clock time of our method DAGPA shown below is an **overestimate** of the actual time it took DAGPA to reach the best resulting solution. This is because DAGPA uses Bayesian sampling (specifically Discrete Langevian Proposal (DLP) sampler) to continue exploring the DAG space. Rather than taking the last sampled DAG as the algorithmic output, we kept the algorithm running, kept track of all DAGs sampled at each step, evaluated them using the metric using only the input data (Appendix C3), and returned the best one. In almost all cases, the algorithm could have been terminated much earlier to return DAGs of equal quality to those shown in the paper’s experimental results.
>
> GPU: 24GB A10
> |  | PC | kPC (k=1) | kPC (k=2) | GES | NOTEARS (Linear) | NOTEARS (Nonlinear) | DAGMA (Linear) | DAGMA (Nonlinear) | **DAGPA (Ours)** |
> |---|---|---|---|---|---|---|---|---|---|
> | $d=10$ | 0.60 | 0.15 s | 0.32 s | 1.80 s | 1.36 s | 10.98 s | 1.26 s | 172.92 s | 1.5 h (1000 steps) |
> | $d=50$ | 2.98 s | 2.60 s | 1.97 s | 921.14 s | 19.66 s | 44.23 s | 4.35 s | 801.98 s | 10.5 h (500 steps) |
>
>
>
> **Computational Complexity:** The computational bottleneck of our approach (DAGPA) is the p-values computation for all low-order CI statements and the reachability computation required by the differentiable d-separation formulae.
>
> The complexity of p-values computation (which we GPU-accelerate) depends on the specific choice of statistical independence test. Take the Chi-squared test for example. Given $n$ data points, computing each single Chi-squared p-value takes $O(n)$ as this is the time required for iterating over all data points and building the contingency. Thus, the overall time required is $O(d^3 n)$ as we obtain a total of $d^2 + d^3$ number of p-values for the unconditional and first-order conditional statements.
>
> For the differentiable d-separation formulae: the reachability subroutine involves a maximum of $d$ recursive steps, and each all-pairs Bellman-Ford update step operates on $O(d^3)$ matrix entries (equation (7)). Thus, the total computation time is $O(d^4)$. The resulting all-pairs reachability matrix will be cached and accessed by the d-connection/d-separation score computation. Now, since in the 1st-order d-connection/d-separation scores (equation (3) and (5)), we need reachability on the node-deleted subgraph $\mathbf{A}_{-z}$ for all conditioning node $z$, this requires a total of $d+1$ all-pairs reachability matrices, thus rendering the total runtime of this part $O(d^5)$.
>
> For the 0th-order d-separation/d-connection formulae (equation (2) and (4)), computing $S^{(0)}\_{\mathbf{A}} (x, y)$ or $C^{(0)}\_{\mathbf{A}} (x, y)$ for each (x, y) takes $O(d)$ time, since it iterates over $d$ possible common ancestors and accessing the reachability cache takes $O(1)$. Thus, computing all-pairs 0th-order d-separation/d-connection scores take $O(d^3)$ time. Now, since later in the 1th-order formulae, we need all-pairs 0th-order d-separation/d-connection scores for the node-deleted subgraphs $A_{-z}$ for all conditioning node $z$, this adds another $d$ dimension. Thus, the total time is $O(d^4)$.
>
> For the 1st-order d-separation/d-connection formulae (equation (3) and (5)), similarly, computing the result for each query triple $(x, y \mid z)$ takes $O(d)$ time. Thus, getting the results for all triple of nodes takes a total of $O(d^4)$ computations.
>
> Thus in summary, computing all-pairs p-values takes $O(d^3 n)$ computations, computing reachability matrix (and cache for later use) takes $O(d^5)$ computations, and computing 0th-order and 1st-order d-separation/d-connection scores each take $O(d^4)$ computations. The total will be dominated by the reachability computation, which is $O(d^5)$.
>
> **Acceleration Opportunities:** That being said, there are ample promising opportunities for acceleration.
>
> - First, we have to note that all the computations of reachability and d-separation/d-connection scores are matrix-vector operations, which can benefit from **GPUs**. This is what we did in our implementation, where we leverage PyTorch GPU tensor library for all such computations, avoiding any explicit for-loops, significantly improving the speed of our algorithm.
>
> - Second, in practice one can limit the reachability computation to only consider **paths of a constant length $k$**, reducing the time complexity from $O(d^5)$ to $O(d^4 k)$, **rendering the entire pipeline $O(d^4)$**. This is what we did in the experiments for the larger graphs with $d = 50$ number of nodes, where we limit the path length to $k = 10$. Our earlier observation during earlier method development suggest negligible performance degradation with significantly improved computational efficiency.
>
> - In addition, the all-pairs p-values computation can be massively parallelized and then cached and reused. This is particularly useful in the case where one wants to obtain multiple output causal graphs for the same input dataset to enhance solution robustness and exploration of the Markov equivalence class. In this case, the computation time for the all-pairs p-values step can be amortized as it can be reused in subsequent runs.
>
> - Finally, we expect future work to further examine how to incorporate sparsity assumptions into the graph to further reduce the time complexity. For example, instead of assuming each node can connect to all $d-1$ other nodes, one reasonable assumption is to restrict to a maximum of constant $k$ degree. In that case, the reachability computation subroutine computation can drastically speed up.
>
>
>
>
> > Q3. The experimental metric plotted in Figures 2 and 3 (“Low-order CI Statement MCC vs. Empirical CDF”) is not self-explanatory.
>
> A3. We appreciate the reviewer’s feedback regarding the clarity of the metric plotted in Figures 2 and 3. We recognize that this visualization may not be immediately intuitive to all readers. In the revised manuscript, we will provide a more detailed explanation of this metric.
> The **Low-order CI Statement Matthews Correlation Coefficient (CI-MCC)** metric, illustrated in **Figure 1**, evaluates how well a model's predicted graph aligns with the ground-truth graph in terms of the low-order conditional independencies they imply. The process is as follows:
> For a predicted graph $G_{\text{pred}}$​ and the ground-truth graph $G_{\text{true}}$​, we generate two lists containing all possible 0th- and 1st-order d-separation statements (e.g., $A \perp\perp B$, $A \perp\perp C \mid D$, etc.).
> We treat this as a binary classification task: the goal is for $G_{\text{pred}}$ to correctly classify which d-separation statements hold true in $G_{\text{true}}$​.
> We then compute the standard **Matthews Correlation Coefficient (MCC)** on the resulting confusion matrix (TP, TN, FP, FN). MCC is a robust metric for binary classification that is well-suited for cases with imbalanced classes (as the number of true independencies vs. dependencies can vary greatly in practice).
> The plots in Figures 2 and 3 show the **Empirical Cumulative Distribution Function (ECDF)** of these CI-MCC scores over multiple runs, which visualizes the distribution of performance for each method.
> We will revise the first paragraph of Section 5 and the captions for Figures 2 and 3 to include this clear, step-by-step explanation of the CI-MCC metric and the ECDF plots.
>
>
> References:
>
> [1] "Gradient surgery for multi-task learning." NeurIPS 2020.

---

> > ### Comment · Reviewer_pEMd · 2025-08-01
> >
> > The authors’ rebuttal is thorough. I find the paper strong and novel, introducing an interesting idea with clear potential for further exploration. Given the dense notation and new definitions, I encourage incorporating the suggested changes to improve readability. All of my concerns have been satisfactorily addressed, and I am raising my score.

---

> > > ### Author Response · Authors · 2025-08-02
> > >
> > > Thank you!
> > >
> > > We are truly grateful to Reviewer pEMd for their thoughtful engagement during the discussion period and for their positive assessment of our work. We sincerely thank them for their time and for raising their score. We will be sure to incorporate their excellent suggestions to improve the paper's readability in the final version.
> > >
> > > If the reviewer has any additional questions, please feel free to let us know. We are more than happy to respond promptly.

---

### Official Review · Reviewer_TmCS · 2025-06-30

**Clarity:** 3
**Significance:** 3
**Originality:** 3
**Rating:** 4
**Confidence:** 3

**Summary:**

This paper studies a new "style" of gradient-based causal discovery approach, where the gradient based optimization incorporates CI constraints as in constraints-based CD methods. This differs from the usual, existing score based approach, which encodes DAG in the constraint. The paper derives theoretical sounds showing soundness. The paper also demonstrates empirically that the method developed works particularly well in low sample settings, showing it is more robust in such settings.

**Questions:**

1. Can the authors comment more about the convergence of the (joint) optimization of the proposed objective function? This will make the paper more complete.

2. Are there other replacements for the p-values, which are used as heuristic conditional independence measures?

3. Is it correct to say that the approach developed in this paper can complement any existing score based approach? Specifically it can be added on top of any gradient-based method in that one can jointly optimize this paper's criterion and the criterion of the existing score based method. If yes, would this be better and if no, why not?

**Ethical Concerns:**

["NO or VERY MINOR ethics concerns only"]

**Final Justification:**

I think the paper is new and interesting, but there is some unknown about its convergence properties so it's a bit unclear if it can actually converge to the desirable optimum. As I said before, my overall rating is positive and I think the paper should be acepted.

**Limitations:**

Please see weakness section.

**Quality:**

3

**Strengths And Weaknesses:**

Strength: The paper's motivation is interesting, aiming to produce a differentiable, constraint based causal discovery method. This could potentially lead to new lines of work just like how NOTEAR opened up a lot of directions and areas for improvement. I thought the paper did a nice job building up from d-separation and S_A to a main theoretical result (Lemma 4.2). The experiments do compare fairly against known baselines like NOTEARS, GES and PC (but the experiments are specifically evaluating for recovery of d-separation statements implied by the true underlying causal structure that generated the data, which is what the method optimizes).

Weakness: With that said, I think more could be said about the optimization guarantees when optimizing the five multi-task loss functions (which was done in NOTEARS). I think the paper could also say more to motivate the necessity of having explicitly encoded constraint based approaches (what are some applications where this is important and NOTEARS is not sufficient).

---

> ### Author Rebuttal · Authors · 2025-07-31
>
> We sincerely thank Reviewer TmCS for their positive and insightful feedback. We are encouraged that the reviewer found our motivation "interesting" and believes our work "could potentially lead to new lines of work just like how NOTEARS opened up a lot of directions." We appreciate the thoughtful questions, which will help us significantly improve the clarity and completeness of our paper.
> Below, we address the specific weaknesses and questions raised.
>
> > Q1. Can the authors comment more about the convergence of the (joint) optimization of the proposed objective function?
>
> **A1.** Thank you for this insightful question. We can provide a convergence analysis for our proposed objective function under certain conditions, which we can include in the revised paper. Below, we outline the key theoretical and practical arguments supporting the convergence of DAGPA.
>
> Theoretically, DAGPA leverages the **Discrete Langevin Proposal (DLP)** sampler [1] to perform posterior sampling of DAGs from the true posterior distribution, whose energy function is defined as the sum of our five multi-task losses. Under the DLP framework, we can establish asymptotic convergence to the true posterior, provided that gradients of the objectives are added rather than processed via PCGrad. This convergence result stems from two key properties of DLP:
> 1. **Metropolis-Hastings (MH) rejection step**: Ensures that the sampler respects the true posterior distribution.
> 2. **Ergodicity of the Markov chain**: Guarantees that every possible DAG has a non-zero probability of being sampled, satisfying the conditions for asymptotic convergence as per [1].
>
>
> Additionally, **Lemma 4.2** in our paper establishes consistency, showing that as the sample size increases and the soft-logic approximation becomes exact, the true underlying DAG minimizes each loss component and thus has the highest posterior probability.
>
> Practically, we adopt several strategies to enhance mixing (but of which we cannot evaluate their effectiveness):
> 1. **Acyclicity constraint**: We use the log-determinant characterization from DAGMA [2], which offers better-behaved gradients compared to NOTEARS, particularly when cycles are present. This choice ensures stable optimization even in complex graph structures.
> 2. **Gradient conflict resolution**: PCGrad [3] is employed to navigate conflicting gradients across tasks. Unlike naive loss combination methods, PCGrad projects gradients onto a consensus direction, provably improving convergence in common scenarios [4]. This is important for multi-task objectives, where gradients may pull the model in opposing directions.
> 3. **Enhanced exploration**: DLP enables parallel edge updates during sampling, significantly accelerating exploration of the parameter space. This is particularly valuable for causal discovery due to the non-uniqueness of solutions (e.g., Markov equivalence classes).
>
> While quantifying an exact convergence rate under our multi-task losses remains a theoretical challenge, we address this by combining rigorous Bayesian guarantees with battle-hardened optimization practices. We will expand on these points in the revised paper and encourage future work to further analyze the complex loss landscape of CI-statement losses. We appreciate the reviewer’s feedback and look forward to incorporating these discussions.
>
>
> > Q2. I think the paper could also say more to motivate the necessity of having explicitly encoded constraint based approaches
>
> A2. We thank the reviewer for their insightful feedback. We will revise the paper to better emphasize the unique advantages of our approach, as outlined below.
>
> 1. **Differentiable Learning of k-Markov Equivalence Class**: Currently, there is no differentiable-learning-based approach similar to NOTEARS that can learn the k-Markov equivalence class in the low-sample regime, where we can only rely on low-degree CI tests since high-order ones are unreliable. We argue that this is a significant shortcoming, since real-world tasks often have a limited number of samples, and the k-Markov equivalence class theoretically quantifies the ceiling of what can be learned under this scenario.
>
> 2. **Interpretability via Conditional Independence (CI) Tests**: Our method leverages CI tests as its core mechanism, which can offer greater interpretability in some scenarios when compared to other score-based methods that rely on likelihoods. This could be particularly valuable in real-world applications where interpretability is critical.
>
> 3. **Robustness in Low-Sample Regimes**: Unlike traditional score-based approaches, which often struggle with overfitting or underfitting when fitting functional models to scarce data, our method (DAGPA) incorporates "soft" CI information derived from p-values. This design choice enhances robustness in low-sample settings, as evidenced by its superior performance at $n = 100$ compared to baselines (Figure 3a). This highlights a practical advantage of DAGPA in data-limited scenarios.
>
> We appreciate the reviewer’s emphasis on the paper’s significance and will strengthen the introduction to explicitly articulate these points, thereby improving the overall motivation and clarity of the work.
>
>
>
> > Q3. Are there other replacements for the p-values, which are used as heuristic conditional independence measures?
>
> Yes, indeed. Our framework happily accepts any CI measurement $p_{\mathcal{D}}$ from input data as long as it satisfies the semantic that $p_{\mathcal{D}} \approx 1$ indicates (likely) independence, and $p_{\mathcal{D}} \approx 1$ indicates (likely) dependence. $p$-values fits this requirement; an alternative choice is the **conditional mutual information (CMI)** with some adequate conversion.
>
> Specifically, mutual information and conditional mutual information are always non-negative quantities, with a value of 0 indicating independence and values greater than 0 indicating dependence. To convert to the aforementioned definition of the CI measurement $p_{\mathcal{D}}$, one could adopt some form of indicator function. One candidate choice is the following function:
> $$
> f(x) = 1 - e^{-x / l} ,
> $$
> where $l$ is a hyperparameter controlling the sharpness such that when $l \to 0$, $f(x) \to 1[x > 0]$ approximates the hard indicator function that checks if input $x$ is greater than 0. Thus, given a raw CMI measurement $\text{CMI}\_{\mathcal{D}}(x, y, \mid z)$ for a triple of variables $x, y, z$, $f(\text{CMI}_{\mathcal{D}}(x, y, \mid z))$ becomes a valid CI measurement ready to be used in our proposed framework.
>
> Although conceptually such an approach makes sense, during earlier method development we found that these converted CMI values are somewhat unreliable, as the values corresponding to the true independent statements vary a lot. Using these converted CMI values as input to the algorithm makes the method fragile and decreases the chances of sampling good DAG solutions. In addition, it introduces yet another hyperparameter $l$ that one needs to tune. In contrast, the p-values varied less and were more stable, and we were working with one less hyperparameter. Thus, we decided to adopt p-values for the paper. But CMI remains a promising avenue that we intend to explore further.
>
> We thank the reviewer for bringing up this point, and we will add this discussion in the revised Appendix.
>
> > Q4. Is it correct to say that the approach developed in this paper can complement any existing score based approach?
>
> Yes, we believe such a hybrid implementation is possible. The main challenge to address is defining how a weighted adjacency matrix $\mathbf{W} \in [0, 1]^{d \times d}$ with edge weights in the range $[0, 1]$ can be derived from the model parameter $\theta$ in a way that is compatible with both our differentiable d-separation framework and the score-based method. Particularly to our method, we require the edge weights to fall in $[0, 1]$ as it needs to be valid probability measures.
>
> For instance, to convert the model parameter to our framework’s probabilistic edge weights, it should be possible to apply a sigmoid activation $\sigma(\cdot)$ for the NOTEARS linear model’s parameter $\theta$. For the nonlinear version of NOTEARS though, its graph weighted adjacency matrix, obtained by $\text{sqrt}({\theta^{(1)}\_j}^T \theta^{(1)}\_j)$ ($\theta^{(1)}$ the first MLP layer weight) consists of unnormalized, nonnegative values. Thus, one needs to find a proper way to normalize such a weight into range $[0, 1]$ to satisfy the probabilistic semantics. The rest is to simply add the prediction loss, such as the MSE loss $\mathcal{L}_{\text{MSE}}(\theta, X) = \frac{1}{2n} \| X - XW \|^2_2$ for the linear NOTEARS model, as an additional task loss and apply the same multi-task optimization routine of DAGPA.
>
> Due to the limited time of the rebuttal period, we were not able to conduct comprehensive analysis of a concrete implementation of such a hybrid method. We will however emphasize this discussion in the revised main text and call for future work to investigate this promising research direction.
>
> References:
>
> [1] "A langevin-like sampler for discrete distributions." PMLR 2022
>
> [2] "Dagma: Learning dags via m-matrices and a log-determinant acyclicity characterization." NeurIPS 2022.
>
> [3] "Gradient surgery for multi-task learning." NeurIPS 2020.
>
> [4] "Revisiting scalarization in multi-task learning: A theoretical perspective." NeurIPS 2023.
>
> [5] "Learning sparse nonparametric dags." PMLR 2020

---

> > ### Comment · Reviewer_TmCS · 2025-08-02
> > **Reply**
> >
> > I thank the author for their detailed reply and I am happy to hear that the review helped with improving the paper. I was just mainly thinking that it would nice if the paper could emulate the set of results as in the NOTEARS paper, which included a pretty clear-cut convergence result.
> >
> > Based on the author's reply, it seems like the optimization problem here is more complicated, but some guarantees can be achieved certain assumptions. It's a bit hard to assess (at least for me) essentially how easily and well can the optimization can work across different settings.
> >
> > Also, I also looked at the reviewer's pointer to Lemma 4.2. This lemma says the true DAG is the minimizer, but it does not imply any other DAG cannot *also* be the true minimizer, no? So the more accurate statement would be that asymptotically, the true DAG has "one of the highest posterior probability", but not necessarily the only one? Please correct me if I'm mistaken.
> >
> > With all this said, on the balance, I think the paper is interesting and I feel positive about the paper.

---

> ### Author Response · Authors · 2025-08-04
>
> We sincerely thank the reviewer for their continued engagement and for the thoughtful follow-up. This dialogue is incredibly helpful for us to refine the paper's core message.
>
> We agree that a formal convergence analysis focusing on the convergence rate is an important and valuable research direction. Unlike NOTEARS that relies on the reconstruction MSE loss and matrix-exponential-based acyclicity loss, which permits the use of standard convex optimization routine (Augmented Lagrangian method), our multi-objective framework presents unique challenges. The "true positive" loss (encouraging d-separation) and "true negative" loss (encouraging d-connection) naturally pull the optimization in conflicting directions. A formal analysis on how fast the optimization could converge would require a deep dive into the gradient landscape of this complex interplay. We believe this is a significant and promising area for future work and hope our paper inspires further investigation into the optimization dynamics of constraint-based differentiable discovery methods.
>
> Regarding your second point on Lemma 4.2, you are correct—the more rigorous statement is that the true DAG is *one of* the minimizers of our objective, not necessarily the unique one. Under the assumptions of Lemma 4.2, the set of optimal solutions corresponds to the **k-Markov Equivalence Class** (where for DAGPA, $k=1$). This is an inherent property of our method: any graph within this class shares the same low-order CI statements and will therefore achieve the same minimal loss. Conversely, any graph outside this class will violate at least one CI statement, resulting in a strictly higher loss.
>
> We are very grateful for this sharp observation. In our revision, we will revise the wording around Lemma 4.2 and the aforementioned detailed discussion on the convergence in the Appendix to make this important distinction clear. Thank you again for your positive outlook on the paper and for the constructive dialogue. Your feedback has been immensely invaluable.

---

> > ### Comment · Reviewer_TmCS · 2025-08-07
> > **Reply**
> >
> > I thank the author for the clarification and the transparent response, which helps to access the scope of the contribution of the paper. It's good that the qualifiers in the (theoretical) statements are made precise. My assessment of the paper remains positive.

---

### Note · Authors · 2025-08-16

Dear reviewers, AC, SAC, and PC members,

We sincerely thank all reviewers for their exceptionally thorough and constructive feedback. We are delighted by the consensus that our work introduces a novel, compelling, and significant new direction for causal discovery by bridging constraint-based methods with gradient-based optimization via differentiable d-separation.

The author-reviewer discussion was invaluable. It allowed us to provide a detailed computational complexity analysis and empirical runtimes comparison, clarifying the practical trade-offs of our approach. We also elaborated on our evaluation philosophy, justifying our focus on the CI-MCC metric, and explored promising future directions, such as integrating our framework with score-based methods like NOTEARS.

We are grateful that our responses and clarifications successfully addressed the reviewers' initial concerns, leading to a positive adjustment of scores from Reviewers pEMd and 523A.

As promised, the final version will be significantly strengthened by incorporating this feedback. We will enhance the paper's clarity by adding concrete illustrating examples for the FOL d-separation formulae, providing a more detailed explanation of our CI-MCC metric, and making our theoretical claims more comprehensive. We will also include a comprehensive computational analysis and transparently discuss scalability as a limitation and an important avenue for future work.

We believe the enthusiastic reception of the paper's core ideas, combined with the successful resolution of all raised concerns, underscores its value and readiness for the NeurIPS community. Thank you again for your time and consideration.

Sincerely, The authors

---

### Decision · Program_Chairs · 2025-09-17

**Decision:**

Accept (poster)

**Comment:**

The paper introduces a new route for causal discovery that optimizes conditional-independence (CI) constraints with gradients by making d-separation differentiable. The authors (1) give formal first-order logic characterizations of 0th- and 1st-order d-separation and prove correctness, (2) relax these logical formulas into continuously differentiable quantities using LogLTN soft logic, yielding lower-bound estimators of expected d-separation over random DAGs, and (3) instantiate the idea in DAGPA, which learns a weighted adjacency by minimizing multi-task losses that align soft d-separation/connection with CI evidence while enforcing acyclicity. Empirically, the method is robust in low-sample regimes and outperforms popular baselines.

Strengths：
All reviewers agree that the paper is interesting and could potentially lead to new lines of work.
Reviewer 523A further emphasizes that the better results obtained in the paper compared to baselines. Reviewer web4 appreciates that the paper presents the results in a coherent and step-by-step manner.


Weaknesses：
1）Relies on CI p-values as heuristics. The training signal uses p-values from Fisher-z tests (even on nonlinear data like Sachs), which can inject bias/mismatch between the statistical test and the ground-truth mechanism.
2）Scope limited to low-order conditioning (|Z| ≤ 1). This is practical and well motivated, but it narrows identification power. Scaling the relaxation to higher-order CI would require exponentially many additional constraints, leading to severe computational challenges.

The most important reasons for the decision：
1）A clear, principled bridge between constraint-based and continuous optimization methods via a novel, well-justified relaxation of d-separation;
2）A practical, consistent learning framework that operationalizes the idea with acyclicity and multi-task CI losses, validated on synthetic and real data.

Summarization of the discussion and changes during the rebuttal period:
Some reviewers raised concerns about the computational complexity and empirical runtimes of the method, which the authors addressed with additional experiments and analysis in the rebuttal.

Overall comments:

After the rebuttal phase, the paper received one strong accept, one accept, and two borderline accepts, resulting in unanimous acceptance. The paper introduces a new way to combine constraint-based and score-based causal discovery methods. All reviewers agree that the proposed methods are interesting. Overall, the quality of the work is high and solid, and it is likely to be of interest to the community. Therefore, I suggest to accept this paper as a poster.